# Laplacian Canonization: A Minimalist Approach to Sign and Basis Invariant Spectral Embedding

**Jiangyan Ma[1]***     **Yifei Wang[2]***     **Yisen Wang[3,4]†**

[1] School of Electronics Engineering and Computer Science, Peking University
[2] School of Mathematical Sciences, Peking University
[3] National Key Lab of General Artificial Intelligence,
School of Intelligence Science and Technology, Peking University
[4] Institute for Artificial Intelligence, Peking University

## Abstract

Spectral embedding is a powerful graph embedding technique that has received a lot of attention recently due to its effectiveness on Graph Transformers. However, from a theoretical perspective, the universal expressive power of spectral embedding comes at the price of losing two important invariance properties of graphs, sign and basis invariance, which also limits its effectiveness on graph data. To remedy this issue, many previous methods developed costly approaches to learn new invariants and suffer from high computation complexity. In this work, we explore a minimal approach that resolves the ambiguity issues by directly finding canonical directions for the eigenvectors, named Laplacian Canonization (LC). As a pure pre-processing method, LC is light-weighted and can be applied to any existing GNNs. We provide a thorough investigation, from theory to algorithm, on this approach, and discover an efficient algorithm named Maximal Axis Projection (MAP) that works for both sign and basis invariance and successfully canonizes more than 90% of all eigenvectors. Experiments on real-world benchmark datasets like ZINC, MOLTOX21, and MOLPCBA show that MAP consistently outperforms existing methods while bringing minimal computation overhead. Code is available at https://github.com/PKU-ML/LaplacianCanonization.

## 1 Introduction

Despite the popularity of Graph Neural Networks (GNNs) for graph representation learning [55, 56, 54, 27, 13], it is found that many existing GNNs have limited expressive power and cannot tell the difference between many non-isomorphic graphs [57, 35]. Existing approaches to improve expressive power, such as high-order GNNs [35] and subgraph aggregation [9], often incur significant computation costs over vanilla GNNs, limiting their use in practice. In comparison, a simple but effective strategy is to use discriminative node identifiers. For example, GNNs with random features can lead to universal expressive power [44, 1]. However, these unique node identifiers often lead to the loss of the permutation invariance property of GNNs, which is an important inductive bias of graph data that matters for sample complexity and generalization [24]. Therefore, during the pursuit of more expressive GNNs, we should also maintain the invariance properties of graph data. Balancing these two conflicting demands presents a significant challenge for the development of advanced GNNs.

Spectral embedding (SE) is a classical approach to encode node positions using eigenvectors $U$ of the Laplacian matrix $L$, which has the advantage of being expressive and permutation equivariant.

---

*Equal Contribution.

†Corresponding Author: Yisen Wang (yisen.wang@pku.edu.cn)

37th Conference on Neural Information Processing Systems (NeurIPS 2023).

Table 1: Comparison between prior works and our method. $n$ is the number of nodes, $m$ is the exponent of the feature dimension of BasisNet [29, 33].

| Method | pre-processing time | universality | permutation invariance | addresses sign ambiguity | addresses basis ambiguity | feature dimension |
|---|---|---|---|---|---|---|
| LapPE [17] | $\mathcal{O}(n^3)$ | ✘ | ✔ | ✘ | ✘ | $n$ |
| RandSign [17] | $\mathcal{O}(n^3)$ | ✘ | ✔ | ✔ | ✘ | $n$ |
| SAN [25] | $\mathcal{O}(n^4)$ | ✔ | ✔ | ✔ | ✘ | $3n$ |
| SignNet [29] | $\mathcal{O}(n^3)$ | ✔ | ✔ | ✔ | ✘ | $2n$ |
| BasisNet [29] | $\mathcal{O}(n^3)$ | ✔ | ✔ | ✔ | ✔ | $n^m$ |
| MAP (ours) | $\mathcal{O}(n^3)$ | ✔ | ✔ | ✔ | ✔ | $n$ |

GNNs using SE can attain universal expressive power (distinguishing *any* pair of non-isomorphic graphs) even under simple architectures (results in Section 2). However, spectral embedding faces two additional challenges in preserving graph invariance: sign ambiguity and basis ambiguity, due to the non-uniqueness of eigendecomposition. These ambiguities could lead to inconsistent predictions for the same graph under different eigendecompositions.

Many methods have been proposed to address sign and basis ambiguities of spectral embedding. A popular heuristic is RandSign [17] which randomly flips the signs of the eigenvectors during training. Although it is simple to use, this data augmentation approach does not offer any formal invariance guarantees and can lead to slower convergence due to all possible $2^n$ sign flips. An alternative involves using sign-invariant eigenfunctions to attain sign invariance [25], which significantly increases the time complexity to $\mathcal{O}(n^4)$. Another solution is to design extra GNN modules for sign and basis invariant embeddings, *e.g.*, SignNet and BasisNet [29], which can also add a substantial computational burden. Therefore, as summarized in Table 1, existing spectral embedding methods are all detrimental in a certain way that either hampers sign and basis invariance, or induces large computational overhead. More discussions about the related work can be found in Appendix B.

In this work, we explore a new approach called Laplacian Canonization (LC) that resolves the ambiguities by identifying a unique canonical direction for each eigenvector, amongst all its sign and basis equivalents. Although it is relatively easy to find a canonization rule that work for certain vectors, up to now, there still lacks a rigorous understanding of what kinds of vectors are canonizable and whether we could find a complete algorithm for all canonizable features. In this paper, we systematically answer this problem by developing a general theory for Laplacian canonization and characterizing the sufficient and necessary conditions for sign and basis canonizable features.

Based on these theoretical properties, we propose a practical canonization algorithm for sign and basis invariance, called Maximal Axis Projection (MAP), that adopts the permutation-invariant axis projection functions to determine the canonical directions. We theoretically characterize the conditions under which MAP can guarantee sign and basis canonization, and empirically verify that this condition holds for most synthetic and real-world graphs. It is worth noting that LC is a lightweight approach since it is only a pre-processing method and does not alter the dimension of the spectral embedding. Empirically, we show that employing the MAP-canonized spectral embedding yields significant improvements over the vanilla RandSign approach, and even matches SignNet on large-scale benchmark datasets like OGBG [20]. We summarize our contributions as follows:

- We explore Laplacian Canonization (LC), a new approach to restoring the sign and basis invariance of spectral embeddings via determining the canonical direction of eigenvectors in the pre-processing stage. We develop a general theoretical framework for LC and characterize the canonizability of sign and basis invariance.

- We propose an efficient algorithm for Laplacian Canonization, named Maximal Axis Projection (MAP), that works well for both sign and basis invariance. In particular, we show that MAP-sign is capable of canonizing all sign canonizable eigenvectors and thus is complete. The assessment of its feasibility shows that MAP can effectively canonize almost all eigenvectors on random graphs and more than 90% eigenvectors on real-world datasets.

- We evaluate the MAP-canonized spectral embeddings on graph classification benchmarks including ZINC, MOLTOX21 and MOLPCBA, and obtain consistent improvements over previous spectral embedding methods while inducing the smallest computational overhead.

## 2 Benefits and Challenges of Spectral Embedding for GNNs

Denote a graph as $\mathcal{G} = (\mathbb{V}, \mathbb{E}, \boldsymbol{X})$ where $\mathbb{V}$ is the vertex set of size $n$, $\mathbb{E}$ is the edge set, and $\boldsymbol{X} \in \mathbb{R}^{n \times d}$ are the input node features. We denote $\boldsymbol{A}$ as the adjacency matrix, and let $\hat{\boldsymbol{A}} = \boldsymbol{D}^{-\frac{1}{2}} \tilde{\boldsymbol{A}} \boldsymbol{D}^{-\frac{1}{2}} = \boldsymbol{D}^{-\frac{1}{2}} (\boldsymbol{I} + \boldsymbol{A}) \boldsymbol{D}^{-\frac{1}{2}}$ be the normalized adjacency matrix, where $\boldsymbol{I}$ denotes the augmented self-loop, $\boldsymbol{D}$ is the diagonal degree matrix of $\tilde{\boldsymbol{A}}$ defined by $D_{i,i} = \sum_{j=1}^{n} \tilde{A}_{i,j}$. A graph function $f([\boldsymbol{X}, \hat{\boldsymbol{A}}])$ is *permutation invariant* if for all permutation matrix $\boldsymbol{P} \in \mathbb{R}^{n \times n}$, we have $f([\boldsymbol{P}\boldsymbol{X}, \boldsymbol{P}\hat{\boldsymbol{A}}\boldsymbol{P}^{\top}]) = f([\boldsymbol{X}, \hat{\boldsymbol{A}}])$. Similarly, $f$ is *permutation equivariant* if $f([\boldsymbol{P}\boldsymbol{X}, \boldsymbol{P}\hat{\boldsymbol{A}}\boldsymbol{P}^{\top}]) = \boldsymbol{P}f([\boldsymbol{X}, \hat{\boldsymbol{A}}])$.

**Spectral Embedding (SE).** Considering the limited expressive power of MP-GNNs, recent works explore more flexible GNNs like graph Transformers [58, 62, 16, 25, 42]. These models bypass the explicit structural inductive bias in MP-GNNs while encoding graph structures via positional embedding (PE). A popular graph PE is spectral embedding (SE), which uses the eigenvectors $\boldsymbol{U}$ of the Laplacian matrix $\boldsymbol{L} = \boldsymbol{I} - \hat{\boldsymbol{A}}$ with eigendecomposition $\boldsymbol{L} = \boldsymbol{U}\boldsymbol{\Lambda}\boldsymbol{U}^{\top}$, where $\boldsymbol{\Lambda} = \mathrm{diag}(\boldsymbol{\lambda})$ is the diagonal matrix of ascending eigenvalues $\lambda_1 \leq \cdots \leq \lambda_n$, and the $i$-th column of $\boldsymbol{U}$ is the eigenvector corresponding to $\lambda_i$. It is easy to see that spectral embedding is permutation equivariant: for any node permutation $\boldsymbol{P}$ of the graph, $\boldsymbol{P}\boldsymbol{U}$ is the spectral embedding of the new Laplacian since $\boldsymbol{P}\boldsymbol{L}\boldsymbol{P}^{\top} = (\boldsymbol{P}\boldsymbol{U})\boldsymbol{\Lambda}(\boldsymbol{P}\boldsymbol{U})^{\top}$. Therefore, a permutation-invariant GNN (*e.g.*, DeepSets [63], GIN [57], Graph Transformer [16]) using the SE-augmented input features $\tilde{\boldsymbol{X}} = [\boldsymbol{X}, \boldsymbol{U}]$ remains permutation invariant.

**Reweighted Spectral Embedding (RSE).** Previous works have shown that using spectral embedding improves the expressive power of MP-GNNs [6]. Nonetheless, we find that SE alone is *insufficient* to approximate an arbitrary graph function, as it does not contain all information about the graph, in particular, the eigenvalues. Consider two non-isomorphic graphs whose Laplacian matrices have identical eigenvectors but different eigenvalues. As their SE is the same, a network only using SE cannot tell them apart. To resolve this issue, we propose **reweighted spectral embedding (RSE)** that additionally reweights each eigenvector $\boldsymbol{u}_i$ with the square-root of its corresponding eigenvalue $\lambda_i$, *i.e.*, $\boldsymbol{U}_{\mathrm{RSE}} = \boldsymbol{U}\boldsymbol{\Lambda}^{\frac{1}{2}}$. With the reweighting technique, RSE incorporates eigenvalue information without need extra dimensions.

**Universality of RSE.** In the following theorem, we prove that with RSE, *any* universal network on sets (*e.g.*, DeepSets [63] and Transformer [61]) is universal on graphs while preserving permutation invariance. All proofs are deferred to Appendix K.

**Theorem 1.** *Let $\Omega \subset \mathbb{R}^{n \times d} \times \mathbb{R}^{n \times n}$ be a compact set of graphs, $[\boldsymbol{X}, \hat{\boldsymbol{A}}] \in \Omega$. Let* NN *be a universal neural network on sets. Given any continuous invariant graph function $f$ defined over $\Omega$ and arbitrary $\varepsilon > 0$, there exist a set of* NN *parameters such that for all graphs $[\boldsymbol{X}, \hat{\boldsymbol{A}}] \in \Omega$,*

$$|\mathrm{NN}([\boldsymbol{X}, \boldsymbol{U}_{\mathrm{RSE}}]) - f([\boldsymbol{X}, \hat{\boldsymbol{A}}])| < \varepsilon.$$

As far as we know, this theorem is the first to show that, with the help of graph embeddings like RSE, even an MLP network like DeepSets [63] (composed of a node-wise MLP, a global pooling layer, and a graph-level MLP) can achieve universal expressive power to distinguish any pair of non-isomorphic graphs. Notably, it does not violate the NP-hardness of graph isomorphism testing, since training NN itself is known to be NP-hard [10]. Actually, it is not always necessary to use all spectra of the graph. Existing studies find that high-frequency components are often unhelpful, or even harmful for representation learning [5, 19]. Thus, in practice, we only use the first $k$ low-frequency components of RSE. An upper bound on the approximation error of truncated RSE can be found in Appendix K.10.

**Ambiguities of eigenvectors.** Although RSE enables universal GNNs, there exist two types of ambiguity in eigenvectors that hinder their applications. The first one, known as **sign ambiguity**, arises when an eigenvector $\boldsymbol{u}_{\lambda_i}$ corresponding to eigenvalue $\lambda_i$ is equally valid with its sign flipped, *i.e.*, $-\boldsymbol{u}_{\lambda_i}$ is also an eigenvector corresponding to $\lambda_i$. The second one, termed **basis ambiguity**, occurs when eigenvalues with multiplicity degree $d_i > 1$ can have any other orthogonal basis in the subspace spanned by the corresponding eigenvectors as valid eigenvectors. To be specific, for multiple eigenvalues with multiplicity degree $d_i > 1$, the corresponding eigenvectors $\boldsymbol{U}_{\lambda_i} \in \mathbb{R}^{n \times d_i}$ form an orthonormal basis of a subspace. Then any orthonormal matrix $\boldsymbol{Q} \in \mathbb{R}^{d_i \times d_i}$ can be applied to $\boldsymbol{U}_{\lambda_i}$ to generate a new set of valid eigenvectors for $\lambda_i$. Because of these ambiguities, we can get distinct GNN outputs for the same graph, resulting in unstable and suboptimal performance [17, 25, 18, 29]. In Appendix A, we elaborate the challenges posed by these ambiguities.

Table 2: The ratio of uncanonizable eigenvectors *w.r.t.* each invariance property with our MAP algorithm on three real-world datasets: ZINC, MOLTOX21, and MOLPCBA.

| Invariance | ZINC | MOLTOX21 | MOLPCBA |
|---|---|---|---|
| Sign | 2.46 % | 3.04 % | 2.24 % |
| Basis | 1.59 % | 3.31 % | 7.37 % |
| Total | 4.05 % | 6.35 % | 9.61 % |

## 3 Laplacian Canonization for Sign and Basis Invariance

Rather than incorporating additional modules to learn new sign and basis invariants [25, 29], we explore a straightforward, learning-free approach named Laplacian Canonization (LC). The general idea of LC is to determine a unique direction for each eigenvector $u$ among all its sign and basis equivalents. In this way, the ambiguities can be directly addressed in the pre-processing stage. For example, for two sign ambiguous vectors $u$ and $-u$, we aim to find an algorithm that determines a unique direction among them to obtain sign invariance. Despite some naïve canonization rules (discussed in Appendix G.1), there still lacks a systematical understanding of the following key questions of LC:

1. What kind of canonization algorithm should we look for? (Section 3.1)
2. What kind of eigenvectors are canonizable or non-canonizable? (Section 3.2)
3. Is there an efficient canonization algorithm for all canonizable features? (Section 3.3)

In this section, we answer the three problems by establishing the first formal theory of Laplacian canonization and characterizing the canonizability for sign and basis invariance; based on these analyses, we also propose an efficient algorithm named MAP for LC that is guaranteed to canonize all sign canonizable features. To get a glimpse of our final results, Table 2 shows that MAP can resolve both sign and basis ambiguities for more than 90% of all eigenvectors on real-world data.

### 3.1 Definition of Canonization

To begin with, we first find out what properties a desirable canonization algorithm should satisfy. The ultimate goal of canonization is to eliminate *ambiguities* by selecting a *canonical form* among these ambiguous outputs. Generally speaking, we can characterize ambiguity as a multivalued function $f: \mathcal{X} \to \mathcal{Y}$, *i.e.*, there could be multiple outputs $y_1, \ldots, y_n$ corresponding to the same input $x$. In our sign/basis ambiguity case, $f$ refers to a mapping from a graph $x \in \mathcal{X}$ to the eigenvectors of a certain eigenvalue via the ambiguous eigendecomposition. The ambiguous eigenvectors are not independent; they are related by a sign/basis transformation. In general, we can assume that all possible outputs $y_1, \ldots, y_n$ of any $x$ belong to the same equivalence class induced by some group action $g \in G$, where $G$ acts on $\mathcal{Y}$. That is, if $f(x) = y_1 = y_2$, then there exists $g \in G$ such that $y_1 = gy_2$. Moreover, eigenvectors of graphs obey a fundamental symmetry: permutation equivariance. In general, we assume that $f$ is equivariant to a group $H$ acting on $\mathcal{X}$ and $\mathcal{Y}$. That is, for any $h \in H$, if $y_1, \ldots, y_n$ are all possible outputs of $x$, then $hy_1, \ldots, hy_n$ are all possible outputs of $hx$.

Specifically, for the goal of canonizing ambiguous eigenvectors, *i.e.*, Laplacian canonization, we are interested in algorithms invariant to sign/basis transformations in the corresponding orthogonal group $G$ ($O(1)$ for sign invariance and $O(d)$ for basis invariance). Meanwhile, to preserve the symmetry of graph data, this algorithm should also maintain the permutation equivariant property of eigenvectors (Section 2) *w.r.t.* the permutation group $H$. Third, it also should still be discriminative enough to produce different canonical forms for different graphs (like the original spectral embedding). Combining these desiderata, we have a formal definition of canonization.

**Definition 1.** *A mapping $\mathcal{A}: \mathcal{Y} \to \mathcal{Y}$ is called a ($f, G, H$)-**canonization** when it satisfies:*

- *$\mathcal{A}$ is G-**invariant**: $\forall y \in \mathcal{Y}, g \in G, \mathcal{A}(y) = \mathcal{A}(gy)$;*

- *$\mathcal{A}$ is H-**equivariant**: $\forall x \in \mathcal{X}, h \in H, \mathcal{A}\big(f(hx)\big) = h\mathcal{A}\big(f(x)\big)$;*

- *$\mathcal{A}$ is **universal**: $\forall x \in \mathcal{X}, h \in H, x \neq hx \Rightarrow \mathcal{A}\big(f(x)\big) \neq \mathcal{A}\big(f(hx)\big)$.*

*Accordingly, for any $x \in \mathcal{X}$, if there exists a canonization $\mathcal{A}$ for $x$, we say $x$ is **canonizable**.*

## 3.2 Theoretical Properties of Canonization

Following Definition 1, we are further interested in the question that whether any eigenvector $\boldsymbol{u}$ is canonizable, *i.e.,,* there exists a canonization that can determine its unique direction. Unfortunately, the answer is NO. For example, the vector $\boldsymbol{u} = (1, -1)$ cannot be canonized by any canonization algorithm, since a permutation of $\boldsymbol{u}$ gives $(-1, 1)$ that equals to $-\boldsymbol{u}$[3]. But as long as there is only a small number of uncanonizable eigenvectors like $\boldsymbol{u}$, a canonization algorithm can still resolve the ambiguity issue to a large extent.

Therefore, we are interested in the fundamental question of which eigenvectors are canonizable. The following theorem establishes a general necessary and sufficient condition of the canonizability for general groups $G, H$, which may be of independent interest.

**Theorem 2.** *An input $x \in \mathcal{X}$ is canonizable on the embedding function $f$ iff there does not exist $h \in H$ and $g \in G$ such that $x \neq hx$ and $f(hx) = gf(x)$.*

This theorem states that for inputs from the same equivalence class in $\mathcal{X}$ (induced by $H$), as long as they are not mapped to the same equivalence class in $\mathcal{Y}$ (induced by $G$), these inputs are canonizable. In particular, by applying Theorem 2 to the specific group $G$ induced by sign/basis invariance, we can derive some simple rules to determine whether there exists a canonizable rule for given eigenvector(s).

**Corollary 1** (Sign canonizability). *A vector $\boldsymbol{u} \in \mathbb{R}^n$ is canonizable under sign ambiguity iff there does not exist a permutation matrix $\boldsymbol{P} \in \mathbb{R}^{n \times n}$ such that $\boldsymbol{u} = -\boldsymbol{P}\boldsymbol{u}$.*

**Corollary 2** (Basis canonizability). *The base eigenvectors $\boldsymbol{U} \in \mathbb{R}^{n \times d}$ of the eigenspace $V$ are canonizable under basis ambiguity iff there does not exist a permutation matrix $\boldsymbol{P} \in \mathbb{R}^{n \times n}$ such that $\boldsymbol{U} \neq \boldsymbol{P}\boldsymbol{U}$ and $\operatorname{span}(\boldsymbol{P}\boldsymbol{U}) = \operatorname{span}(\boldsymbol{U}) = V$.*

**Remark.** From a probabilistic view, almost all eigenvectors are canonizable. Let $\boldsymbol{U} \in \mathbb{R}^{n \times d}$ be basis vectors sampled from a continuous distribution in $\mathbb{R}^n$, then $\Pr\{\boldsymbol{U} \text{ is canonizable}\} = 1$.

In the next subsection, we will further design an efficient algorithm to canonize all sign and basis canonizable eigenvectors in the pre-processing stage, so the network does not have to bear the ambiguities of these eigenvectors.

## 3.3 MAP: A Practical Algorithm for Laplacian Canonization

Built upon theoretical properties in Section 3.2, we aim to design a *general, powerful, and efficient* canonization to resolve both sign and basis ambiguities for as many eigenvectors as possible. Here, we choose to adopt axis projection as the basic operator in our canonization algorithm named Maximal Axis Projection (MAP). The key observation is that the standard basis vectors (*i.e.*, the axis) of the Euclidean space $\boldsymbol{e}_i \in \mathbb{R}^n$ are permutation equivariant, and in the meantime, the eigenspace spanned by the eigenvectors $V = \operatorname{span}(\boldsymbol{U})$ are also permutation equivariant. This means that when projecting the axis to the eigenspace, the obtained angles are also permutation equivariant, based on which we could apply permutation invariant functions (such as $\max$) to obtain permutation invariant statistics that can be used for canonization. Meanwhile, due to the generality of projection, it can be applied to both sign ambiguity (for a single eigenvector) and basis ambiguity (for the eigenspace with multiple eigenvectors). We provide illustrative examples to help understand this algorithm in Appendix F.

**Preparation step: Axis projection.** Consider unit eigenvector(s) $\boldsymbol{U} \in \mathbb{R}^{n \times d}$ corresponding to an eigenvalue $\lambda$ with geometric multiplicity $d \geq 1$. These eigenvectors span a $d$-dimensional eigenspace $V = \operatorname{span}(\boldsymbol{U})$ with the projection matrix $\boldsymbol{P} = \boldsymbol{U}\boldsymbol{U}^\top$. To start with, we calculate the projected angle between $V$ and each standard basis (*i.e.*, the axis) of the Euclidean space $\boldsymbol{e}_i$ (a one-hot vector whose $i$-th element is 1), *i.e.*, $\alpha_i = |\boldsymbol{P}\boldsymbol{e}_i|, i = 1, \ldots, n$. Assume that there are $k$ distinct values in $\{\alpha_i, i = 1, \ldots, n\}$, according to which we can divide all basis vectors $\{\boldsymbol{e}_i\}$ into $k$ disjoint groups $\mathcal{B}_i$ (arranged in descending order of the distinct angles). Each $\mathcal{B}_i$ represents an equivalent class of axes that $\boldsymbol{u}_i$ has the same projection on. Then, we define a summary vector $\boldsymbol{x}_i$ for the axes in each group $\mathcal{B}_i$ as their total sum $\boldsymbol{x}_i = \sum_{\boldsymbol{e}_j \in \mathcal{B}_i} \boldsymbol{e}_j + c$, where $c$ is a tunable constant. Next, we introduce how to adopt these summary vectors to canonize the eigenvectors for sign and basis invariance, respectively.

---

[3]In this case, sign invariance conflicts with permutation equivariance: making $\boldsymbol{u}$ and $-\boldsymbol{u}$ output the same vector violates permutation equivariance, and vice versa.

### 3.3.1 Sign Canonization with MAP

**Step 1. Find non-orthogonal axis.** For sign canonization, we calculate the angles between the eigenvalue $\boldsymbol{u}$ and each summary vector $\boldsymbol{x}_i$ one by one, and terminate the procedure as long as we find a summary vector $\boldsymbol{x}_h$ with non-zero angle $\alpha_h = \boldsymbol{u}^\top \boldsymbol{x}_h \neq 0$, and return `<none>` otherwise.

$$\boldsymbol{x}_h = \begin{cases} \min_i \Phi, & \text{if } \Phi \neq \varnothing, \\ \text{<none>}, & \text{if } \Phi = \varnothing, \end{cases} \text{ where } \Phi = \{i \mid \boldsymbol{u}^\top \boldsymbol{x}_i \neq 0\}. \tag{1}$$

**Assumption 1.** *There exists a summary vector $\boldsymbol{x}_h$ that is non-orthogonal to $\boldsymbol{u}$, i.e., $\boldsymbol{u}^\top \boldsymbol{x}_h \neq 0$,*

**Step 2. Sign canonization.** As long as Assumption 1 holds, we can utilize $\boldsymbol{x}_h$ to determine the canonical direction $\boldsymbol{u}^*$ by requiring its projected angle to be positive, *i.e.*,

$$\boldsymbol{u}^* = \begin{cases} \boldsymbol{u}, & \text{if } \boldsymbol{u}^\top \boldsymbol{x}_h > 0, \\ -\boldsymbol{u}, & \text{if } \boldsymbol{u}^\top \boldsymbol{x}_h < 0. \end{cases} \tag{2}$$

We call this algorithm MAP-sign, and summarize it in Algorithm 1 in Appendix C.1. The following theorem proves that it yields a valid canonization for sign invariance under Assumption 1.

**Theorem 3.** *Under Assumption 1, our MAP-sign algorithm gives a sign-invariant, permutation-equivariant and universal canonization of $\boldsymbol{u}$.*

One would wonder how restrictive the non-orthogonality condition (Assumption 1) is for sign canonization. In the following theorem, we establish a strong result showing that sign canonizability is equivalent to non-orthogonality. In other words, *any sign canonizable eigenvectors can be canonized by MAP-sign*. Due to this equivalence, MAP-sign can also serve as a complete algorithm to determine sign canonizability: a vector $\boldsymbol{u}$ is sign canonizable iff $\boldsymbol{x}_h$ is not `<none>` in equation 1.

**Theorem 4.** *A vector $\boldsymbol{u} \in \mathbb{R}^n$ is sign canonizable iff there exists a summary vector $\boldsymbol{x}_h$ s.t. $\boldsymbol{u}^\top \boldsymbol{x}_h \neq 0$.*

**Remark.** Theorem 3 is proved in Appendix K.5 and experimentally verified in Appendix C.2. We also show that MAP is not the only complete canonization algorithm for sign sinvariance by proposing another polynomial-based algorithm in Appendix K.9. We observe that most eigenvectors in real-world datasets are sign canonizable. For instance, on the ogbg-molhiv dataset, the ratio of non-canonizable eigenvectors is only 2.8%. A thorough discussion on the feasibilty of Assumption 1 is in Appendix G.1. For the left non-canonizable eigenvectors, we could apply previous methods (like RandSign, SAN, and SignNet) to further eliminate their ambiguity, which could save more compute since there are only a few non-canonizable eigenvectors. In practice, we also obtain good performance by simply ignoring these non-canonizable features.

### 3.3.2 Basis Canonization with MAP

We further extend MAP-sign to solve the more challenging basis ambiguity with multiple eigenvectors, named MAP-basis. Now, MAP relies on two conditions to produce canonical eigenvectors. The first one requires there are enough distinctive summary vectors to determine $d$ canonical eigenvectors.

**Assumption 2.** *The number of distinctive angles $k$ (i.e., the number of summary vectors $\{\boldsymbol{x}_i\}$) is larger or equal to the multiplicity $d$, i.e., $k \geq d$.*

Under this condition, we can determine each $\boldsymbol{u}_i$ iteratively with the corresponding summary vector $\boldsymbol{x}_i$. At the $i$-th step where $\boldsymbol{u}_1, \ldots, \boldsymbol{u}_{i-1}$ have already been determined, we choose $\boldsymbol{u}_i$ to be the vector that is closest to the summary vector $\boldsymbol{x}_i$ in the orthogonal complement space of $\langle \boldsymbol{u}_1, \ldots, \boldsymbol{u}_{i-1} \rangle$ in $V$:

$$\boldsymbol{u}_i = \arg\max_{\boldsymbol{u}} \boldsymbol{u}^\top \boldsymbol{x}_i,$$
$$s.t.\ \boldsymbol{u} \in \langle \boldsymbol{u}_1, \ldots, \boldsymbol{u}_{i-1} \rangle^\perp, |\boldsymbol{u}| = 1. \tag{3}$$

With a compact feasibility region, the maximum is attainable, and we can further show that the solution $\boldsymbol{u}_i$ is unique (*c.f.*, the proof of Theorem 5). Equation 3 can be directly solved using the projection matrix of $\langle \boldsymbol{u}_1, \ldots, \boldsymbol{u}_{i-1} \rangle^\perp$ (see details in Algorithm 3 in Appendix C.1). Repeating this process gives us a canonical basis of $V$. We also require non-orthogonality condition at each step to obtain a valid eigenvector.

**Assumption 3.** *For any $1 \leq i \leq d$, $\boldsymbol{x}_i$ is not perpendicular to $\langle \boldsymbol{u}_1, \ldots, \boldsymbol{u}_{i-1} \rangle^{\perp}$.*

We summarize MAP-basis in Algorithm 2 in Appendix C.1. In Theorem 5, we prove under Assumption 2 and Assumption 3, MAP-basis gives a basis-invariant, permutation-equivariant and universal canonization of $\boldsymbol{U}$. Though, in this scenario, MAP-basis cannot canonize all basis canonizable features, and whether such an algorithm exists remains an open problem for future research.

**Theorem 5.** *Under Assumption 2 and Assumption 3, our MAP-basis algorithm gives a basis-invariant, permutation-equivariant and universal canonization of $\boldsymbol{U}$.*

**Remark.** Assumption 2 and Assumption 3 exclude some symmetries of the eigenspace, which we discuss in details in Appendix G.2. For random orthonormal matrices, the possibility that either assumption is violated is equal to 0, which we verify with random simulation in Appendix C.3. For real-world datasets, these assumptions are not restrictive either. For instance, on the large ogbg-molpcba dataset, the eigenvalues violating Assumption 2 or Assumption 3 make up 0.87% of all eigenvalues. More statistics and discussions are provided in Appendix G.2.

### 3.3.3 Summary

We provide the complete pseudo-code of MAP to address both sign and basis invariance in Appendix C.1, along with a detailed time complexity analysis. Overall, the extra complexity introduced by MAP is $\mathcal{O}(n^2 \log n)$, which is better than eigendecomposition itself with $\mathcal{O}(n^3)$. It only needs to be computed once per dataset and can be easily incorporated with various GNN architectures.

Combining Theorem 1, Theorem 3 and Theorem 5 gives us the universality of first-order GNNs [33] such that it respects all graph symmetries under certain assumptions (Assumptions 1, 2 & 3). We show that the probability of violating these assumptions asymptotically converges to zero on random graphs (see Appendix H). We also count the ratio of violation in real-world datasets in Table 2, and observe that the ratio of uncanonizable eigenvectors is less than 10% on all datasets. Thus, MAP greatly eases the harm caused by sign and basis ambiguities.

So far only GNNs using higher-order tensors have achieved universality while respecting *all* graph symmetries [33, 24], but they are typically computationally prohibitive in practice. It is still an open problem whether it is also possible for first-order GNNs. The proposed Laplacian Canonization presents a new approach in this direction trying to alleviate the harm of sign and basis ambiguities in a minimalist approach. By establishing universality-invariance results for first-order GNNs under certain assumptions, LC could hopefully could bring some insights to the GNN community.

## 4 Experiments

We evaluate the proposed MAP positional encoding on sparse MP-GNNs and Transformer GNNs using PyTorch [40] and DGL [53]. For sparse MP-GNNs we consider GatedGCN [11] and PNA [15], and for Transformer GNNs we consider SAN [25] and GraphiT [34]. We conduct experiments on three standard molecular benchmarks—ZINC [22], OGBG-MOLTOX21 and OGBG-MOLPCBA [20]. ZINC and MOLTOX21 are of medium scale with 12K and 7.8K graphs respectively, whereas MOLPCBA is of large scale with 437.9K graphs. Details about these datasets are described in Appendix L. We follow the same protocol as in Lim et al. [29], where we replace their SignNet with a normal GNN and the Laplacian eigenvectors with our proposed MAP. We fairly compare several models on a fixed number of 500K model parameters on ZINC and relax the model sizes to larger parameters for evaluation on the two OGB datasets, as being practised on their leaderboards [20]. We also compare our results with GNNs using LapPE and random sign (RS) [17], GNNs using SignNet [29], and GNNs with no PE. For a fair comparison, all models use a limited number $k$ of eigenvectors for positional encodings. The results of all our experiments are presented in Table 3, 4 & 5. Further implementation details are included in Appendix M.

### 4.1 Performance on Benchmark Datasets

As shown in Table 3, 4 & 5, using MAP improves the performance of all GNNs on all datasets, demonstrating that removing ambiguities of the eigenvectors is beneficial for graph-level tasks. First, by comparing models with LapPE and RS with models with no PE, we observe that the use of LapPE significantly improves the performance on ZINC, showing the benefits of incorporating expressive

PEs with GNNs, especially with MP-GNNs whose expressive power is limited by the 1-WL test. However, on MOLTOX21 and MOLPCBA, using LapPE has no significant effects. This is because unlike ZINC, OGB-MOL* datasets contain additional structural features that are informative, *e.g.*, if an atom is in ring, among others [18]. Thus the performance gain by providing more positional information is less obvious. Second, MAP outperms LapPE with RS by a large margin especially on ZINC. Although RS also alleviates sign ambiguity by randomly flipping signs during training, MAP removes such ambiguity *before* training, enabling the network to focus on the real meaningful features and achieves a better performance. Third, we also observe that MAP and SignNet achieve comparable performance. This is because both methods aim at the same goal—eliminating ambiguity. However, SignNet does so in the training stage while MAP does so in the pre-processing stage, thus the latter is more computationally efficient. Lastly, we would also like to highlight that as a kind of positional encoding, MAP can be easily incorporated with any GNN architecture by passing the `pre_transform` function to the dataset class with a single line of code.

Table 3: Results on ZINC. All scores are averaged over 4 runs with 4 different seeds.

| Model | PE | $k$ | #Param | MAE $\downarrow$ |
|---|---|---|---|---|
| GatedGCN | None | 0 | 504K | 0.251 ± 0.009 |
| GatedGCN | LapPE + RS | 8 | 505K | 0.202 ± 0.006 |
| GatedGCN | SignNet ($\phi(v)$ only) | 8 | 495K | 0.148 ± 0.007 |
| GatedGCN | SignNet | 8 | 495K | 0.121 ± 0.005 |
| GatedGCN | MAP | 8 | 486K | **0.120 ± 0.002** |
| PNA | None | 0 | 369K | 0.141 ± 0.004 |
| PNA | LapPE + RS | 8 | 474K | 0.132 ± 0.010 |
| PNA | SignNet | 8 | 476K | 0.105 ± 0.007 |
| PNA | MAP | 8 | 462K | **0.101 ± 0.005** |
| SAN | None | 0 | 501K | 0.181 ± 0.004 |
| SAN | MAP | 16 | 230K | **0.170 ± 0.012** |
| GraphiT | None | 0 | 501K | 0.181 ± 0.006 |
| GraphiT | MAP | 16 | 329K | **0.160 ± 0.006** |

Table 4: Results on MOLTOX21. All scores are averaged over 4 runs with 4 different seeds.

| Model | PE | $k$ | #Param | ROCAUC $\uparrow$ |
|---|---|---|---|---|
| GatedGCN | None | 0 | 1004K | 0.772 ± 0.006 |
| GatedGCN | LapPE + RS | 3 | 1004K | 0.774 ± 0.007 |
| GatedGCN | MAP | 3 | 1505K | **0.784 ± 0.005** |
| PNA | None | 0 | 5245K | 0.755 ± 0.008 |
| PNA | MAP | 16 | 1951K | **0.761 ± 0.002** |
| SAN | None | 0 | 958K | 0.744 ± 0.007 |
| SAN | MAP | 12 | 1152K | **0.766 ± 0.007** |
| GraphiT | None | 0 | 958K | 0.743 ± 0.003 |
| GraphiT | MAP | 16 | 590K | **0.769 ± 0.011** |

Table 5: Results on MOLPCBA. All scores are averaged over 4 runs with 4 different seeds.

| Model | PE | $k$ | #Param | AP $\uparrow$ |
|---|---|---|---|---|
| GatedGCN | None | 0 | 1008K | 0.262 ± 0.001 |
| GatedGCN | LapPE + RS | 3 | 1009K | 0.266 ± 0.002 |
| GatedGCN | MAP | 3 | 2658K | **0.268 ± 0.002** |
| PNA | None | 0 | 6551K | 0.279 ± 0.003 |
| PNA | MAP | 16 | 4612K | **0.281 ± 0.003** |

## 4.2 Empirical Understandings

**Computation time.** We demonstrate the efficiency of MAP by measuring the pre-processing time and training time on the large OGBG-MOLPCBA dataset, and compare them with SignNet. For a fair comparison, we use identical model size, hyperparameters and random seed and conduct experiments on the same NVIDIA 3090 GPU. The results are shown in Table 6. We observe that model with MAP train 41% faster than its SignNet counterpart, saving 44 hours of training time. Since SignNet takes the form $\rho\big(\phi(\boldsymbol{u}) + \phi(-\boldsymbol{u})\big)$ while we use models taking the form $\rho\big(\phi(\boldsymbol{u})\big)$, models with MAP would always train faster than those with SignNet under the same hyperparameters. We also observe that the pre-processing time is negligible compared with training time ($< 3\%$), since pre-processing only needs to be done once. This makes MAP overall more efficient while achieving the same goal of tackling ambiguities.

Table 6: Comparison of pre-processing and training time between models with MAP or SignNet as PE, on the MOLPCBA dataset. Experiments are run with the same model size and hyperparameters, the same random seed, on the same NVIDIA 3090 GPU.

| Model | Pre-processing time | Training Time | Total Time |
|---|---|---|---|
| GatedGCN + MAP | 1.70 h | 63.02 h | 64.72 h |
| GatedGCN + SignNet | 0.27 h | 108.51 h | 108.78 h |

**Spectral embedding dimension.** Next, we study the effects of $k$. We train GatedGCN with MAP on ZINC with different number of eigenvectors used in the PE and report the results in Table 7. The hyperparameters are the same across these experiments. It can be observed that the performance drops when $k$ is too small, meaning that SE provides crucial structural information to the model. Using larger $k$ has limited influence on the performance, meaning that the model relies more on low-frequency information of spectral embedding.

Table 7: Effects of $k$, where $k$ is the number of eigenvectors used.

| $k$ | 0 | 4 | 8 | 16 |
|---|---|---|---|---|
| Test MAE | $0.256 \pm 0.012$ | $0.138 \pm 0.004$ | $\mathbf{0.120 \pm 0.002}$ | $0.124 \pm 0.002$ |

**Ablation study.** Finally, we conduct ablation study of MAP by removing each component of MAP and evaluate the performance. The results are shown in Table 8. Removing sign invariance hurts the performance the most, because most eigenvectors are single and thus sign ambiguity has the most influence on the model's performance. Removing basis invariance also has negative effects since a small portion of eigenvalues are multiple. Removing eigenvalues has moderate negative effects, showing that the incorporation of eigenvalue information is beneficial for the network.

Table 8: Effects of the three components of MAP (GatedGCN on ZINC).

| PE | full MAP | without can. sign | without can. basis | without eigenvalues |
|---|---|---|---|---|
| Test MAE | $\mathbf{0.120 \pm 0.002}$ | $0.131 \pm 0.003$ | $0.122 \pm 0.003$ | $0.125 \pm 0.001$ |

## 5  Conclusion

In this paper, we explored a new approach called Laplacian Canonization for addressing sign and basis ambiguities of Laplacian eigenvectors while also preserving permutation invariance and universal expressive power. We developed a novel theoretical framework to characterize canonization and the canonizability of eigenvectors. Then we proposed a practical canonization algorithm called Maximal Axis Projection (MAP). Theoretically, it is sign/basis-invariant, permutation-equivariant and universal. Empirically, it canonizes most eigenvectors on synthetic and real-world data while showing promising performance on various datasets and GNN architectures.

## Acknowledgements

Yisen Wang was supported by National Key R&D Program of China (2022ZD0160304), National Natural Science Foundation of China (62006153, 62376010, 92370129), Open Research Projects of Zhejiang Lab (No. 2022RC0AB05), and Beijing Nova Program (20230484344).

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

# Appendix

## Table of Contents

# A  Why sign and basis invariance matters

It is perfectly understandable that one may not see the significance and benefits of sign and basis invariance at once. For instance, below is a quote from one of the reviews of Lim et al. [29] when they first submitted their paper to NeurIPS 2022:

> It is not clear "why" to preserve the two symmetries this paper proposes to preserve including sign invariance and basis invariance. In graph & molecule setting, two important symmetries are permutation (*i.e.* if we permute the nodes/atoms, the output stays the same) and rotation (*i.e.* if we rotate the 3D coordinates associating with each atom, the prediction for a physical property stays the same). I don't see a strong evidence that suggests preserving sign & basis invariances on top of permutation & rotation invariances brings any benefits.

Indeed, it is not easy to see why issues related with something matters if they don't even use it. One may overlook the importance of sign and basis invariance for two reasons: (1) people not familiar with Laplacian PE may not even realize sign/basis ambiguity is a thing; (2) people that does have some knowledge about Laplacian PE may not realize sign and basis ambiguities are markedly hindering its performance. Nonetheless, Laplacian PE does have advantages over other commonly used PEs that it is worth the efforts addressing their drawbacks. For these reasons we feel it is of great significance that we explain the motivation behind preserving sign and basis invariances in more details.

**Why use spectral embedding: trade-off between universality and ambiguities.** Sign and basis ambiguities arise only in spectral embeddings, so the first important question is why to use them. There are many kinds of positional encodings in the literature, such as Random Walk PE (RWPE), Position-aware Encoding, Relational Pooling (RP), random features, *etc*. Without addressing sign and basis ambiguities, some works even reported superior performance of RWPE over spectral embedding [18, 42]. However, these other PEs all have drawbacks either in their expressive power or permutation equivariance. RP and random features are universal, but they do not guarantee permutation equivariance, which is an important inductive bias in graph-structured data. RWPE and positional-aware encoding are permutation equivariant, but their expressive power is limited. Thus one may wonder whether there exists a kind of positional encoding method that achieves both. Spectral Embedding, as it turns out, is both permutation equivariant and universal (when coupled with eigenvalue information, see Sec 2). The good thing about SE is that through eigendecomposition, it transforms a "harder" permutation equivariance formulated by

$$f(\boldsymbol{P}^\top \hat{\boldsymbol{A}} \boldsymbol{P}) = \boldsymbol{P}^\top f(\hat{\boldsymbol{A}}) \boldsymbol{P}, \tag{4}$$

into a "easier" permutation equivariance formulated by

$$f(\boldsymbol{P}\boldsymbol{U}) = \boldsymbol{P}f(\boldsymbol{U}), \tag{5}$$

where $\boldsymbol{P}$ is an arbitrary permutation matrix, $\hat{\boldsymbol{A}}$ is the normalized adjacency matrix of the graph and $\boldsymbol{U}$ are its eigenvectors. Universality and equivariant results for equation 4 are only known for high-order tensor networks [32], but similar results for equation 5 have long been known to be achievable even by first-order networks such as DeepSets [63]. Thus SE enables efficient and universal graph neural networks that respects permutation equivariance. However, SE also brings new problems, sign and basis ambiguities. It is not fair to say models with SE are universal without achieving sign and basis invariance, since we expect graph neural networks to respect all symmetries among isomorphic graphs.

**Sign and basis invariances are well-established problems in the literature.** Sign and basis ambiguities have been recognized by numerous works in the literature to be challenging and important issues when incorporating SE as positional encoding. Below we list some existing discussions in these papers.

Quote from Dwivedi et al. [17]:

> We propose an alternative to reduce the sampling space, and therefore the amount of ambiguities to be resolved by the network. Laplacian eigenvectors are hybrid positional and structural encodings, as they are invariant by node reparametrization. However, they are also limited by natural symmetries such as the arbitrary sign of eigenvectors (after being normalized to have unit length). The

number of possible sign flips is $2^k$, where $k$ is the number of eigenvectors. In practice we choose $k \ll n$, and therefore $2^k$ is much smaller than $n!$ (the number of possible ordering of the nodes). During the training, the eigenvectors will be uniformly sampled at random between the $2^k$ possibilities. If we do not seek to learn the invariance *w.r.t.* all possible sign flips of eigenvectors, then we can remove the sign ambiguity of eigenvectors by taking the absolute value. This choice seriously degrades the expressivity power of the positional features.

Quote from Kreuzer et al. [25]:

As noted earlier, there is a sign ambiguity with the eigenvectors. With the sign of $\phi$ being independent of its normalization, we are left with a total of $2^k$ possible combination of signs when choosing $k$ eigenvectors of a graph. Previous work has proposed to do data augmentation by randomly flipping the sign of the eigenvectors [6, 16, 17], and although it can work when $k$ is small, it becomes intractable for large $k$.

Quote from Beaini et al. [6]:

For instance, a pair of eigenvalues can have a multiplicity of 2 meaning that they can be generated by different pairs of orthogonal eigenvectors. For an eigenvalue of multiplicity 1, there are always two unit norm eigenvectors of opposite sign, which poses a problem during the directional aggregation. We can make a choice of sign and later take the absolute value. An alternative is to take a sample of orthonormal basis of the eigenspace and use each choice to augment the training.

Quote from Mialon et al. [34]:

Note that eigenvectors of the Laplacian computed on different graphs could not be compared to each other in principle, and are also only defined up to a $\pm 1$ factor. While this raises a conceptual issue for using them in an absolute positional encoding scheme, it is shown in Dwivedi and Bresson [16]—and confirmed in our experiments—that the issue is mitigated by the Fourier interpretation, and that the coordinates used in LapPE are effective in practice for discriminating between nodes in the same way as the position encoding proposed in Vaswani et al. [51] for sequences. Yet, because the eigenvectors are defined up to a $\pm 1$ factor, the sign of the encodings needs to be randomly flipped during the training of the network.

Quote from Dwivedi and Bresson [16]:

In particular, Dwivedi et al. [17] make the use of available graph structure to pre-compute Laplacian eigenvectors [7] and use them as node positional information. Since Laplacian PEs are generalization of the PE used in the original transformers [51] to graphs and these better help encode distance-aware information (*i.e.*, nearby nodes have similar positional features and farther nodes have dissimilar positional features), we use Laplacian eigenvectors as PE in Graph Transformer. Although these eigenvectors have multiplicity occuring due to the arbitrary sign of eigenvectors, we randomly flip the sign of the eigenvectors during training, following Dwivedi et al. [17].

Quote from Dwivedi et al. [18]:

Another PE candidate for graphs can be Laplacian Eigenvectors [17, 16] as they form a meaningful local coordinate system, while preserving the global graph structure. However, there exists sign ambiguity in such PE as eigenvectors are defined up to $\pm 1$, leading to $2^k$ number of possible sign values when selecting $k$ eigenvectors which a network needs to learn. Similarly, the eigenvectors may be unstable due to eigenvalue multiplicities.

Quote from Lim et al. [29]:

However, there are nontrivial symmetries that should be accounted for when processing eigenvectors. If $v$ is a unit-norm eigenvector, then so is $-v$, with the same eigenvalue. More generally, if an eigenvalue has higher multiplicity, then there are infinitely many unit-norm eigenvectors that can be chosen. Indeed, a full set of orthonormal eigenvectors is only defined up to a change of basis in each eigenspace. In the case of sign invariance, for any $k$ eigenvectors there are $2^k$ possible choices of sign. Accordingly, prior works randomly flip eigenvector signs during training in order to approximately learn sign invariance [25, 17]. However, learning all $2^k$ invariances is challenging and limits the effectiveness of Laplacian eigenvectors for encoding positional information. Sign invariance is a special case of basis invariance when all eigenvalues are distinct, but higher dimensional basis invariance is even more difficult to deal with, and we show that these higher dimensional eigenspaces are abundant in real datasets.

Quote from Wang et al. [52]:

Srinivasan and Ribeiro [45] states that PE using the eigenvectors of the randomly permuted graph Laplacian matrix keeps permutation equivariant. Dwivedi et al. [17], Kreuzer et al. [25] argue that such eigenvectors are unique up to their signs and thus propose PE that randomly perturbs the signs of those eigenvectors. Unfortunately, these methods may have risks. They cannot provide permutation equivariant GNNs when the matrix has multiple eigenvalues, which thus are dangerous when applying to many practical networks. For example, large social networks, when not connected, have multiple 0 eigenvalues; small molecule networks often have non-trivial automorphism that may give multiple eigenvalues. Even if the eigenvalues are distinct, these methods are unstable. We prove that the sensitivity of node representations to the graph perturbation depends on the inverse of the smallest gap between two consecutive eigenvalues, which could be actually large when two eigenvalues are close.

**Sign and basis invariances make the learning tasks easier.** Permutation invariance is an important inductive bias in graphs in the sense that applying permutations on graphs results in isomorphic graphs and they should have the same properties and produce the same outputs. Empirically constraining the network to be invariant to such permutations benefits the performance, since otherwise the network has to learn these $n!$ ambiguities by itself. Similarly, sign and basis ambiguities result in $2^k$ and infinite possible choices of positional features respectively that a network has to learn, which could degrade its performance greatly. By preserving sign and basis invariance, a network will not need to learn the equivalence between all these possible choices and the learning task is significantly easier.

**There are less ambiguity in signs than in permutations.** For a given graph, there are $n!$ possible ordering of the nodes that all represent the same graph, resulting in $n!$ number of ambiguities. Considering that the vast majority of eigenvectors on real-world data have distinct eigenvalue (Table 9), when selecting $k$ eigenvectors as positional encoding to present structural information, the number of ambiguities is $2^k$, which is much smaller than $n!$. This transformation from permutation ambiguity to sign ambiguity greatly reduces the amount of random noise in the input data and makes the model much more stable.

**When sign invariance meets permutation equivariance.** Sign ambiguity is not an issue when we do not take permutation symmetry into account. For each eigenvector pair $\pm u$, we can choose the one whose first non-zero entry is positive. This simple *canonization* solves the sign ambiguity problem with ease. The problem of sign ambiguity is only *non-trivial* when combined with permutation equivariance, that is, we require the canonization process to be both sign-invariant and permutation-equivariant. Since the eigenvectors can now be permuted arbitrarily, there is no "first" non-zero entry and the simple canonization above fails.

Moreover, such canonization only makes sense when we require it to be *universal*. For instance, mapping all eigenvectors to $\boldsymbol{j} = (1, 1, \ldots, 1)$ is also a sign-invariant and permutation-equivariant canonization, but this canonization provides us with no information. Thus one may wonder whether a network can be both sign-invariant, permutation-equivariant and universal at the same time. Unfortunately, as far as we know, no existing work addresses all of them. The universality of SignNet only considers sign invariance, as stated in their theorem:

**Theorem 6** (Universal representation for SignNet). *A continuous function $f\colon (\mathbb{S}^{n-1})^k \to \mathbb{R}^s$ is sign invariant, i.e. $f(s_1v_1, \ldots, s_kv_k) = f(v_1, \ldots, v_k)$ for any $s_i \in \{-1, 1\}$, if and only if there exists a continuous $\phi\colon \mathbb{R}^n \to \mathbb{R}^{2n-2}$ and a continuous $\rho\colon \mathbb{R}^{(2n-2)k} \to \mathbb{R}^s$ such that*

$$f(v_1, \ldots, v_k) = \rho\big([\phi(v_i) + \phi(-v_i)]_{i=1}^k\big).$$

In fact, being sign-invariant, permutation-equivariant and universal at the same time is not even *well-defined*. As mentioned in the main text, for an uncanonizable eigenvector $\boldsymbol{u}$ with $\boldsymbol{u} = -\boldsymbol{Pu}$ for some permutation matrix $\boldsymbol{P}$, let $f$ be an arbitrary continuous sign-invariant, permutation-equivariant and universal function and denote $f(\boldsymbol{u}) = \boldsymbol{y}$. The sign invariance property requires that $f(-\boldsymbol{u}) = \boldsymbol{y}$, while the permutation equivariance property requires that $f(\boldsymbol{Pu}) = \boldsymbol{Py}$. Since $-\boldsymbol{u} = \boldsymbol{Pu}$, we have $\boldsymbol{y} = \boldsymbol{Py}$. However $\boldsymbol{u} \neq \boldsymbol{Pu}$, thus universality is not satisfied, leading to a contradiction. This can be illustrated in Figure 1, where the three properties cannot be met at the same time.

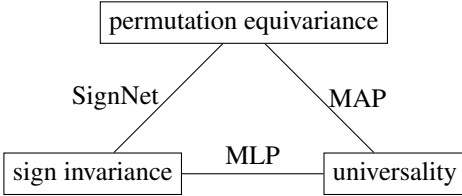

Figure 1: The dilemma where permutation-equivariance, sign-invariance and universal expressive power cannot be achieved at the same time.

**Remark.** The *universality* in Definition 1 is defined on eigenvectors $\boldsymbol{U} \in \mathbb{R}^{n \times n}$. The permutation equivariance property of eigenvectors is defined as $f(\boldsymbol{PU}) = \boldsymbol{P}f(\boldsymbol{U})$, and here universality implies that if two eigenvectors are not equal up to permutation, then $f$ should be able to tell them apart. As mentioned above, such *set-universal* networks (like DeepSets or MLP) cannot be permutation equivariant and sign invariant at the same time. In the context of graphs, we may also care about universality defined on the adjacency matrix $\boldsymbol{A} \in \mathbb{R}^{n \times n}$, in which case the permutation equivariance property is defined as $f(\boldsymbol{PAP}^\top) = \boldsymbol{P}f(\boldsymbol{A})$. Such *graph-universal* networks are always able to tell two non-isomorphic adjacency matrices apart. We can view $f(\boldsymbol{A})$ as a function of $\boldsymbol{U}$ (we omit the eigenvalues for simplicity), and we denote $g(\boldsymbol{U}) = f(\boldsymbol{A})$, then to achieve the graph universality of $f$, $g$ does not have to be set-universal, thus may not suffer from the dilemma above.

Much effort has been devoted to the structure of invariant and equivariant networks in the literature, but few has studied the case when invariance and equivariance combine (in this case you may have to sacrifice some invariance, or some equivariance, or some expressive power). Our work takes a step in this direction, providing some insights to this dilemma and characterize the necessary and sufficient conditions where these three properties conflict with each other.

# B  Related work

**Theoretical Expressivity**   It has been shown that neighborhood-aggregation based GNNs are not more powerful than the WL test [57, 35]. Since then, many attempts have been made to endow graph neural networks with expressive power beyond the 1-WL test, such as injecting random attributes [36, 44, 59, 45, 17], injecting positional encodings [28, 60], and designing higher-order GNNs [35, 32, 33, 31, 14].

**Positional Encodings**   Recently, there are many works that aim to improve the expressive power of GNNs by positional encoding (PE). Murphy et al. [36] assigned each node an identifier depending on its index. Li et al. [28] proposed to use distances between nodes characterized by powers of the random walk matrix. You et al. [59] proposed to use distances of nodes to a sampled anchor set of nodes. Using random node features can also improve the expressiveness of GNNs, even making them universal [44, 1], but it has several defects: (1) loss of permutation invariance, (2) slower convergence, (3) poor generalization on unseen graphs [59, 30]. More detailed discussion can be found in Appendix E.

**Laplacian PE**   Another PE candidate for graphs is Laplacian Eigenvectors [16, 17] which form a meaningful local coordinate system while preserving global structure. However, there exists

ambiguities in sign and basis that the network needs to learn, which harms the performance of GNNs. To address these ambiguities, many approaches have been proposed such as randomly flip the signs [17], use eigenfunctions over the edges [25] or use invariant network architectures [29].

**Sign Invariance**    Several prior works have proposed ways of addressing sign ambiguity. A popular heuristic is Random Sign (RS), which randomly flips the signs of eigenvectors during training to encourage the insensitivity to different signs [17]. However, the network still has to learn these signs, leading to slower convergence and harder curve fitting task. Bro et al. [12] developed a data-dependent method to choose signs for each singular vector of a singular value decomposition. Still, in the worst case the signs chosen will be arbitrary, and they do not handle rotational ambiguities in higher dimensional eigenspaces. Kreuzer et al. [25] proposed to use the relative Laplace embeddings of two nodes. However, their approach suffers from a major computational bottleneck with $\mathcal{O}(n^4)$ complexity. Lim et al. [29] proposed SignNet that passes both $\boldsymbol{u}$ and $-\boldsymbol{u}$ to the same network, adds the outputs together and then passes them to another network. As both outputs of $\boldsymbol{u}$ and $-\boldsymbol{u}$ need to be computed, their approach adds to the training cost. Our approach happens at pre-processing time with $\mathcal{O}(n^3)$ complexity.

**Basis Invariance**    The only works we know addressing basis ambiguity are Lim et al. [29] and Huang et al. [21]. Lim et al. [29] proposed BasisNet that is basis invariant. However, to achieve universality, the components of BasisNet need to use higher order tensors in $\mathbb{R}^{n^k}$ where $k$ can be as large as $\frac{n(n-1)}{2}$ [33], rendering BasisNet impractical. Huang et al. [21] generalized BasisNet and proposed SPE that is both basis-invariant and stable to perturbation to the Laplacian matrix, but it still suffers from exponential complexity as BasisNet. In this paper, we showed that under certain assumptions, the issue of basis ambiguity can be resolved more efficiently in the pre-processing stage.

There is also a literature beyond GNN that also considers spectral embeddings of graphs and going around the sign/basis ambiguity problems. Lai and Zhao [26] proposed to address sign/basis ambiguities using optimal transport theory that involves solving a non-convex optimization problem, thus it could be less efficient than our approach. Tam and Dunson [48] proposed to symmetrize the embedding using a heuristic measure called ELD that is quite similar to the form of SignNet, while our MAP algorithm offers an axis projection approach and establish its theoretical guarantees.

**Canonical Forms**    The theory of canonical forms has been widely studied in mathematics [39] and applied to many fields of machine learning. Sajnani et al. [43] proposed a self-supervised method named ConDor that learns to canonicalize the 3D orientation and position for full and partial 3D point clouds. Agaram et al. [2] presented Canonical Field Network (CaFi-Net), a self-supervised method to canonicalize the 3D pose of instances from an object category represented as neural fields, specifically neural radiance fields (NeRFs). Sun et al. [47] proposed an unsupervised capsule architecture for 3D point clouds. Kaba et al. [23] proposed to decouple the equivariance and prediction components of neural networks by predicting a transformation that can be used to align the input data to some canonical pose. After this alignment the remaining layers do not have to be equivariant anymore. However, as far as we know, no existing work addresses the issue of *ambiguities*, which arises when some function (*e.g.*, eigendecomposition) is required to be both *equivariant* on the input space and *invariant* on the output space. No existing work addresses the *canonizability* of input data, which only arises when the invariance and equivariance property conflicts. Instead, we develop a novel theoretical framework that addresses these issues.

## C    Implementation and verification of MAP

### C.1    Complete pseudo-code of MAP

A simplified pseudo-code for sign canonization with MAP is shown in Algorithm 1. A simplified pseudo-code for basis canonization with MAP is shown in Algorithm 2. ($\boldsymbol{j} = (1, 1, \ldots, 1) \in \mathbb{R}^n$)

The complete pseudo-code of MAP is shown in Algorithm 3. Despite the sophisticated workflow, programmers do not have to know the principles of our algorithm to use it. The entire module can be passed as a `pre_transform` function to the dataset class with a single line of code.

There are three steps in Algorithm 3. The first is to eliminate sign ambiguity with $\mathcal{O}(n^2 \log n)$ time complexity. The second is to eliminate basis ambiguity with $\mathcal{O}(n^2 dm)$ time complexity, where $d$ is the multiplicity of the eigenvalue and $m$ is the number of multiple eigenvalues of $\hat{\boldsymbol{A}}$. The third is

---

**Algorithm 1** Maximal Axis Projection for eliminating sign ambiguity

---

**Require:** Input graph $\mathcal{G} = (\mathbb{V}, \mathbb{E}, \boldsymbol{X})$
**Ensure:** Spectral embedding of $\mathcal{G}$
   Calculate the eigendecomposition $\hat{\boldsymbol{A}} = \boldsymbol{U}\boldsymbol{\Lambda}\boldsymbol{U}^\top$          $\triangleright \mathcal{O}(n^3)$ complexity
   **for** each eigenvector $\boldsymbol{u}$ **do**          $\triangleright \mathcal{O}(n^2 \log n)$ complexity
      $\boldsymbol{x}_i \leftarrow \sum_{\boldsymbol{e}_j \in \mathcal{B}_i} \boldsymbol{e}_j + c\boldsymbol{j},\ i = 1, \ldots, k$
      $\boldsymbol{x}_h \leftarrow$ non-orthogonal summary vector with smallest $h$ (equation 1)
      **if** $\boldsymbol{x}_h$ is not `<none>` **then**
         $\boldsymbol{u} \leftarrow \boldsymbol{u}^*$ (choose the direction with $\boldsymbol{u}^\top \boldsymbol{x}_h > 0$ as in equation 2)
      **else**
         $\boldsymbol{u} \leftarrow \boldsymbol{u}$ (no canonization)
      **end if**
   **end for**
   **return** $\boldsymbol{U}$

---

---

**Algorithm 2** Maximal Axis Projection for eliminating basis ambiguity

---

**Require:** Eigenvalue $\lambda$ with multiplicity $d > 1$
**Ensure:** Spectral embedding corresponding to $\lambda$
   Calculate the eigenvectors $\boldsymbol{U} \in \mathbb{R}^{n \times d}$ of $\lambda$ through eigendecomposition    $\triangleright \mathcal{O}(n^3)$ complexity
   **for** $i = 1, 2, \ldots, d$ **do**          $\triangleright \mathcal{O}(n^2 d)$ complexity
      $\boldsymbol{x}_i \leftarrow \sum_{\boldsymbol{e}_j \in \mathcal{B}_i} \boldsymbol{e}_j + c\boldsymbol{j}$
      Choose $\boldsymbol{u}_i \in \langle \boldsymbol{u}_1, \ldots, \boldsymbol{u}_{i-1} \rangle^\perp, |\boldsymbol{u}_i| = 1$, *s.t.* $f(\boldsymbol{u}_i) = \boldsymbol{u}_i^\top \boldsymbol{x}_i$ is maximized (equation 3)
   **end for**
   **return** $\boldsymbol{U}_0 \coloneqq [\boldsymbol{u}_1, \ldots, \boldsymbol{u}_d]$

---

to incorporate eigenvalue information with $\mathcal{O}(n^2)$ time complexity. The overall time complexity is $\mathcal{O}(n^3)$ with the bottleneck being the eigendecomposition.

The second part of our algorithm (eliminating basis ambiguity) has time complexity $\mathcal{O}(n^2 dm)$. We point out that in real-world datasets $m$ is often quite small. As shown in Table 9, multiple eigenvalues only make up around $5\%$ of all eigenvalues, thus in practice the time complexity of the second part is $\mathcal{O}(n^2 d)$, better than eigendecomposition. (Note it is important to take floating-point errors into account when counting these multiple eigenvalues)

Table 9: The number of multiple eigenvalues in real-world datasets.

| Dataset | ogbg-molesol | ogbg-molfreesolv | ogbg-molhiv | ogbg-mollipo | ogbg-moltox21 | ogbg-moltoxcast | ogbg-molpcba |
|---|---|---|---|---|---|---|---|
| #multiple eigenvalues | 738 | 286 | 52367 | 5391 | 8772 | 10556 | 491247 |
| #all eigenvalues | 13420 | 4654 | 952055 | 104669 | 129730 | 141042 | 10627757 |
| Ratio | 5.50% | 6.15% | 5.50% | 5.15% | 6.76% | 7.48% | 4.62% |

Some parts of Algorithm 3 have time complexity $\mathcal{O}(nd^4)$. We also point out that $d$ is usually small in real datasets. We show the number of eigenvalues and their multiplicities in logarithmic scale in Figure 2.

The details of the datasets are listed in Table 10.

## C.2   Verifying the correctness of Theorem 3

We verify the correctness of Theorem 3 through random simulation. The program is shown in Algorithm 4. Let $\boldsymbol{U}$ be a random orthonormal matrix, $\boldsymbol{P}$ be a random permutation matrix and $\boldsymbol{S}$ be a random sign matrix (diagonal matrix of 1 and $-1$). We pass $\boldsymbol{U}, \boldsymbol{PU}, \boldsymbol{US}, \boldsymbol{PUS}$ to the UNIQUESIGN function (Algorithm 1) and compare the outputs. If our algorithm is correct, $\boldsymbol{U}$ and $\boldsymbol{US}$ should have invariant outputs, while $\boldsymbol{U}, \boldsymbol{PU}$ and $\boldsymbol{PUS}$ should have equivariant outputs.

We conduct 1000 trials. The results are $p\_correct = q\_correct = pq\_correct = total = 1000$, showing that Algorithm 1 is both permutation-equivariant and unique (unambiguous).

**Algorithm 3** Maximal Axis Projection

**Require:** Graph $\mathcal{G} = (\mathbb{V}, \mathbb{E})$, its normalized adjacency matrix $\hat{A}$
**Ensure:** Maximal Axis Projection of $\mathcal{G}$

   Calculate the eigendecomposition $\hat{A} = U\Lambda U^\top$                             $\triangleright \mathcal{O}(n^3)$ complexity
   **for** each single eigenvector $u \in \mathbb{R}^n$ **do**                    $\triangleright \mathcal{O}(n^2 \log n)$ complexity
      $proj \leftarrow (|uu^\top e_1|, |uu^\top e_2|, \ldots, |uu^\top e_n|)$          $\triangleright \mathcal{O}(n)$ complexity
      $len, ind \leftarrow \text{SORT}(proj)$                     $\triangleright \mathcal{O}(n \log n)$ complexity
      $len \leftarrow \text{UNIQUE}(len)$
      $k \leftarrow |len|$
      **for** $i = 1, 2, \ldots, k$ **do**                         $\triangleright \mathcal{O}(n)$ complexity
         $x_i \leftarrow \sum_j e_{ind[j]}$ such that $proj_{ind[j]} = len[i]$
         $x_i \leftarrow x_i + c\boldsymbol{j}$
         $u_0 \leftarrow \text{NORMALIZE}(uu^\top x_i)$
         **if** $\|u_0\| > \varepsilon$ **then**            $\triangleright$ floating-point errors are considered
            substitute $u$ with $u_0$
            **break**
         **end if**
      **end for**
   **end for**
   **for** each multiple eigenvalue and its eigenvectors $U \in \mathbb{R}^{n \times d}$ **do**    $\triangleright \mathcal{O}(n^2 dm)$ complexity
      $proj \leftarrow (|UU^\top e_1|, |UU^\top e_2|, \ldots, |UU^\top e_n|)$      $\triangleright \mathcal{O}(n^2 d)$ complexity
      $len, ind \leftarrow \text{SORT}(proj)$
      $len \leftarrow \text{UNIQUE}(len)$
      **if** $k < d$ **then**
         **break**                           $\triangleright$ Assumption 2 not satisfied
      **end if**
      **for** $i = 1, 2, \ldots, d$ **do**                       $\triangleright \mathcal{O}(n)$ complexity
         $x_i \leftarrow \sum_j e_{ind[j]}$ such that $proj_{ind[j]} = len[i]$
         $x_i \leftarrow x_i + c\boldsymbol{j}$
      **end for**
      $U_{\text{span}} \leftarrow$ empty matrix of shape $n \times 0$
      $U_{\text{perp}} \leftarrow U$                      $\triangleright$ orthogonal complementary space
      **for** $i = 1, 2, \ldots, d$ **do**                       $\triangleright \mathcal{O}(nd^4)$ complexity
         $u_i \leftarrow U_{\text{perp}} U_{\text{perp}}^\top x_i$              $\triangleright \mathcal{O}(nd)$ complexity
         **if** $\|u_i\| < \varepsilon$ **then**           $\triangleright$ floating-point errors are considered
            **break**                   $\triangleright$ Assumption 3 not satisfied
         **end if**
         $u_i \leftarrow \text{NORMALIZE}(u_i)$
         substitute $U_{:,i}$ with $u_i$
         $U_{\text{span}} \leftarrow [U_{\text{span}}, u_i]$
         $U_{\text{base}} \leftarrow U_{\text{span}}$
         **for** $j = 1, 2, \ldots, d$ **do**                $\triangleright \mathcal{O}(nd^3)$ complexity
            $U_{\text{temp}} \leftarrow [U_{\text{base}}, U_{:,j}]$
            **if** $\text{rank}(U_{\text{temp}}) = j + 1$ **then**       $\triangleright \mathcal{O}(nd^2)$ complexity
               $U_{\text{base}} \leftarrow U_{\text{temp}}$
            **end if**
            **if** $U_{\text{base}} \in \mathbb{R}^{n \times d}$ **then**
               **break**
            **end if**
         **end for**
         $U_{\text{base}} \leftarrow \text{GRAMSCHMIDTORTHOGONALIZATION}(U_{\text{base}})$    $\triangleright \mathcal{O}(nd^2)$ complexity
         $U_{\text{perp}} \leftarrow (U_{\text{base}})_{:,i+1:d}$
      **end for**
   **end for**
   $U \leftarrow U\Lambda^{\frac{1}{2}}$                         $\triangleright \mathcal{O}(n^2)$ complexity
   **return** $U$

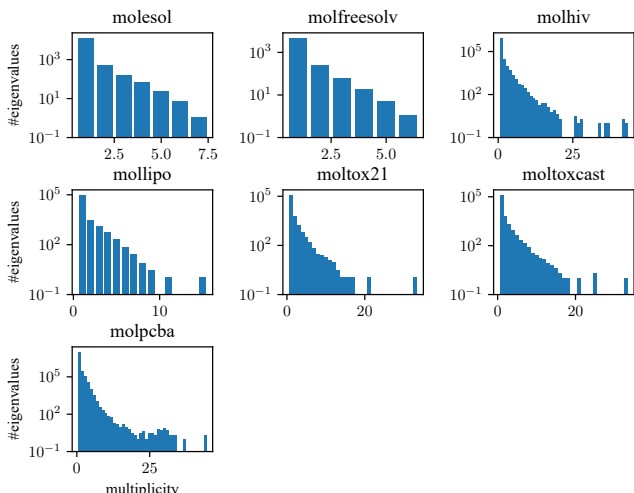

Figure 2: #Eigenvalues *w.r.t.* their multiplicities in real-world datasets (in logarithmic scale).

Table 10: The details of the datasets. Reg: Regression; Bin: Binary classification.

| Dataset | ogbg-molesol | ogbg-molfreesolv | ogbg-molhiv | ogbg-mollipo | ogbg-moltox21 | ogbg-moltoxcast | ogbg-molpcba |
|---|---|---|---|---|---|---|---|
| #Graphs | 1128 | 642 | 41127 | 4200 | 7831 | 8576 | 437929 |
| Avg #Nodes | 13.3 | 8.7 | 25.5 | 27.0 | 18.6 | 18.8 | 26.0 |
| Avg #Edges | 13.7 | 8.4 | 27.5 | 29.5 | 19.3 | 19.3 | 28.1 |
| #Tasks | 1 | 1 | 1 | 1 | 12 | 617 | 128 |
| Task Type | Reg | Reg | Bin | Reg | Bin | Bin | Bin |
| Metric | RMSE | RMSE | ROC-AUC | RMSE | ROC-AUC | ROC-AUC | AP |

---

**Algorithm 4** Verify the correctness of Theorem 3

$p\_correct \leftarrow 0$, $q\_correct \leftarrow 0$, $pq\_correct \leftarrow 0$, $total \leftarrow 0$
**for** $i = 1, 2, \ldots, trials$ **do**
    $n \leftarrow$ a random positive integer
    $U \leftarrow$ a random orthonormal matrix in $\mathbb{R}^{n \times n}$
    $U_0 \leftarrow \text{UNIQUESIGN}(U)$
    $P \leftarrow$ a random permutation matrix
    $V \leftarrow PU$
    $V_0 \leftarrow \text{UNIQUESIGN}(V)$
    $p\_correct \leftarrow p\_correct + 1$ **if** $|PU_0 - V_0| < \varepsilon$            ▷ test permutation-equivariance
    $S \leftarrow$ a random sign matrix (diagonal matrix of 1 and $-1$)
    $W \leftarrow US$
    $W_0 \leftarrow \text{UNIQUESIGN}(W)$
    $q\_correct \leftarrow q\_correct + 1$ **if** $|U_0 - W_0| < \varepsilon$                  ▷ test uniqueness
    $Y \leftarrow PW$
    $Y_0 \leftarrow \text{UNIQUESIGN}(Y)$
    $pq\_correct \leftarrow pq\_correct + 1$ **if** $|PU_0 - Y_0| < \varepsilon$               ▷ test both
    $total \leftarrow total + 1$
**end for**
print out the values of $p\_correct$, $q\_correct$, $pq\_correct$ and $total$

---

## C.3 Verifying the correctness of Theorem 5

We verify the correctness of Theorem 5 through random simulation. The program is shown in Algorithm 5. Let $U$ be a random orthonormal matrix in $\mathbb{R}^{n \times d}$, $P$ be a random permutation matrix and $Q$ be a random orthonormal matrix in $\mathbb{R}^{d \times d}$. We pass $U$, $PU$, $UQ$, $PUQ$ to the UNIQUEBASIS function (Algorithm 2) and compare the outputs. If our algorithm is correct, $U$ and $UQ$ should have invariant outputs, while $U$, $PU$ and $PUQ$ should have equivariant outputs.

---

**Algorithm 5** Verify the correctness of Theorem 5

---

$p\_correct \leftarrow 0$, $q\_correct \leftarrow 0$, $pq\_correct \leftarrow 0$, $total \leftarrow 0$
**for** $i = 1, 2, \ldots, trials$ **do**
    $n \leftarrow$ a random positive integer (greater than 1)
    $d \leftarrow$ a random positive integer (less than $n$)
    $U \leftarrow$ a random orthonormal matrix in $\mathbb{R}^{n \times d}$
    $U_0 \leftarrow$ UNIQUEBASIS($U$)
    $P \leftarrow$ a random permutation matrix
    $V \leftarrow PU$
    $V_0 \leftarrow$ UNIQUEBASIS($V$)
    $p\_correct \leftarrow p\_correct + 1$ **if** $|PU_0 - V_0| < \varepsilon$         ▷ test permutation-equivariance
    $Q \leftarrow$ a random orthonormal matrix in $\mathbb{R}^{d \times d}$
    $W \leftarrow UQ$
    $W_0 \leftarrow$ UNIQUEBASIS($W$)
    $q\_correct \leftarrow q\_correct + 1$ **if** $|U_0 - W_0| < \varepsilon$         ▷ test uniqueness
    $Y \leftarrow PW$
    $Y_0 \leftarrow$ UNIQUEBASIS($Y$)
    $pq\_correct \leftarrow pq\_correct + 1$ **if** $|PU_0 - Y_0| < \varepsilon$         ▷ test both
    $total \leftarrow total + 1$
**end for**
print out the values of $p\_correct$, $q\_correct$, $pq\_correct$ and $total$

---

We conduct 1000 trials. The results are $p\_correct = q\_correct = pq\_correct = total = 1000$, showing that Algorithm 2 is both permutation-equivariant and unique. The function UNIQUEBASIS raises an assertion error when either Assumption 2 or Assumption 3 is violated, so Algorithm 5 also shows that random orthonormal matrices violate these assumptions with probability 0.

## C.4 Verifying the correctness of Theorem 1

We conduct experiment on the EXP dataset proposed in Abboud et al. [1], which is designed to explicitly evaluate the expressive power of GNNs. The dataset consists of a set of 1-WL indistinguishable non-isomorphic graph pairs, and each graph instance is a graph encoding of a propositional formula. The classification task is to determine whether the formula is satisfiable (SAT). Since the graph pairs in the EXP dataset are not distinguishable by 1-WL test, if a model shows above 50% accuracy on this dataset, it should have expressive power beyond the 1-WL algorithm.

The models we used on EXP dataset are as follows: an 8-layer GCN, GIN [57], PPGN [31], 1-2-3-GCN-L [35], 3-GCN [1], DeepSets-RNI (DeepSets with Random Node Initialization (RNI) [1]), GCN-RNI (GCN with Random Node Initialization (RNI)), Linear-RSE (linear network with RSE), and DeepSets-RSE. GCN and GIN belongs to the family of MP-GNNs whose expressive power is bounded by 1-WL, and RNI is a method to improve the expressive power of MP-GNNs. We verify the expressive power gain of RSE-based models by comparing them with GCN, and evaluate their efficiency by comparing them with other expressive models.

We evaluate all models on the EXP dataset using 10-fold cross-validation, and train each model for 500 epochs per fold. Mean test accuracy across all folds is measured and reported. The results are reported in Table 11. In addition, we also measure the learning curves of models to show their convergence rate, as shown in Figure 3.

In Table 11, we observe that vanilla GCN and DeepSets-RNI only achieves 50% accuracy, because they do not have expressive power beyond the 1-WL test. DeepSets-RSE achieves the best performance among all models with a near 100% accuracy, which demonstrates the expressive power of

Table 11: Accuracy results on EXP.

| Model | Test Accuracy (%) |
|---|---|
| GCN | 50.0 |
| GIN | 50.0 |
| PPGN | 50.0 |
| 1-2-3-GCN-L | 50.0 |
| 3-GCN | $99.7 \pm 0.0$ |
| DeepSets-RNI | 50.0 |
| GCN-RNI | $97.6 \pm 2.5$ |
| Linear-RSE | $99.1 \pm 1.8$ |
| DeepSets-RSE | $\mathbf{99.8 \pm 0.5}$ |

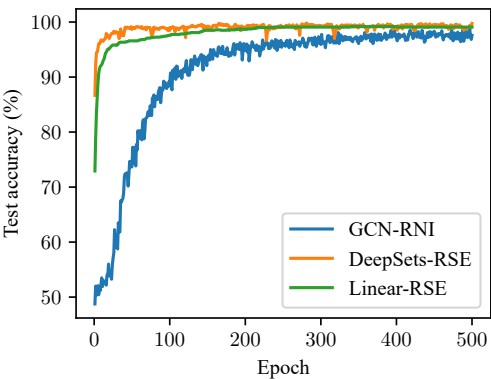

Figure 3: Learning curves on EXP.

RSE-based models is beyond the 1-WL test. Other models, namely Linear-RSE and GCN-RNI, also achieve comparable performance. It's worth mentioning that even a simple linear model (Linear-RSE) could achieve performance above 99%. This indicates that the universal expressiveness of models with RSE is mostly from RSE itself, rather than the network structure.

From Figure 3, we find that the convergence rate of RSE-based models are much faster than their RNI-based counterpart. This is because RSE-based models are deterministic, while RNI-based models are random and require more training epochs to converge. The structures of DeepSets-RSE and Linear-RSE are simpler, which means they also train much faster than GCN-RNI.

## D Synthetic experiment on basis invariance

As mentioned in Appendix B, SignNet [29] only deals with sign ambiguity, while BasisNet [29] has a prohibitive computational overhead. On the other hand, our proposed MAP addresses basis ambiguity efficiently, albeit with the existence of uncanonizable eigenspaces. In this section, we conduct a synthetic experiment to verify the ability of MAP on addressing basis ambiguity. We use graph isomorphic testing, a traditional graph task. Our focus is on 10 non-isomorphic random weighted graphs $\mathcal{G}_1, \ldots, \mathcal{G}_{10}$, all exhibiting basis ambiguity issues (with the first three eigenvectors belonging to the same eigenspace). We sample 20 instances for each graph, introducing different permutations and basis choices for the initial eigenspace. The dataset is then split into a 9:1 ratio for training and testing, respectively. The task is a 10-way classification, where the aim is to determine the isomorphism of a given graph to one of the 10 original graphs. The model is given the first 3 eigenvectors as input (*i.e.* $k = 3$). The results are averaged over 4 different runs.

Table 12: Test accuracy of the synthetic graph isomorphic testing task with DeepSets [63] using different PEs.

| Positonal Encoding | Accuracy |
|---|---|
| LapPE | $0.11 \pm 0.08$ |
| LapPE + RS | $0.10 \pm 0.09$ |
| LapPE + SignNet | $0.10 \pm 0.03$ |
| LapPE + MAP | $\mathbf{0.84 \pm 0.21}$ |

As evident from the results in Table 12, approaches that address sign ambiguity (like RandSign and SignNet) cannot obtain nontrivial performance on this task. Conversely, MAP shows commendable performance. The 84% accuracy, although impressive, indicates potential avenues for further enhancement. We believe this synthetic task could also serve as a valuable benchmark for future studies addressing basis invariance through canonization. Code for this experiment is available at https://github.com/GeorgeMLP/basis-invariance-synthetic-experiment.

# E  Other positional encoding methods

In this paper, we proposed MAP, which is a kind of positional encodings. In the field of graph representation learning, many other positional encoding methods have also been proposed. Murphy et al. [36] proposed Relational Pooling (RP) that assigns each node with an identifier that depends on the index ordering. They showed that RP-GNN is strictly more expressive than the original WL-GNN. However, to ensure permutation equivalence, we have to account for all possible $n!$ node orderings, which is computationally intractable. You et al. [59] proposed learnable position-aware embeddings by computing the distance of a target node to an anchor-set of nodes, and showed that P-GNNs have more expressive power than existing GNNs. However, the expressive power of their model is dependent on the random selection of anchor sets. Sato et al. [44], Abboud et al. [1] proposed to use full or partial random node features and proved that their model has universal expressive power, but it has several defects: (1) the loss of permutation invariance, (2) slower convergence, and (3) poor generalization on unseen graphs [59, 30]. Li et al. [28] proposed Distance Encoding (DE) that captures the distance between the node set whose representation is to be learned and each node in the graph. They proved that DE can distinguish node sets embedded in almost all regular graphs where traditional GNNs always fail. However, their approach fails on distance regular graphs, and computation of power matrices can be a limiting factor for their model's scalability. Dwivedi et al. [18] proposed Learnable Structural and Positional Encodings (LSPE) that decouples structural and positional representations by inserting MPGNNs-LSPE layers and showed promising performance on three molecular benchmarks.

In particular, we will show that with Random Node Initialization (RNI), (1) A linear network is universal on a fixed graph, and (2) An MLP with just one additional message passing layer can be universal on arbitrary graphs.

In our work, we denote RNI as concatenating a random matrix to the input node features. The random matrix can be sampled from Gaussian distribution, uniform distribution, etc. Without loss of generality, we will assume that each entry of the random matrix is sampled independently from the standard Gaussian distribution $N(0, 1)$.

**Definition 2.** *A GNN with RNI is defined by concatenating a random matrix $\mathbf{R}$ to the input node features, i.e., $\boldsymbol{X}' = [\boldsymbol{X}, \mathbf{R}]$, where $\boldsymbol{X}$ are the original node features, $\boldsymbol{X}'$ are the modified node features and each entry of $\mathbf{R}$ is sampled independently from the standard Gaussian distribution $N(0, 1)$. The value of $\mathbf{R}$ is sampled at every forward pass of GNN.*

To study the effects of RNI on the expressiveness of GNNs, we consider two types of tasks: tasks on a fixed graph (e.g., node classification) and tasks on arbitrary graphs (e.g. graph classification). On a fixed graph, we aim to learn a function $f: \mathbb{R}^{n \times d} \to \mathbb{R}^{n \times d'}$ that transforms the feature of each node $v_i$ to a presentation vector $\boldsymbol{Z}_{i,:} \in \mathbb{R}^{1 \times d'}$. We claim that a linear GNN with RNI in the form

$$[\boldsymbol{X}, \mathbf{R}]\boldsymbol{W} = \boldsymbol{Z} \tag{6}$$

is universal, where $\boldsymbol{W} \in \mathbb{R}^{d \times d'}$ are the network parameters, and $\boldsymbol{Z} \in \mathbb{R}^{n \times d'}$ is the desired output. In other words, we have the universality theorem of linear GNNs with RNI on a fixed graph:

**Theorem 7.** *On a fixed graph $\mathcal{G}$, a linear GNN with RNI defined by Equation 6 is equivariant and can produce any prediction $\boldsymbol{Z} \in \mathbb{R}^{n \times d'}$ with probability $1$.*

We prove Theorem 7 in Appendix K.8.

On arbitrary graphs, the target function is not only dependent on the node features, but on the graph structure $\hat{\boldsymbol{A}} \in \mathbb{R}^{n \times n}$ as well. Let $\Omega \subset \mathbb{R}^{n \times d} \times \mathbb{R}^{n \times n}$ be a compact set of graphs with $[\boldsymbol{X}, \hat{\boldsymbol{A}}] \in \Omega$, where $\boldsymbol{X}$ are the node features and $\hat{\boldsymbol{A}}$ is the normalized adjacency matrix. We wish to learn a function $f: \Omega \to \mathbb{R}$ that transforms each graph to its label. Since $f$ is also dependent on $\hat{\boldsymbol{A}}$, an MLP-based network with $\boldsymbol{X}'$ as input is not expressive enough, and we need additional graph convolutional layers to obtain information about the graph structure. However, Puny et al. [41] proved that with just *one* additional message passing layer, an MLP with RNI can approximate any continuous invariant graph function $f$.

**Theorem 8** (Puny et al. [41]). *Given a compact set of graphs $\Omega \subset \mathbb{R}^{n \times d} \times \mathbb{R}^{n \times n}$, a GNN with one message passing layer, an MLP network with RNI can approximate an arbitrary continuous invariant graph function $f: \Omega \to \mathbb{R}$ to an arbitrary precision.*

Theorem 8 is a direct inference of the proof of Proposition 1 in Puny et al. [41], where the authors constructed a RGNN that first transfers the graph structural information to the node features through a message passing layer, and then approximates $f$ with a DeepSets network, which is an MLP-based network.

# F  Toy Examples

## F.1  Toy examples for Algorithm 1

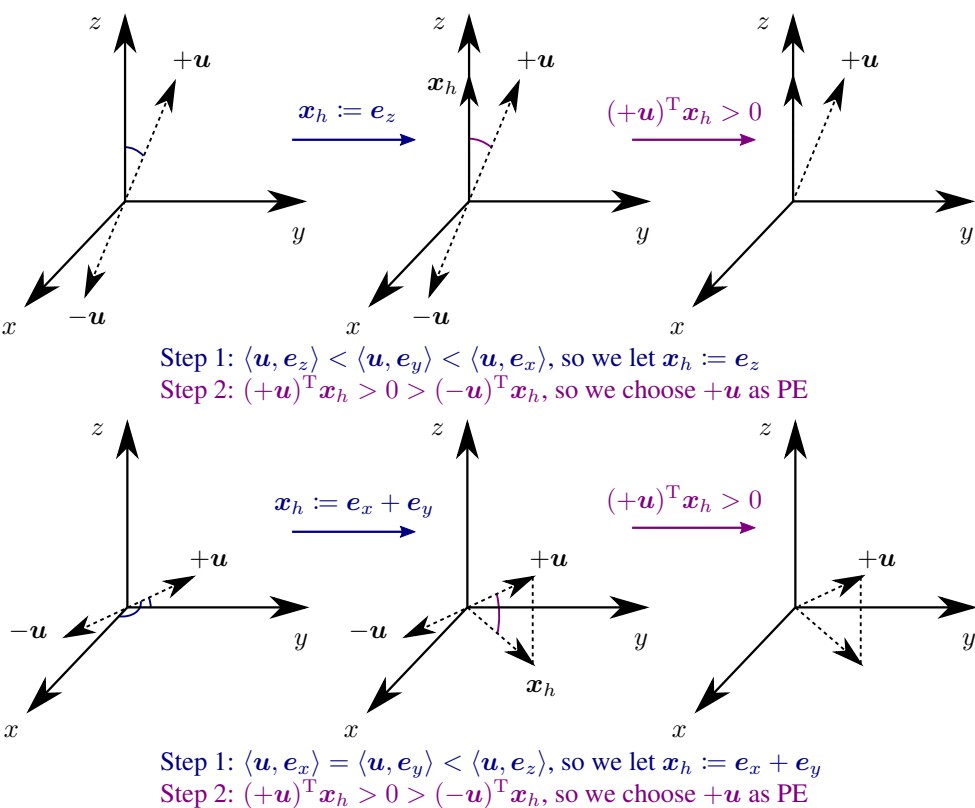

Step 1: $\langle \boldsymbol{u}, \boldsymbol{e}_z \rangle < \langle \boldsymbol{u}, \boldsymbol{e}_y \rangle < \langle \boldsymbol{u}, \boldsymbol{e}_x \rangle$, so we let $\boldsymbol{x}_h := \boldsymbol{e}_z$
Step 2: $(+\boldsymbol{u})^{\mathrm{T}} \boldsymbol{x}_h > 0 > (-\boldsymbol{u})^{\mathrm{T}} \boldsymbol{x}_h$, so we choose $+\boldsymbol{u}$ as PE

Step 1: $\langle \boldsymbol{u}, \boldsymbol{e}_x \rangle = \langle \boldsymbol{u}, \boldsymbol{e}_y \rangle < \langle \boldsymbol{u}, \boldsymbol{e}_z \rangle$, so we let $\boldsymbol{x}_h := \boldsymbol{e}_x + \boldsymbol{e}_y$
Step 2: $(+\boldsymbol{u})^{\mathrm{T}} \boldsymbol{x}_h > 0 > (-\boldsymbol{u})^{\mathrm{T}} \boldsymbol{x}_h$, so we choose $+\boldsymbol{u}$ as PE

Figure 4: A toy example illustrating our algorithm for eliminating sign ambiguity. **Top**: The angle between the $z$-axis and $\boldsymbol{u}$ is the smallest, so we choose $+\boldsymbol{u}$ to maximize $\boldsymbol{u}^{\top} \boldsymbol{e}_z$. **Bottom**: The angle between both $x$ and $y$-axes and $\boldsymbol{u}$ are the smallest, so we choose $+\boldsymbol{u}$ to maximize $\boldsymbol{u}^{\top} (\boldsymbol{e}_x + \boldsymbol{e}_y)$.

We give toy examples to help illustrate our MAP-sign algorithm. As shown on the top row of Figure 4, we have $n = 3$ and two possible sign choices for the eigenvector $\boldsymbol{u}$. Our algorithm first compares the angles (or equivalently, the absolute value of inner product) between $\boldsymbol{u}$ and the standard basis: $\boldsymbol{e}_x, \boldsymbol{e}_y, \boldsymbol{e}_z$, and pick the smallest one (the one with the largest absolute inner product), in this case $\boldsymbol{e}_z$. Thus we let $\boldsymbol{x}_h = \boldsymbol{e}_z$ and choose the sign that maximize $\boldsymbol{u}^{\top} \boldsymbol{x}_h$. In the first example we have $(+\boldsymbol{u})^{\top} \boldsymbol{x}_h > 0 > (-\boldsymbol{u})^{\top} \boldsymbol{x}_h$, thus $+\boldsymbol{u}$ is chosen instead of $-\boldsymbol{u}$. This choice is *unique* and *permutation-equivariant*.

It is possible though, that the angle between $\boldsymbol{u}$ and more than 1 basis vectors are equal. As shown on the bottom row of Figure 4, the angle between both $\boldsymbol{e}_x, \boldsymbol{e}_y$ and $\boldsymbol{u}$ are maximum. In this case we let $\boldsymbol{x}_h = \boldsymbol{e}_x + \boldsymbol{e}_y$ be their sum and maximize $\boldsymbol{u}^{\top} \boldsymbol{x}_h$, thus $+\boldsymbol{u}$ is chosen. A special case is when $\boldsymbol{u}$ and $\boldsymbol{x}_h$ are perpendicular. If this happens, we go on to pick the basis vector with the second largest angle and continue this process.

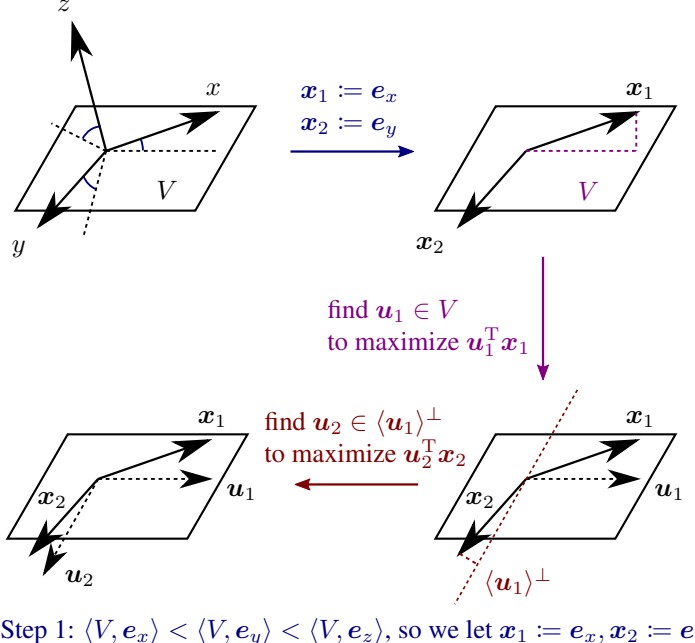

Step 1: $\langle V, \boldsymbol{e}_x \rangle < \langle V, \boldsymbol{e}_y \rangle < \langle V, \boldsymbol{e}_z \rangle$, so we let $\boldsymbol{x}_1 := \boldsymbol{e}_x, \boldsymbol{x}_2 := \boldsymbol{e}_y$
Step 2: find $\boldsymbol{u}_1 \in V$ to maximize $\boldsymbol{u}_1^{\mathrm{T}} \boldsymbol{x}_1$
Step 3: find $\boldsymbol{u}_2 \in \langle \boldsymbol{u}_1 \rangle^{\perp}$ to maximize $\boldsymbol{u}_2^{\mathrm{T}} \boldsymbol{x}_2$

Figure 5: A toy example illustrating our algorithm for eliminating basis ambiguity. First, we sort the angles between $V$ and the standard basis vectors, and set $\boldsymbol{x}_1 := \boldsymbol{e}_x, \boldsymbol{x}_2 := \boldsymbol{e}_y$. Next, we choose $\boldsymbol{u}_1 \in V$ that maximizes $\boldsymbol{u}_1^{\mathrm{T}} \boldsymbol{x}_1$. Finally, we choose $\boldsymbol{u}_2 \in \langle \boldsymbol{u}_1 \rangle^{\perp}$ that maximizes $\boldsymbol{u}_2^{\mathrm{T}} \boldsymbol{x}_2$. This gives us a unique basis $\boldsymbol{u}_1, \boldsymbol{u}_2$ of $V$.

### F.2 Toy examples for Algorithm 2

We give a toy example to help illustrate our MAP-basis algorithm. As shown in Figure 5, we have $n = 3, d = 2$ with a two-dimensional eigenspace $V$. Our algorithm first compares the angles (or equivalently, the length of orthogonal projection) between $V$ and the standard basis: $\boldsymbol{e}_x, \boldsymbol{e}_y, \boldsymbol{e}_z$, and pick the two smallest one (the two with the largest and second largest lengths of orthogonal projection), in this case $\boldsymbol{e}_x$ and $\boldsymbol{e}_y$. Thus we let $\boldsymbol{x}_1 = \boldsymbol{e}_x, \boldsymbol{x}_2 = \boldsymbol{e}_y$. Then we choose $\boldsymbol{u}_1 \in V$ that maximizes $\boldsymbol{u}_1^{\top} \boldsymbol{x}_1$, which is just the orthogonal projection of $\boldsymbol{x}_1$ onto $V$. Finally we choose $\boldsymbol{u}_2 \in \langle \boldsymbol{u}_1 \rangle^{\perp}$ that maximizes $\boldsymbol{u}_2^{\top} \boldsymbol{x}_2$, which is the orthogonal projection of $\boldsymbol{x}_2$ onto $\langle \boldsymbol{u}_1 \rangle^{\perp}$. This gives us a basis $\boldsymbol{u}_1, \boldsymbol{u}_2$ of $V$. This choice is *unique* and *permutation-equivariant*.

It is possible though, that the angle between $V$ and more than 1 basis vectors are equal. For example, if $\langle V, \boldsymbol{e}_x \rangle = \langle V, \boldsymbol{e}_y \rangle$ are both the smallest angles between $V$ and the standard basis, then we let $\boldsymbol{x}_1 = \boldsymbol{e}_x + \boldsymbol{e}_y$. The same goes for the second smallest angle, the third smallest angle, and so on.

## G Discussions on the assumptions

### G.1 Discussions on Assumption 1

The purpose of Section 3.3.1 is to uniquely determine the signs of the eigenvectors while also preserving their permutation-equivariance. One could easily come up with simple solutions such as choosing the signs such that the sum of the entries of eigenvectors are positive, or signs such that the element with the greatest absolute value is positive. In fact, Lim et al. [29] has proposed a more general way of choosing signs:

> We also consider ablations in which we ... choose a canonical sign for each eigenvector by maximizing the norm of positive entries.

This "canonical sign" approach did not work well because (1) a large percentage of eigenvectors in real-world datasets have the same norm for positive and negative entries, thus this approach does not actually solve sign ambiguity of these eigenvectors; (2) this approach cannot be generalized to multiple eigenvalues. As a reference, we list the number of eigenvectors that "canonical sign" fails in Table 13.

Table 13: The number of eigenvectors with the same norm for positive and negative entries, the total number of eigenvectors, and the ratio of eigenvectors that "canonical sign" approach fails in real-world datasets.

| Dataset | ogbg-molesol | ogbg-molfreesolv | ogbg-molhiv | ogbg-mollipo | ogbg-moltox21 | ogbg-moltoxcast | ogbg-molpcba |
|---|---|---|---|---|---|---|---|
| #Violation | 4060 | 2135 | 209854 | 15115 | 31365 | 35174 | 1536415 |
| #Eigenvectors | 14991 | 5600 | 1049163 | 113568 | 145459 | 161088 | 11373137 |
| Ratio | 27.08 % | 38.13 % | 20.00 % | 9.84 % | 21.56 % | 21.84 % | 13.51 % |

Some examples of such eigenvectors in real-world datasets are as follows.

```
[ 0.0000,  0.0000, -0.0000,  0.0000,  0.1364, -0.0965, -0.0965, -0.1575,
  0.0965, -0.0000,  0.1575, -0.0965, -0.2784,  0.2227, -0.1364, -0.0000,
  0.3341, -0.2363, -0.2363, -0.1364,  0.2227, -0.0000, -0.0487,  0.0345,
  0.0345, -0.0877,  0.0620,  0.0620,  0.2784, -0.2227,  0.1364,  0.0000,
 -0.3341,  0.2363,  0.2363,  0.1364, -0.2227],
[ 0.0000,  0.0000,  0.0000, -0.0796,  0.0796,  0.0975, -0.0796, -0.0563,
  0.0000,  0.2087, -0.3615,  0.2951,  0.0000, -0.1815,  0.1815,  0.4767,
 -0.4767, -0.0029,  0.0051, -0.0042,  0.0000, -0.0458,  0.0458,  0.0416,
 -0.0416, -0.1495,  0.2589, -0.2114,  0.0000, -0.0975,  0.0975, -0.1139,
  0.1139],
[ 0.0000,  0.0000,  0.0000, -0.0025,  0.0025,  0.0031, -0.0025, -0.0018,
  0.0000, -0.0025,  0.0043, -0.0035, -0.0000, -0.0987,  0.0987,  0.0952,
 -0.0952, -0.1271,  0.2202, -0.1798,  0.0000, -0.3234,  0.3234,  0.1436,
 -0.1436,  0.1313, -0.2275,  0.1858,  0.0000, -0.2524,  0.2524,  0.4381,
 -0.4381].
```

Assumption 1, on the other hand, is less restrictive. It requires that the eigenvectors are not perpendicular to at least one of the vectors $x_i$. For random unit vectors or random weighted graphs, Assumption 1 has 0 possibility of being violated. The number of eigenvectors violating Assumption 1 in real-world datasets are listed in Table 14. It can be observed that the ratio of violation tends to become smaller as the graph size becomes larger.

Table 14: The number of eigenvectors violating Assumption 1, the total number of eigenvectors, and the ratio of violation in real-world datasets. We ignore small graphs with no more than 5 nodes.

| Dataset | ogbg-molesol | ogbg-molfreesolv | ogbg-molhiv | ogbg-mollipo | ogbg-moltox21 | ogbg-moltoxcast | ogbg-molpcba |
|---|---|---|---|---|---|---|---|
| #Violation | 727 | 388 | 29558 | 3328 | 5418 | 6032 | 343088 |
| #Eigenvectors | 14551 | 5048 | 1049101 | 113568 | 144421 | 159987 | 11372381 |
| Ratio | 5.00 % | 7.69 % | 2.82 % | 2.93 % | 3.75 % | 3.77 % | 3.02 % |

### G.2 Discussions on Assumption 2 and Assumption 3

Assumption 2 requires that the projections of $e_1, e_2, \ldots, e_n$ on $V$ have at least $d$ distinct lengths, while Assumption 3 requires that each $x_i$ is not perpendicular to $\langle u_1, \ldots, u_{i-1} \rangle^\perp$, which is a subspace of $V$. We illustrate two examples of these assumptions being violated when $n = 3$ and $d = 2$ in Figure 6. In the left figure, the projections of $e_1, e_2, e_3$ on $V$ all have the same lengths, thus $k = 1 < d$, violating Assumption 2. In the right figure, $x_2$ is perpendicular to the orthogonal complementary space of $\text{span}(u_1)$ in $V$, violating Assumption 3. We observe that the eigenspace $V$ needs to obey certain kinds of symmetries in order to violate either assumptions. For random orthonormal matrices and random weighted graphs, these assumptions have 0 possibility of being violated; and in real-world datasets, the ratio of violation tends to become smaller as the graph size becomes larger.

The number of multiple eigenvalues violating these assumptions in real-world datasets are listed in Table 15. Indeed there is still around 20 % of violation on large datasets, but since multiple

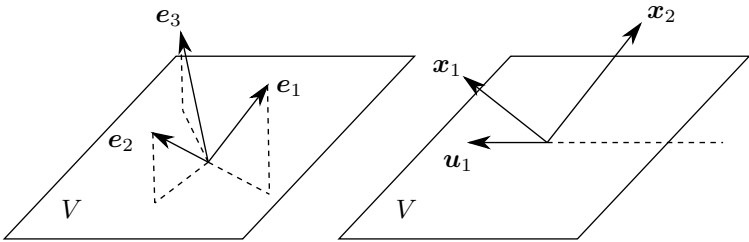

Figure 6: Examples of Assumption 2 and Assumption 3 being violated when $n = 3$ and $d = 2$.
**Left:** the projections of $e_1, e_2, e_3$ are of the same length, thus $k = 1 < d$, violating Assumption 2.
**Right:** $x_2$ is perpendicular to the orthogonal complementary space of $\text{span}(u_1)$ in $V$, violating Assumption 3.

eigenvalues only make up a small portion of all eigenvalues (Table 9), the ratio of violation is relatively small and has negligible influence on the model performance.

Table 15: The number of eigenvalues violating Assumption 2 or Assumption 3 in real-world datasets. $n_1$: the number of eigenvalues violating Assumption 2. $n_2$: the number of eigenvalues not violating Assumption 2 but violating Assumption 3. $N_1$: the total number of multiple eigenvalues. $p_1$: the ratio of multiple eigenvalues violating Assumption 2. $p_2$: the ratio of multiple eigenvalues violating Assumption 3. $N_2$: the total number of eigenvalues. $p_3$: the ratio of all eigenvalues violating Assumption 2. $p_4$: the ratio of all eigenvalues violating Assumption 3. We ignore small graphs with no more than 5 nodes.

| Dataset | ogbg-molesol | ogbg-molfreesolv | ogbg-molhiv | ogbg-mollipo | ogbg-moltox21 | ogbg-moltoxcast | ogbg-molpcba |
|---|---|---|---|---|---|---|---|
| $n_1$ | 39 | 30 | 5315 | 440 | 646 | 873 | 30844 |
| $n_2$ | 126 | 45 | 6329 | 599 | 1073 | 1450 | 61318 |
| $N_1$ | 738 | 286 | 52367 | 5391 | 8772 | 10556 | 491247 |
| $p_1$ | 5.28 % | 10.49 % | 10.15 % | 8.16 % | 7.36 % | 8.27 % | 6.28 % |
| $p_2$ | 17.07 % | 15.73 % | 12.09 % | 11.11 % | 12.23 % | 13.74 % | 12.48 % |
| $N_2$ | 13420 | 4654 | 952055 | 104669 | 129730 | 141042 | 10627757 |
| $p_3$ | 0.29 % | 0.64 % | 0.56 % | 0.42 % | 0.50 % | 0.62 % | 0.29 % |
| $p_4$ | 0.94 % | 0.97 % | 0.66 % | 0.57 % | 0.83 % | 1.03 % | 0.58 % |

## H   Discussions on random graphs

In this section we summarize some existing results on random graphs and more generally random matrices. These discussions help us to have a better understanding of how eigenvalues and eigenvectors of random graphs distribute, and the probability that they are uncanonizable by our MAP algorithm. Due to the theoretical complexity, we do make some simplifications in our discussions.

We first give some basic definitions about random matrices and random graphs.

**Definition 3** (Wigner matrix). *Let $\xi, \zeta$ be real random variables with mean zero. We say $W$ is a **Wigner matrix** of size $n$ with atom variables $\xi, \zeta$ if $W = (w_{ij})_{i,j=1}^{n}$ is a random real symmetric $n \times n$ matrix that satisfies the following conditions.*

- *(independence) $\{w_{ij} : 1 \le i \le j \le n\}$ is a collection of independent random variables.*

- *(off-diagonal entries) $\{w_{ij} : 1 \le i < j \le n\}$ is a collection of independent and identically distributed (iid) copies of $\xi$.*

- *(diagonal entries) $\{w_{ii} : 1 \le i \le n\}$ is a collection of iid copies of $\zeta$.*

If $\xi$ and $\zeta$ have the same distribution, we say $W$ is a Wigner matrix with atom variable $\xi$. We always assume that $\xi$ is *non-degenerate*, namely that there is no value $c$ such that $\mathbb{P}(\xi = c) = 1$.

**Definition 4** (symmetric Bernoulli matrix). *Let $0 < p < 1$, and take $\xi$ to be the random variable*

$$\xi := \begin{cases} 1 - p, & \text{with probability } p, \\ -p, & \text{with probability } 1 - p. \end{cases}$$

*Then $\xi$ has zero mean. Let $B_n(p)$ denote the $n \times n$ Wigner matrix with atom variable $\xi$. We refer to $B_n(p)$ as a **symmetric Bernoulli matrix** (with parameter $p$).*

**Definition 5** (sub-exponential). *A random variable $\xi$ is called **sub-exponential** with exponent $\alpha > 0$ if there exists a constant $\beta > 0$ such that*

$$\mathbb{P}(|\xi| > t) \leq \beta \exp(-t^\alpha/\beta) \quad \text{for all } t > 0.$$

**Definition 6** (Erdős-Rényi random graph). *Let $G(n, p)$ denote the Erdős-Rényi random graph on $n$ vertices with edge density $p$. That is, $G(n, p)$ is a simple graph on $n$ vertices such that each edge $\{i, j\}$ is in $G(n, p)$ with probability $p$, independent of other edges.*

Let $A_n(p)$ be the zero-one adjacency matrix of $G(n, p)$. $A_n(p)$ is not a Wigner matrix since its entries do not have mean zero. Let $\tilde{G}(n, p)$ denote the Erdős-Rényi random graph with loops on $n$ vertices with edge density $p$. Let $\tilde{A}_n(p)$ denote the zero-one adjacency matrix of $\tilde{G}(n, p)$. Technically, $\tilde{A}_n(p)$ is not a Wigner random matrix because its entries do not have mean zero. However, we can view $\tilde{A}_n(p)$ as a low rank deterministic perturbation of a Wigner matrix. That is, we can write $\tilde{A}_n(p)$ as

$$\tilde{A}_n(p) = p\boldsymbol{J}_n + B_n(p),$$

where $\boldsymbol{J}_n$ is the all-ones matrix.

**Simplification in our discussions.** The adjacency matrix of Erdős-Rényi random graphs can be viewed as a rank-one perturbation of a zero-mean symmetric random matrix, which has no effect on the distribution of eigenvalues in the limit $n \to \infty$ [8]. In the finite case, the difference between eigenvalues and eigenvectors of the perturbed and unperturbed random matrices can be bounded as well, as stated in the following theorems.

**Theorem 9** (O'Rourke et al. [38]). *Let $E$ be an $n \times n$ Bernoulli random matrix, and let $A$ be an $n \times n$ matrix with rank $r$. Let $\sigma_1 \geq \sigma_2 \geq \cdots \geq \sigma_{\min\{m,n\}} \geq 0$ be singular values of $A$, $v_1, v_2, \ldots, v_{\min\{m,n\}}$ be corresponding singular vectors of $A$, $\sigma'_1 \geq \cdots \geq \sigma'_{\min\{m,n\}} \geq 0$ be singular values of $A + E$, $v'_1, \ldots, v'_{\min\{m,n\}}$ be corresponding singular vectors of $A + E$. For every $\varepsilon > 0$ there exists constants $C_0, \delta_0 > 0$ (depending only on $\varepsilon$) such that if $\delta > \delta_0$ and $\sigma_1 \geq \max\{n, \sqrt{n}\delta\}$, then, with probability at least $1 - \varepsilon$,*

$$\sin \angle(v_1, v'_1) \leq C\frac{\sqrt{r}}{\delta}.$$

**Theorem 10** (O'Rourke et al. [38]). *Let $E$ be an $n \times n$ Bernoulli random matrix, and let $A$ be an $n \times n$ matrix with rank $r$ satisfying $\sigma_1 \geq n$. For every $\varepsilon > 0$, there exists a constant $C_0 > 0$ (depending only on $\varepsilon$) such that, with probability at least $1 - \varepsilon$,*

$$\sigma_1 - C \leq \sigma'_1 \leq \sigma_1 + C\sqrt{r}.$$

In particular, when the rank $r$ is significantly smaller than $n$, the bounds in the above theorems are significantly better. Thus the rank-one perturbation has little or no effect on the spectral distribution of the adjacency matrix. In the following discussions we will ignore the rank-one perturbation and assume that the adjacency matrix of random graphs have zero mean.

We will study two cases: (1) the adjacency matrix of the random graph is continuously distributed (random weighted graph); (2) the adjacency matrix of the random graph is discretely distributed (random unweighted graph).

It is easy to see that for a Wigner matrix with atom variables $\xi, \zeta$, if the distribution of $\xi$ is continuous, then with probability 1 it has simple spectrum (*i.e.*, all eigenvalues have multiplicity one), thus no basis ambiguity exists. For its single eigenvectors, we also expect these eigenvectors to be continuously distributed on the unit sphere, thus the probability that they are uncanonizable is equal to 1. In fact, we can even give a explicit formula for the distribution of some simple random matrices, as follows.

**Definition 7** (GOE). *The Gaussian orthogonal ensemble (GOE) is defined as a Wigner random matrix with atom variables $\xi, \zeta$, where $\xi$ is a standard normal random variable and $\zeta$ is a normal random variable with mean zero and variance 2.*

**Theorem 11** (Anderson et al. [4], Section 2.5.1). *Let $M$ be a $n \times n$ real symmetric matrix drawn from the GOE, with eigenvalues $\lambda_1 \leq \cdots \leq \lambda_n$ and corresponding eigenvectors $v_1, \ldots, v_n$. Then the eigenvectors $v_1, \ldots, v_n$ are uniformly distributed on*

$$S_+^{n-1} := \{x = (x_1, \ldots, x_n) \in S^{n-1} : x_1 > 0\},$$

*and the eigenvalues $\lambda_1, \ldots, \lambda_n$ have joint density*

$$p_n(\lambda_1, \ldots, \lambda_n) := \begin{cases} \frac{1}{Z_n} \prod_{1 \leq i < j \leq n} |\lambda_i - \lambda_j| \prod_{i=1}^n e^{-\lambda_i^2/4}, & \text{if } \lambda_1 \leq \cdots \leq \lambda_n, \\ 0, & \text{otherwise,} \end{cases}$$

*where $Z_n$ is a normalization constant.*

Thus the eigenvectors of such Gaussian random matrices are uniformly distributed on the unit sphere, meaning they are canonizable with probability 1.

The discrete case is trickier, because when $n$ is finite, the probability that the eigenvectors are uncanonizable is no longer 0. We wish that this probability can be upper bounded and asymptotically converges to zero. The first fact is that for a large class of random matrices, perturbed or not, they have simple spectrum with probability $1 - o(1)$.

**Theorem 12** (Tao and Vu [50]). *Consider real symmetric random matrices $M_n = (\xi_{ij})_{1 \leq i,j \leq n}$, where the entries $\xi_{ij}$ for $i \leq j$ are jointly independent with $\xi_{ji} = \xi_{ij}$, the upper-triangular entries $\xi_{ij}, i < j$ have non-trivial distribution $\xi$ for some fixed $\mu > 0$ such that $\mathbb{P}(\xi = x) \leq 1 - \mu$ for all $x \in \mathbb{R}$. The diagonal entries $\xi_{ii}, 1 \leq i \leq n$ can have an arbitrary real distribution (and can be correlated with each other), but are required to be independent of the upper diagonal entries $\xi_{ij}, 1 \leq i < j \leq n$. Then for every fixed $A > 0$ and $n$ sufficiently large (depending on $A, \mu$), the spectrum of $M_n$ is simple with probability at least $1 - n^{-A}$.*

Thus with probability $1 - o(1)$ no basis ambiguity exists.

For sign ambiguity, we expect that the probability that the single eigenvectors are canonizable asymptotically converges to 1 as $n \to \infty$. Denote $\boldsymbol{j} = (1, 1, \ldots, 1) \in \mathbb{R}^n$ the all-one vector, $\hat{\boldsymbol{j}} = (\frac{1}{\sqrt{n}}, \frac{1}{\sqrt{n}}, \ldots, \frac{1}{\sqrt{n}}) \in \mathbb{R}^n$ the normalized all-one vector. One obvious fact is that for an eigenvector $\boldsymbol{u} \in \mathbb{R}^n$, if it is non-canonizable, then $\boldsymbol{u} \cdot \boldsymbol{j} = 0$. Thus it suffices to show that the probability of the inner product between $\boldsymbol{u}$ and $\boldsymbol{j}$ being zero converges to 0 as $n \to \infty$. This can be derived from the following theorem.

**Theorem 13** (Tao and Vu [49]). *Let $\xi, \zeta$ be random variables such that*

- *$\xi$ and $\zeta$ are sub-exponential random variables,*

- *$\mathbb{E}(\xi) = \mathbb{E}(\zeta) = \mathbb{E}(\xi^3) = 0$,*

- *$\mathbb{E}(\xi^2) = 1, \mathbb{E}(\xi^4) = 3, \mathbb{E}(\zeta^2) = 2$,*

*and assume $\xi$ is a symmetric random variable. For each $n \geq 1$, let $W_n$ be an $n \times n$ Wigner matrix with atom variables $\xi, \zeta$. Let $\{a_n\}$ be a sequence of unit vectors with $a_n \in S^{n-1}$ such that $\lim_{n \to \infty} \|a_n\|_{\ell^\infty} = 0$, and let $\{i_n\}$ be a sequence of indices with $i_n \in [n]$. Then*

$$\sqrt{n} v_{i_n}(W_n) \cdot a_n \to N(0, 1)$$

*in distribution as $n \to \infty$.*

Thus in the limit as $n \to \infty$, by taking $a_n = \hat{\boldsymbol{j}}$, we have the inner product between the eigenvector and $\boldsymbol{j}$ following a normal distribution. The probability that they have zero inner product is 0, indicating these eigenvectors are almost always canonizable.

**Related readings.** We refer the readers to Anderson et al. [4] for a introduction of random matrices, Benaych-Georges and Nadakuditi [8] for a survey about the singular values and vectors of low rank perturbations of large random matrices, O'Rourke et al. [37] for a comprehensive survey about eigenvectors of random matrices.

# I  Further weakening the assumptions

## I.1  Further weakening Assumption 1

In Section 3.2, we mentioned that some eigenvectors are intrinsically *uncanonizable*, meaning that it is impossible to canonize them based on themselves. However, in graph-level tasks, the input is a whole graph with $n$ eigenvectors, not a single eigenvector. When the input graph is permuted by a permutation $\sigma$, all the eigenvectors are also permuted in the same way. This raises the question of whether we can canonize one eigenvector based on other eigenvectors in the same graph.

Thus we propose to further weaken Assumption 1 in the following steps. For a given input graph, we first divide its eigenvectors into two sets: $S_1$, containing all uncanonizable eigenvectors; and $S_2$, containing all canonizable eigenvectors. Suppose $|S_1| = d_1, |S_2| = d_2$, where $d_1 + d_2 = n$. Define the matrix of canonizable eigenvectors

$$U_{\text{can}} = [u_1, \ldots, u_{d_2}], \text{ where } u_1, \ldots, u_{d_2} \text{ are all the eigenvectors in } S_2.$$

We fist canonize all the eigenvectors in $S_2$, using our MAP-sign algorithm. This gives us a matrix of canonized eigenvectors $U_{\text{can}}^* \in \mathbb{R}^{n \times d_2}$. Using a hash function, we can compress the matrix $U_{\text{can}}^*$ into a vector $u_{\text{can}}$ that preserves all the information of $S_2$:

$$u_{\text{can}} \in \mathbb{R}^n, \text{ where } (u_{\text{can}})_i = \text{hash}\{(U_{\text{can}}^*)_{i,:}\}, 1 \leq i \leq n.$$

Then we can define the summary vectors $x_i$ of $u_{\text{can}}$ in the same way as in Section 3.3, and use them to canonize the eigenvectors in $S_1$. Denote the projection matrix $P = u_{\text{can}} u_{\text{can}}^\top$ and the projected angles $\alpha_i = |Pe_i|, 1 \leq i \leq n$. Assume that there are $k$ distinct values in $\{\alpha_i, i = 1, \ldots, n\}$, according to which we can divide all basis vectors $\{e_i\}$ into $k$ disjoint groups $\mathcal{B}_i$ (arranged in descending order of the distinct angles). Each $\mathcal{B}_i$ represents an equivalent class of axis that $u_{\text{can}}$ has the same projection on. Then we define a summary vector $x_i$ of $u_{\text{can}}$ for the axes in each group $\mathcal{B}_i$ as their total sum $x_i = \sum_{e_j \in \mathrm{B}_i} e_j + c$, where $c$ is a tunable constant.

For each $u \in S_1$, we can try to canonize it using the MAP-sign algorithm as in Section 3.3.1, with the summary vectors of $u$ replaced by the summary vectors of $u_{\text{can}}$. We find the first non-orthongonal summary vector with $u$, denoted by $x_h$, and choose the sign that maximizes $u^\top x_h$. In this way, we are able to canonize some eigenvectors in $S_1$ that are originally uncanonizable, thus further weakening Assumption 1. The complete workflow is shown in Algorithm 6.

---

**Algorithm 6** A stronger algorithm for eliminating sign ambiguity

---

**Require:** Input graph $\mathcal{G} = (\mathbb{V}, \mathbb{E}, X)$
**Ensure:** Spectral embedding of $\mathcal{G}$

   Calculate the eigendecomposition $\hat{A} = U \Lambda U^\top$
   $S_1 \leftarrow$ the set of all uncanonizable eigenvectors of $\mathcal{G}$
   $S_2 \leftarrow$ the set of all canonizable eigenvectors of $\mathcal{G}$
   Canonize all eigenvectors in $S_2$, using Algorithm 1
   $U_{\text{can}}^* \leftarrow [u_1^*, \ldots, u_{d_2}^*]$, where $u_1^*, \ldots, u_{d_2}^*$ are all the canonized eigenvectors, $d_2 = |S_2|$
   $u_{\text{can}} \leftarrow \left( \text{hash}\{(U_{\text{can}}^*)_{1,:}\}, \ldots, \text{hash}\{(U_{\text{can}}^*)_{n,:}\} \right)^\top$
   $P \leftarrow u_{\text{can}} u_{\text{can}}^\top$
   $\alpha_i \leftarrow |Pe_i|, 1 \leq i \leq n$
   $k \leftarrow$ the number of distinct values in $\{\alpha_i\}$
   Divide all basis vectors $\{e_i\}$ into $k$ disjoint groups $\mathcal{B}_i$ according to the values of $\{\alpha_i\}$
   $x_i \leftarrow \sum_{e_j \in \mathcal{B}_i} e_j + cj, i = 1, \ldots, k$
   **for** each eigenvector $u \in S_1$ **do**
      $x_h \leftarrow$ non-orthogonal summary vector with smallest $h$
      $u \leftarrow -u$ if $u^\top x_h < 0$
   **end for**
   **return** the canonized eigenvectors

---

## I.2  Further weakening Assumption 2 and Assumption 3

When reading Algorithm 2, some may find it not as strong as Algorithm 1. In Algorithm 1, we search over the summary vectors $x_i$ one by one, until we find a summary vector $x_h$ that is not

orthogonal to $\boldsymbol{u}$. However, in Algorithm 2, we just assumed that $\boldsymbol{x}_1$ is not orthogonal to $V$, $\boldsymbol{x}_2$ is not orthogonal to $\langle \boldsymbol{u}_1 \rangle^\perp$, $\boldsymbol{x}_3$ is not orthogonal to $\langle \boldsymbol{u}_1, \boldsymbol{u}_2 \rangle^\perp$, *etc.* We did not search for non-orthogonal summary vectors; we just assumed that they are. This observation is more obvious when we look at the special case of Algorithm 2 when $d = 1$. In this case, Assumption 2 is always satisfied, and Assumption 3 requires that $\boldsymbol{x}_1$ is not orthogonal to $\boldsymbol{u}$, which is stricter than Assumption 1 in our MAP-sign algorithm. This raises the question of whether we can strengthen Algorithm 2 such that it is as powerful as Algorithm 1 when taking $d = 1$.

This can be achieved by searching for non-orthogonal summary vectors at each step just as in Algorithm 1. Suppose we have obtained the summary vectors $\boldsymbol{x}_i, i = 1, \ldots, k$. First we search $\{\boldsymbol{x}_i\}$ for the first summary vector $\boldsymbol{x}_1^*$ that is not orthogonal to $V$, and choose $\boldsymbol{u}_1 \in V$ that maximizes $\boldsymbol{u}_1^\top \boldsymbol{x}_1^*$. Next we search $\{\boldsymbol{x}_i\}$ for the first summary vector $\boldsymbol{x}_2^*$ that is not orthogonal to $\langle \boldsymbol{u}_1 \rangle^\perp$, and choose $\boldsymbol{u}_2 \in \langle \boldsymbol{u}_1 \rangle^\perp$ that maximizes $\boldsymbol{u}_2^\top \boldsymbol{x}_2^*$. Then we search $\{\boldsymbol{x}_i\}$ for the first summary vector $\boldsymbol{x}_3^*$ that is not orthogonal to $\langle \boldsymbol{u}_1, \boldsymbol{u}_2 \rangle^\perp$, and choose $\boldsymbol{u}_3 \in \langle \boldsymbol{u}_1, \boldsymbol{u}_2 \rangle^\perp$ that maximizes $\boldsymbol{u}_3^\top \boldsymbol{x}_3^*$, and so on. The complete workflow is shown in Algorithm 7.

---

**Algorithm 7** A stronger algorithm for eliminating basis ambiguity

---

**Require:** Eigenvalue $\lambda$ with multiplicity $d > 1$
**Ensure:** Spectral embedding corresponding to $\lambda$
    Calculate the eigenvectors $\boldsymbol{U} \in \mathbb{R}^{n \times d}$ of $\lambda$ through eigendecomposition
    $\boldsymbol{P} \leftarrow \boldsymbol{U}\boldsymbol{U}^\top$
    $\alpha_i \leftarrow |\boldsymbol{P}\boldsymbol{e}_i|, 1 \le i \le n$
    $k \leftarrow$ the number of distinct values in $\{\alpha_i\}$
    Divide all basis vectors $\{\boldsymbol{e}_i\}$ into $k$ disjoint groups $\mathcal{B}_i$ according to the values of $\{\alpha_i\}$
    $\boldsymbol{x}_i \leftarrow \sum_{\boldsymbol{e}_j \in \mathcal{B}_i} \boldsymbol{e}_j + c\boldsymbol{j}, i = 1, \ldots, k$
    **for** $i = 1, 2, \ldots, d$ **do**
        $\boldsymbol{x}_i^* \leftarrow$ the first summary vector in $\{\boldsymbol{x}_i\}$ not perpendicular to $\langle \boldsymbol{u}_1, \ldots, \boldsymbol{u}_{i-1} \rangle^\perp$
        Choose $\boldsymbol{u}_i \in \langle \boldsymbol{u}_1, \ldots, \boldsymbol{u}_{i-1} \rangle^\perp, |\boldsymbol{u}_i| = 1$, *s.t.* $f(\boldsymbol{u}_i) = \boldsymbol{u}_i^\top \boldsymbol{x}_i^*$ is maximized
    **end for**
    **return** $\boldsymbol{U}_0 \coloneqq [\boldsymbol{u}_1, \ldots, \boldsymbol{u}_d]$

---

In order for Algorithm 7 to succeed, we require that such non-orthogonal summary vector $\boldsymbol{x}_i^*$ exists at each step, which is less restrictive than Assumption 2 and Assumption 3. Thus we obtain a stronger algorithm for eliminating basis ambiguity.

## J An alternative approach to dealing with sign ambiguity

In our design for algorithms that eliminate sign ambiguity, we find one type of eigenvectors especially difficult to deal with. That is, uncanonizable eigenvectors $\boldsymbol{u} \in \mathbb{R}^n$ such that there exists a permutation matrix $\boldsymbol{P} \in \mathbb{R}^{n \times n}$ satisfying $\boldsymbol{u} = -\boldsymbol{P}\boldsymbol{u}$. This means that $\boldsymbol{u}$ and $-\boldsymbol{u}$ only differ by a permutation. Since they are uncanonizable by Corollary 1, none of the solutions mentioned in Appendix G.1 can handle such eigenvectors. However, depending on the model architecture, such eigenvectors may not cause ambiguities at all. For example, consider the DeepSets-like architecture $\rho(\sum \phi(\boldsymbol{u}_i))$, where $\phi$ is permutation-invariant. Then since $\boldsymbol{u}$ and $-\boldsymbol{u}$ only differ by a permutation, they produce the same output when fed to a permutation-invariant network, thus no ambiguities exist. This shows we can delay the handling of uncanonizable eigenvectors to the training stage, though it may result in loss of expressive power.

On the other hand, for eigenvectors that are canonizable, we already know that Algorithm 1 canonizes them. Here we give an equivalent algorithm for removing sign ambiguity, shown in Algorithm 8.

**Theorem 14.** *Algorithm 8 uniquely decides the signs of canonizable eigenvectors and is permutation-equivariant.*

Theorem 14 is proved in Appendix K.9.

Algorithm 8 is well-motivated and better to understand than Algorithm 1 in some sense. Consider a naïve canonization of $\boldsymbol{u}$ where we choose the sign such that $\boldsymbol{u}$ has positive sum. This canonization algorithm is quite simple, but it cannot canonize all canonizable eigenvectors, such as the ones with

---

**Algorithm 8** An alternative approach to dealing with sign ambiguity

---

**Require:** Input graph $\mathcal{G} = (\mathbb{V}, \mathbb{E}, \boldsymbol{X})$
**Ensure:** Spectral embedding of $\mathcal{G}$
   Calculate the eigendecomposition $\hat{\boldsymbol{A}} = \boldsymbol{U}\boldsymbol{\Lambda}\boldsymbol{U}^\top$
   **for** each eigenvector $\boldsymbol{u} \in \mathbb{R}^n$ **do**
      $h \leftarrow$ the smallest positive odd integer such that $\sum_{i=1}^n u_i^h \neq 0$
      Substitute $\boldsymbol{u}$ with $-\boldsymbol{u}$ if $\sum_{i=1}^n u_i^h < 0$
   **end for**
   **return** $U$

---

zero sum. What Algorithm 8 does is to go on to look at the sum of the 3rd power of $\boldsymbol{u}$ and, if it is non-zero, choose the sign such that it is positive. If unfortunately it is zero, we go on to look at the sum of the 5th power and so on. It can be proved that there must exists a positive odd integer $h \leq n$ such that the sum of the $h$-th power of $\boldsymbol{u}$ is non-zero. Thus Algorithm 8 terminates within $\frac{n+1}{2}$ steps, successfully canonizing all canonizable eigenvectors.

Algorithm 8 offers an alternative approach to dealing with sign ambiguity in addition to Algorithm 1, though it cannot generalize to the basis ambiguity case. The time complexity of Algorithm 8 is $\mathcal{O}(n^2 \log n)$.

# K  Proofs

## K.1  Proof of Theorem 1

We first prove that $\boldsymbol{U}\boldsymbol{\Lambda}^{\frac{1}{2}}$ is a real-valued matrix.

**Lemma 1.** *Suppose $\hat{\boldsymbol{A}}$ is the normalized adjacency matrix of a graph $\mathcal{G}$, and $\hat{\boldsymbol{A}} = \boldsymbol{U}\boldsymbol{\Lambda}\boldsymbol{U}^\top$ is its spectral decomposition. Then, $\boldsymbol{U}\boldsymbol{\Lambda}^{\frac{1}{2}} \in \mathbb{R}^{n \times n}$.*

*Proof.* Let $\boldsymbol{\Lambda} = \mathrm{diag}(\boldsymbol{\lambda})$. It suffices to show that $\lambda_i \geq 0$ for $i = 1, 2, \ldots, n$.

Let $\mathcal{G} = (\mathbb{V}, \mathbb{E})$, where $\mathbb{V}$ is the vertex set and $\mathbb{E}$ is the edge set. For node $i$, we denote the degree of node $i$ by $d_i$. For any $\boldsymbol{x} \in \mathbb{R}^n$, we have

$$\boldsymbol{x}^\top \hat{\boldsymbol{A}} \boldsymbol{x} = \boldsymbol{x}^\top (\boldsymbol{I} + \tilde{\boldsymbol{A}}) \boldsymbol{x} = \sum_{i \in \mathbb{V}} x_i^2 + \sum_{(i,j) \in \mathbb{E}} \frac{2 x_i x_j}{\sqrt{d_i d_j}} = \sum_{(i,j) \in \mathbb{E}} \left( \frac{x_i}{\sqrt{d_i}} + \frac{x_j}{\sqrt{d_j}} \right)^2 \geq 0,$$

thus the Rayleigh quotient of $\hat{\boldsymbol{A}}$ is bounded by $\frac{\boldsymbol{x}^\top \hat{\boldsymbol{A}} \boldsymbol{x}}{\boldsymbol{x}^\top \boldsymbol{x}} \geq 0$. The Rayleigh quotient gives the lower bound of eigenvalues of $\hat{\boldsymbol{A}}$, therefore we have $\lambda_i \geq 0$, and this completes the proof. $\qquad\square$

Then we give the proof of Theorem 1.

*Proof.* We can rewrite $f$ such that it is a continuous set invariant function on the set consisting of the rows of its input:

$$f([\boldsymbol{X}, \hat{\boldsymbol{A}}]) = f([\boldsymbol{X}, \boldsymbol{U}\boldsymbol{\Lambda}\boldsymbol{U}^\top]) = f\left( \left[ \boldsymbol{X}, (\boldsymbol{U}\boldsymbol{\Lambda}^{\frac{1}{2}})(\boldsymbol{U}\boldsymbol{\Lambda}^{\frac{1}{2}})^\top \right] \right) = F([\boldsymbol{X}, \boldsymbol{U}\boldsymbol{\Lambda}^{\frac{1}{2}}]).$$

Using the permutation invariance property of $f$, we can verify that $F$ is set invariant by observing:

$$F([\boldsymbol{P}\boldsymbol{X}, \boldsymbol{P}\boldsymbol{U}\boldsymbol{\Lambda}^{\frac{1}{2}}]) = f\left( \left[ \boldsymbol{P}\boldsymbol{X}, (\boldsymbol{P}\boldsymbol{U}\boldsymbol{\Lambda}^{\frac{1}{2}})(\boldsymbol{P}\boldsymbol{U}\boldsymbol{\Lambda}^{\frac{1}{2}})^\top \right] \right)$$
$$= f([\boldsymbol{P}\boldsymbol{X}, \boldsymbol{P}\hat{\boldsymbol{A}}\boldsymbol{P}^\top]) = f([\boldsymbol{X}, \hat{\boldsymbol{A}}]) = F([\boldsymbol{X}, \boldsymbol{U}\boldsymbol{\Lambda}^{\frac{1}{2}}]).$$

Thus a universal network on sets can approximate $F$ to an arbitrary precision. $\qquad\square$

## K.2 Proof of Theorem 2

*Proof.* On the one hand, if $f(hx) = gf(x)$ holds for some $h \in H$ and $g \in G$, then we have $\mathcal{A}(f(hx)) = h\mathcal{A}(f(x))$ by the equivariance property of $\mathcal{A}$ and $\mathcal{A}(gf(x)) = \mathcal{A}(f(x))$ by the invariance property of $\mathcal{A}$. However, since $\mathcal{A}$ is a *mapping* and $f(hx) = gf(x)$, there must be $\mathcal{A}(f(hx)) = \mathcal{A}(gf(x))$ and thus $\mathcal{A}(f(x)) = h\mathcal{A}(f(x))$. Now we have $x \neq hx$ and $\mathcal{A}(f(x)) = h\mathcal{A}(f(x))$, contradicting the universality property of $\mathcal{A}$.

On the other hand, if there does not exist $h \in H$ and $g \in G$ such that $x \neq hx$ and $f(hx) = gf(x)$, we can construct a canonization of $x$ as follows. First arbitrarily choose $y_0$ such that $f(x) = y_0$, and let $\mathcal{A}(f(x)) = y_0$. For any $h \in H$ such that $x \neq hx$, by the equivariance of $f$ we know that $f(hx) = hy_0$, thus we let $\mathcal{A}(f(hx)) = hy_0$. Since $f(hx) \neq gf(x)$ for any $g$, we know that $y_0 \neq hy_0$, thus the universality property of $\mathcal{A}$ holds. $\mathcal{A}$ is also invariant, since $\mathcal{A}(f(x))$ is uniquely determined; and equivariant, since $\mathcal{A}(f(hx)) = hy_0 = h\mathcal{A}(f(x))$ for any $h$. We can repeat this process for all equivalence classes in $\mathcal{X}$ and obtain an invariant, equivariant and universal canonization for all inputs. $\square$

## K.3 Proof of Corollary 1

*Proof.* Under sign ambiguity, we have $H = S_n$ and $G = \{+1, -1\}$. By Theorem 2, $\boldsymbol{u}$ is canonizable if and only if there does not exist a permutation $\sigma$ such that $\boldsymbol{u} \neq \sigma(\boldsymbol{u})$ and $\boldsymbol{u} = \pm\sigma(\boldsymbol{u})$. This is equivalent to say that there does not exist a permutation matrix $\boldsymbol{P} \in \mathbb{R}^{n \times n}$ such that $\boldsymbol{u} = -\boldsymbol{P}\boldsymbol{u}$. $\square$

## K.4 Proof of Corollary 2

*Proof.* Under basis ambiguity, we have $H = S_n$ and $G = O(d)$. By Theorem 2, $\boldsymbol{U}$ is canonizable if and only if there does not exist a permutation $\sigma$ and an orthonormal matrix $\boldsymbol{Q} \in O(d)$ such that $\boldsymbol{U} \neq \sigma(\boldsymbol{U})$ and $\boldsymbol{U} = \sigma(\boldsymbol{U})\boldsymbol{Q}$. This is equivalent to say that there does not exist a permutation matrix $\boldsymbol{P} \in \mathbb{R}^{n \times n}$ such that $\boldsymbol{U} \neq \boldsymbol{P}\boldsymbol{U}$ and $\boldsymbol{U}$ and $\boldsymbol{P}\boldsymbol{U}$ span the same $d$-dimensional subspace $V \subseteq \mathbb{R}^n$. Note here we used a lemma from Lim et al. [29]: for any orthonormal bases $\boldsymbol{V}$ and $\boldsymbol{W}$ of the same subspace, there exists an orthogonal $\boldsymbol{Q} \in O(d)$ such that $\boldsymbol{V}\boldsymbol{Q} = \boldsymbol{W}$. $\square$

## K.5 Proof of Theorem 3

*Proof.* Without loss of generality, we can always assume that the angles $\{\alpha_i\}$ are *sorted* (if they are not, we can simply rearrange $\boldsymbol{e}_i$ to make them sorted and proceed in the same way):

$$|\boldsymbol{P}\boldsymbol{e}_1| = \cdots = |\boldsymbol{P}\boldsymbol{e}_{n_1}| > |\boldsymbol{P}\boldsymbol{e}_{n_1+1}| = \cdots = |\boldsymbol{P}\boldsymbol{e}_{n_1+n_2}| > \cdots$$
$$> |\boldsymbol{P}\boldsymbol{e}_{n_1+\cdots+n_{k-1}+1}| = \cdots = |\boldsymbol{P}\boldsymbol{e}_{n_1+\cdots+n_k}|,$$

where $k$ is the number of distinct lengths of $\boldsymbol{P}\boldsymbol{e}_i$, $\sum_{i=1}^{k} n_i = n$. Here we divide $\boldsymbol{e}_i$ into $k$ groups according to the angles between them and the eigenspace, with each group sharing the same $|\boldsymbol{P}\boldsymbol{e}_i|$. Define $\boldsymbol{j} = (1, 1, \ldots, 1) \in \mathbb{R}^n$, then $\boldsymbol{x}_i$ can be expressed as

$$\boldsymbol{x}_i = \boldsymbol{e}_{n_1+\cdots+n_{i-1}+1} + \cdots + \boldsymbol{e}_{n_1+\cdots+n_i} + c\boldsymbol{j}, \ 1 \leq i \leq k.$$

The sign of $\boldsymbol{u}_0$ is selected based on the sign of $\boldsymbol{u}^\top \boldsymbol{x}_h(\neq 0)$, which is sign-equivariant. No matter what the sign of $\boldsymbol{u}$ is, we will always choose the one that maximizes $\boldsymbol{u}^\top \boldsymbol{x}_h$, thus $\boldsymbol{u}_0$ is sign-invariant.

Suppose the entries of the input eigenvector $\boldsymbol{u}$ is permutated by $\sigma \in S_n$, where $S_n$ is the permutation group of order $n$. Then we have $u'_i = u_{\sigma(i)}(1 \leq i \leq n)$ and

$$|\boldsymbol{u}'\boldsymbol{u}'^\top \boldsymbol{e}_{\sigma(1)}| = \cdots = |\boldsymbol{u}'\boldsymbol{u}'^\top \boldsymbol{e}_{\sigma(n_1)}| > |\boldsymbol{u}'\boldsymbol{u}'^\top \boldsymbol{e}_{\sigma(n_1+1)}| = \cdots = |\boldsymbol{u}'\boldsymbol{u}'^\top \boldsymbol{e}_{\sigma(n_1+n_2)}| > \cdots$$
$$> |\boldsymbol{u}'\boldsymbol{u}'^\top \boldsymbol{e}_{\sigma(n_1+\cdots+n_{k-1}+1)}| = \cdots = |\boldsymbol{u}'\boldsymbol{u}'^\top \boldsymbol{e}_{\sigma(n_1+\cdots+n_k)}|,$$

thus $(x'_i)_i = (x_i)_{\sigma(i)}$, *i.e.*, the vectors $\boldsymbol{x}_i(1 \leq i \leq k)$ are permutation-equivariant. Let $\boldsymbol{x}'_h$ be defined as in Section 3.3.1 (the number $h$ is permutation-invariant because each $\boldsymbol{u}'^\top \boldsymbol{x}'_i$ is permutation-invariant). The sign of $\boldsymbol{u}'_0$ is determined by the sign of the dot product of $\boldsymbol{u}'$ and $\boldsymbol{x}'_h$, both of which are permutation-equivariant. This indicates that the sign of $\boldsymbol{u}'^\top \boldsymbol{x}'_h$ (and thus the sign of $\boldsymbol{u}'_0$) is

permutation-invariant (*i.e.*, unique). Since $u_0' = u'$ or $u_0' = -u'$, $u_0'$ is permutation-equivariant as well.

If there exists a permutation $\sigma$ (acting on entries of $u$) such that $u \neq \sigma(u)$ but they have the same canonical form, then either $u = +\sigma(u)$ or $u = -\sigma(u)$. The former is impossible since we already assumed they are not equal, and the latter violates Assumption 1, leading to a contradiction. Thus the canonization of $u$ is universal. $\qquad\square$

### K.6 Proof of Theorem 4

*Proof.* Without loss of generality, we can always assume that the angles $\{\alpha_i\}$ are *sorted* (if they are not, we can simply rearrange $e_i$ to make them sorted and proceed in the same way):

$$|Pe_1| = \cdots = |Pe_{n_1}| > |Pe_{n_1+1}| = \cdots = |Pe_{n_1+n_2}| > \cdots$$
$$> |Pe_{n_1+\cdots+n_{k-1}+1}| = \cdots = |Pe_{n_1+\cdots+n_k}|,$$

where $k$ is the number of distinct lengths of $Pe_i$, $\sum_{i=1}^{k} n_i = n$. Here we divide $e_i$ into $k$ groups according to the angles between them and the eigenspace, with each group sharing the same $|Pe_i|$. Define $j = (1, 1, \ldots, 1) \in \mathbb{R}^n$, then $x_i$ can be expressed as

$$x_i = e_{n_1+\cdots+n_{i-1}+1} + \cdots + e_{n_1+\cdots+n_i} + cj, \ 1 \leq i \leq k.$$

Notice that $|Pe_i| = |uu^\top e_i| = u_i|u|$, thus

$$|u_1| = \cdots = |u_{n_1}| > \cdots > |u_{n_1+\cdots+n_{k-1}+1}| = \cdots = |u_{n_1+\cdots+n_k}|.$$

Suppose $u$ violates Assumption 1. Then for any $1 \leq j \leq k$, we have

$$u_{n_1+\cdots+n_{j-1}+1} + \cdots + u_{n_1+\cdots+n_j} = 0.$$

Let $\tilde{u}_j := (u_{n_1+\cdots+n_{j-1}+1}, \ldots, u_{n_1+\cdots+n_j})^\top$. The above equations show that (1) the absolute value of the entries of $\tilde{u}_j$ are all equal; (2) the sum of the entries of $\tilde{u}_j$ is 0. Thus for any entry of $\tilde{u}_j$, either it is 0 or its positive and negative value appears in pairs. In conclusion, $+\tilde{u}_j$ and $-\tilde{u}_j$ are equal up to a permutation for all $j$, thus $+u$ and $-u$ are equal up to a permutation. By Theorem 1, it is not canonizable.

On the other hand, suppose $u$ satisfies Assumption 1. Then there exists $1 \leq j \leq k$ such that

$$u_{n_1+\cdots+n_{j-1}+1} + \cdots + u_{n_1+\cdots+n_j} \neq 0.$$

Let $\tilde{u}_j := (u_{n_1+\cdots+n_{j-1}+1}, \ldots, u_{n_1+\cdots+n_j})^\top$. $\tilde{u}_j$ contains all entries of $u$ with absolute value $|u_{n_1+\cdots+n_j}|$. Since the sum of $\tilde{u}_j$ is non-zero, the numbers of positive and negative entries in $\tilde{u}_j$ are different. No matter how we permute $+u$ and $-u$, their corresponding entries in $\tilde{u}_j$ will not align. Thus $+u$ and $-u$ are not equal up to any permutation. By Theorem 1, it is canonizable. $\qquad\square$

### K.7 Proof of Theorem 5

We first point out that the orthogonal projector $P = UU^\top$ is invariant to the choice of basis.

**Lemma 2.** *Let $U = [u_1, \ldots, u_d] \in \mathbb{R}^{n \times d}$ and $V = [v_1, \ldots, v_d] \in \mathbb{R}^{n \times d}$ be two sets of orthonormal vectors that span the same $d$-dimensional subspace $V \subseteq \mathbb{R}^n$. Then $UU^\top = VV^\top$.*

*Proof.* We have $U^\top U = V^\top V = I$ by the definition of $U$ and $V$. Let $U = VQ$ where $Q$ is an invertible matrix, then

$$UU^\top = U(U^\top U)^{-1}U^\top$$
$$= VQ(Q^\top V^\top VQ)^{-1}Q^\top V^\top$$
$$= VQQ^{-1}(V^\top V)^{-1}(Q^\top)^{-1}Q^\top V^\top = VV^\top.$$

$\qquad\square$

Lemma 2 shows that $\boldsymbol{P}\boldsymbol{e}_i$ (and thus $\boldsymbol{x}_i$) is invariant to the choice of basis in $V$. If we permute $\boldsymbol{e}_1, \ldots, \boldsymbol{e}_n$ by $\sigma \in S_n$, then the sorted sequence of $|\boldsymbol{P}\boldsymbol{e}_i|$ is also permuted by $\sigma$. Thus the elements of each $\boldsymbol{x}_i$ is permuted by $\sigma$ as well. This shows the choice of $\boldsymbol{x}_i$ is *permutation-equivariant*.

Then we prove Theorem 5.

*Proof.* Without loss of generality, we can always assume that the angles $\{\alpha_i\}$ are *sorted* (if they are not, we can simply rearrange $\boldsymbol{e}_i$ to make them sorted and proceed in the same way):

$$|\boldsymbol{P}\boldsymbol{e}_1| = \cdots = |\boldsymbol{P}\boldsymbol{e}_{n_1}| > |\boldsymbol{P}\boldsymbol{e}_{n_1+1}| = \cdots = |\boldsymbol{P}\boldsymbol{e}_{n_1+n_2}| > \cdots$$
$$> |\boldsymbol{P}\boldsymbol{e}_{n_1+\cdots+n_{k-1}+1}| = \cdots = |\boldsymbol{P}\boldsymbol{e}_{n_1+\cdots+n_k}|,$$

where $k$ is the number of distinct lengths of $\boldsymbol{P}\boldsymbol{e}_i$, $\sum_{i=1}^{k} n_i = n$. Here we divide $\boldsymbol{e}_i$ into $k$ groups according to the angles between them and the eigenspace, with each group sharing the same $|\boldsymbol{P}\boldsymbol{e}_i|$. Define $\boldsymbol{j} = (1, 1, \ldots, 1) \in \mathbb{R}^n$, then $\boldsymbol{x}_i$ can be expressed as

$$\boldsymbol{x}_i = \boldsymbol{e}_{n_1+\cdots+n_{i-1}+1} + \cdots + \boldsymbol{e}_{n_1+\cdots+n_i} + c\boldsymbol{j}, \ 1 \le i \le k.$$

We have already shown the existence of the maximum value of $f(\boldsymbol{u})$. To show that basis-invariance of $\boldsymbol{U}_0$, it suffices to show the uniqueness of the maximum point of $f(\boldsymbol{u})$. Thus no matter what basis of $\boldsymbol{U}$ is, Algorithm 2 always yields the same output.

Notice that $f(-\boldsymbol{u}) = -f(\boldsymbol{u})$, thus either the maximum value of $f(\boldsymbol{u})$ is positive, or $f(\boldsymbol{u}) = \boldsymbol{u}^\top \boldsymbol{x}_i \equiv 0$. However, $f(\boldsymbol{u}) \equiv 0$ implies that $\boldsymbol{x}_i$ is perpendicular to $\langle \boldsymbol{u}_1, \ldots, \boldsymbol{u}_{i-1} \rangle^\perp$, violating Assumption 3. Thus we conclude the maximum value of $f(\boldsymbol{u})$ is positive.

Suppose there exists $\boldsymbol{u}' \ne \boldsymbol{u}''$ such that $f(\boldsymbol{u}') = f(\boldsymbol{u}'')$ takes maximum value. Consider the vector $\alpha \boldsymbol{u}' + \alpha \boldsymbol{u}''$ where $\alpha = \sqrt{\frac{1}{2(1+\boldsymbol{u}'^\top \boldsymbol{u}'')}} > \frac{1}{2}$. Obviously, $\alpha \boldsymbol{u}' + \alpha \boldsymbol{u}'' \in \langle \boldsymbol{u}_1, \ldots, \boldsymbol{u}_{i-1} \rangle^\perp$, $|\alpha \boldsymbol{u}' + \alpha \boldsymbol{u}''| = 1$, and $f(\alpha \boldsymbol{u}' + \alpha \boldsymbol{u}'') = 2\alpha \boldsymbol{u}'^\top \boldsymbol{x}_i > \boldsymbol{u}'^\top \boldsymbol{x}_i$. This leads to a contradiction. Therefore, the choice of $\boldsymbol{u}_i$ is unique.

The permutation-equivariance of Algorithm 2 can be proved by observing that each step of Algorithm 2 is permutation-equivariant. Since each $\boldsymbol{x}_i$ is permutation-equivariant, its eigenprojection on the subspace $V$ (and thus $\boldsymbol{U}_0$) is also permutation-equivariant.

If there exists a permutation $\sigma$ (acting on rows of $\boldsymbol{U}$) such that $\boldsymbol{U} \ne \sigma(\boldsymbol{U})$ but they have the same canonical form, then $\boldsymbol{U}$ and $\sigma(\boldsymbol{U})$ spans the same subspace. On the one hand, $\boldsymbol{U} \ne \sigma(\boldsymbol{U})$ means that at least one of the eigenvectors in $\boldsymbol{U}$ is not $\sigma$-invariant. On the other hand, since all $\boldsymbol{u}_1, \ldots, \boldsymbol{u}_d$ are permutation-equivariant but unchanged after $\sigma$, they are all $\sigma$-invariant. This leads to a contradiction, since it is impossible to have a non-$\sigma$-invariant eigenvector in a $\sigma$-invariant eigenspace. $\square$

### K.8 Proof of Theorem 7

We first prove the following lemmas.

**Lemma 3.** *Let $\mathbf{R} \in \mathbb{R}^{n \times n}$ be a random matrix, and each entry of $\mathbf{R}$ is sampled independently from the standard Gaussian distribution $N(0, 1)$. Then with probability 1, $\mathbf{R}$ has full rank.*

*Proof.* We denote the first column of $\mathbf{R}$ by $\mathbf{R}_{:,1}$. It is linearly independent because $\mathbf{R}_{:,1} = \boldsymbol{0}$ with probability 1. Then we view $\mathbf{R}_{:,1}$ as fixed, and consider the second column $\mathbf{R}_{:,2}$. The probability that $\mathbf{R}_{:,2}$ falls into the span of $\mathbf{R}_{:,1}$ is 0, thus with probability 1, $\mathbf{R}_{:,1}$ and $\mathbf{R}_{:,2}$ are linearly independent.

Generally, let us consider the $k$-th column $\mathbf{R}_{:,k}$. The first $k-1$ columns of $\mathbf{R}$ forms a subspace in $\mathbb{R}^n$ whose Lebesgue measure is 0. Thus $\mathbf{R}_{:,k}$ falls into this subspace with probability 0. By inference, we have all the columns of $\mathbf{R}$ are linearly independent with probability 1, *i.e.*, $P(\text{rank}(\mathbf{R}) = n) = 1$. $\square$

**Lemma 4.** *Let $\boldsymbol{A} \in \mathbb{R}^{s \times n}$, $\boldsymbol{B} \in \mathbb{R}^{s \times m}$ be two matrices. Then the equation $\boldsymbol{A}\boldsymbol{X} = \boldsymbol{B}$ has a solution iff. $\text{rank}(\boldsymbol{A}) = \text{rank}([\boldsymbol{A}, \boldsymbol{B}])$.*

*Proof.* First we prove the necessity. Suppose $\boldsymbol{A}\boldsymbol{X} = \boldsymbol{B}$ has a solution. Then,

$$[\boldsymbol{A}_{:,1}, \boldsymbol{A}_{:,2}, \ldots, \boldsymbol{A}_{:,n}]\boldsymbol{X}_{:,i} = \boldsymbol{B}_{:,i},$$

where $M_{:,i}$ denotes the $i$-th column of matrix $M$. This means each column of $B$ can be expressed as a linear combination of the columns of $A$, and therefore each column of $[A, B]$ can be expressed as a linear combination of the columns of $A$.

On the other hand, it is obvious that each column of $A$ can be expressed as a linear combination of the columns of $[A, B]$. Thus we have $\text{rank}(A) = \text{rank}([A, B])$.

Then we prove the sufficiency. Since $\text{rank}(A) = \text{rank}([A, B])$, and each column of $A$ can be expressed as a linear combination of columns of $[A, B]$, we have the columns of $A$ and the columns of $[A, B]$ are equivalent. Therefore, each column of $B$ can be expressed as a linear combination of columns of $A$, i.e., $Ax = B_{:,i}$ has a solution for $i = 1, \ldots, m$. Thus the equation $AX = B$ has a solution. $\square$

Then we give the proof of Theorem 7.

*Proof.* For any prediction $Z \in \mathbb{R}^{n \times d'}$, we wish to prove that with probability 1, there exists parameters of a linear GNN with RNI $W \in \mathbb{R}^{d \times d'}$ such that

$$[X, \mathbf{R}]W = Z. \tag{7}$$

By Lemma 4, the necessary and sufficient condition that Equation 7 has a solution $W$ is $\text{rank}([X, \mathbf{R}]) = \text{rank}([X, \mathbf{R}, Z])$.

By Lemma 3, with probability 1, $\text{rank}(\mathbf{R}) = n$, therefore $\text{rank}([X, \mathbf{R}]) = \text{rank}([X, \mathbf{R}, Z]) = n$.

Thus, in conclusion, with probability 1, there exists parameters of a linear GNN with RNI $W \in \mathbb{R}^{d \times d'}$ such that the GNN produces $Z$.

We can also prove linear GNNs' equivariance by observing that for any permutation matrix $P \in \mathbb{R}^{n \times n}$,

$$[PX, \mathbf{R}]W = P[X, \mathbf{R}]W = PZ,$$

where we used $P\mathbf{R} = \mathbf{R}$ because each entry of $P\mathbf{R}$ is also sampled from the standard Gaussian matrix $N(0, 1)$. $\square$

### K.9 Proof of Theorem 14

The following lemmas are used in our proof.

**Lemma 5** (Newton's Identities). *Let $x_1, x_2, \ldots, x_n$ be variables, denote for $k \geq 1$ by $P_k$ the $k$-th power sum:*

$$P_k = \sum_{i=1}^{n} x_i^k = x_1^k + \cdots + x_n^k,$$

*and for $k \geq 0$ denote by $e_k$ the elementary symmetric polynomial. Then we have*

$$P_k = (-1)^{k-1} k e_k + \sum_{i=1}^{k-1} (-1)^{k-1+i} e_{k-i} P_i,$$

*for all $n \geq 1$ and $n \geq k \geq 1$.*

**Lemma 6** (Vieta's Formulas). *Let $f(x) = a_n x^n + a_{n-1} x^{n-1} + \cdots + a_1 x + a_0$ be a polynomial of degree $n$, $e_k$ be the elementary symmetric polynomial. Then we have*

$$e_1 = -\frac{a_{n-1}}{a_n}, \quad e_2 = \frac{a_{n-2}}{a_n}, \quad \ldots, \quad e_n = (-1)^n \frac{a_0}{a_n}.$$

We first show that $h$ exists.

**Lemma 7.** *Let $u \in \mathbb{R}^n$. Assume that there does not exists a permutation matrix $P \in \mathbb{R}^{n \times n}$ such that $u = -Pu$. Then there exists a positive odd integer $h \leq n$ such that $\sum_{i=1}^{n} u_i^h \neq 0$.*

*Proof.* Suppose the opposite holds, *i.e.*, $P_i = 0$ for all odd $1 \leq i \leq n$, where $P_i$ is the $i$-th power sum of entries of $\boldsymbol{u}$. Let $u_1, u_2, \ldots, u_n$ be the roots of the polynomial $f(x) = a_n x^n + a_{n-1} x^{n-1} + \cdots + a_1 x + a_0$.

If $n$ is even, by Lemma 5,

$$
\begin{aligned}
0 &= P_1 = e_1, \\
0 &= P_3 = e_1 P_2 - e_2 P_1 + 3e_3, \\
0 &= P_5 = e_1 P_4 - e_2 P_3 + e_3 P_2 - e_4 P_1 + 5e_5, \\
&\vdots \\
0 &= P_{n-1} = e_1 P_{n-2} - e_2 P_{n-3} + \cdots + (n-1)e_{n-1},
\end{aligned}
$$

which gives us $e_1 = e_3 = \cdots = e_{n-1} = 0$. Then, by Lemma 6, we have $a_{n-1} = a_{n-3} = \cdots = a_1 = 0$. This indicates that $f(x) = g(x^2)$ for some polynomial $g(x)$. Similarly, if $n$ is odd, by Lemma 5,

$$
\begin{aligned}
0 &= P_1 = e_1, \\
0 &= P_3 = e_1 P_2 - e_2 P_1 + 3e_3, \\
0 &= P_5 = e_1 P_4 - e_2 P_3 + e_3 P_2 - e_4 P_1 + 5e_5, \\
&\vdots \\
0 &= P_n = e_1 P_{n-1} - e_2 P_{n-2} + \cdots + ne_n,
\end{aligned}
$$

which gives us $e_1 = e_3 = \cdots = e_n = 0$. Then, by Lemma 6, we have $a_{n-1} = a_{n-3} = \cdots = a_0 = 0$. This indicates that $f(x) = xg(x^2)$ for some polynomial $g(x)$. Either way, all $n$ roots of $f(x)$ are symmetric with respect to the $y$-axis. Then there must exist a permutation matrix such that $\boldsymbol{u} = -\boldsymbol{P}\boldsymbol{u}$, leading to a contradiction. Thus Lemma 7 holds. □

Then we prove Theorem 14.

*Proof.* By Lemma 7, we have shown the existence of $h$. Since flipping the sign of $\boldsymbol{u}$ also flips the sign of $\sum_{i=1}^n u_i^h$ (because $h$ is odd), Algorithm 8 uniquely decides the sign of $\boldsymbol{u}$. Since the algorithm outputs either $\boldsymbol{u}$ or $-\boldsymbol{u}$, it is also permutation-equivariant. □

### K.10 Proof of Theorem 15

We prove that under mild conditions, the loss of expressive power induced by truncating RSE can be upper bounded, as shown in the following theorem.

**Theorem 15.** *Let $\Omega \subset \mathbb{R}^{n \times d} \times \mathbb{R}^{n \times n}$ be a compact set of graphs, $[\boldsymbol{X}, \hat{\boldsymbol{A}}] \in \Omega$. Let* NN *be a universal neural network on sets. Given an invariant graph function $f$ defined over $\Omega$ that can be $\varepsilon$-approximated by an $L_p$-Lipschitz continuous function and arbitrary $\varepsilon > 0$, for any integer $0 < k \leq n$, there exist parameters of* NN *such that for all graphs $[\boldsymbol{X}, \hat{\boldsymbol{A}}] \in \Omega$,*

$$
\left| f([\boldsymbol{X}, \hat{\boldsymbol{A}}]) - \mathrm{NN}([\boldsymbol{X}, (\boldsymbol{U}\boldsymbol{\Lambda}^{\frac{1}{2}})_{:,-k:}, \boldsymbol{0}]) \right| < \sqrt{n-k} L_p \lambda_{n-k} + \varepsilon.
$$

*Here the $L_p$-Lipschitz continuity of $f$ is defined using the Frobenius norm on the input domain, $0 \leq \lambda_1 \leq \cdots \leq \lambda_n \leq 2$ are the eigenvalues of $\hat{\boldsymbol{A}}$, $\boldsymbol{0} \in \mathbb{R}^{n \times (n-k)}$.*

We can see from Theorem 15 that the upper bound of the loss of expressive power decreases when $k$ increases, and when $k = n$, the network becomes universal. We give its proof as follows.

By Lemma 1, we know that $\lambda_i \geq 0$ for $i = 1, 2, \ldots, n$. Next we prove that $\lambda_i \leq 2$.

**Lemma 8.** *Suppose $\hat{\boldsymbol{A}}$ is the normalized adjacency matrix of a graph $\mathcal{G}$, and $\lambda_1 < \cdots < \lambda_n$ are its eigenvalues. Then $\lambda_i \leq 2$, for $i = 1, 2, \ldots, n$.*

*Proof.* In the proof of Lemma 1, we proved $\boldsymbol{x}^\top (\boldsymbol{I} + \tilde{\boldsymbol{A}})\boldsymbol{x} \geq 0$. Similarly, we have

$$
\boldsymbol{x}^\top (\boldsymbol{I} - \tilde{\boldsymbol{A}})\boldsymbol{x} = \sum_{(i,j) \in \mathbb{E}} \left( \frac{x_i}{\sqrt{d_i}} - \frac{x_j}{\sqrt{d_j}} \right)^2 \geq 0.
$$

Thus,
$$\boldsymbol{x}^\top \hat{\boldsymbol{A}} \boldsymbol{x} = \boldsymbol{x}^\top (-\boldsymbol{I} + \tilde{\boldsymbol{A}})\boldsymbol{x} + 2\boldsymbol{x}^\top \boldsymbol{x} \le 2\boldsymbol{x}^\top \boldsymbol{x}.$$

This shows that the Rayleigh quotient is bounded by $\frac{\boldsymbol{x}^\top \hat{\boldsymbol{A}} \boldsymbol{x}}{\boldsymbol{x}^\top \boldsymbol{x}} \le 2$, therefore $\lambda_i \le 2$. $\qquad\square$

Then we give the proof of Theorem 15.

*Proof.* Let $0 \le \lambda_1 \le \cdots \le \lambda_n \le 2$ be the eigenvalues of $\hat{\boldsymbol{A}}$ and $\boldsymbol{u}_1, \ldots, \boldsymbol{u}_n$ be the corresponding eigenvectors. Then
$$\hat{\boldsymbol{A}} = \lambda_1 \boldsymbol{u}_1 \boldsymbol{u}_1^\top + \cdots + \lambda_n \boldsymbol{u}_n \boldsymbol{u}_n^\top.$$
We also define
$$\hat{\boldsymbol{A}}' \coloneqq \lambda_{n-k+1} \boldsymbol{u}_{n-k+1} \boldsymbol{u}_{n-k+1}^\top + \cdots + \lambda_n \boldsymbol{u}_n \boldsymbol{u}_n^\top.$$

By Theorem 1 and the assumptions in our theorem, we know that there exists a permutation-invariant network on sets such that
$$\left| F([\boldsymbol{X}, (\boldsymbol{U}\boldsymbol{\Lambda}^{\frac{1}{2}})_{:,-k:}, \boldsymbol{0}]) - \mathrm{NN}([\boldsymbol{X}, (\boldsymbol{U}\boldsymbol{\Lambda}^{\frac{1}{2}})_{:,-k:}, \boldsymbol{0}]) \right| < \frac{\varepsilon}{2}.$$

Since $f$ can be approximated by an $L_p$-Lipschitz continuous function, we have
$$\begin{aligned}
\left| f([\boldsymbol{X}, \hat{\boldsymbol{A}}]) - F([\boldsymbol{X}, (\boldsymbol{U}\boldsymbol{\Lambda}^{\frac{1}{2}})_{:,-k:}, \boldsymbol{0}]) \right| &= \left| f([\boldsymbol{X}, \hat{\boldsymbol{A}}]) - f([\boldsymbol{X}, \hat{\boldsymbol{A}}']) \right| \\
&\le L_p \left\| [\boldsymbol{X}, \hat{\boldsymbol{A}}] - [\boldsymbol{X}, \hat{\boldsymbol{A}}'] \right\|_\mathrm{F} + \frac{\varepsilon}{2} \\
&= L_p \left\| [\boldsymbol{0}, \lambda_1 \boldsymbol{u}_1 \boldsymbol{u}_1^\top + \cdots + \lambda_{n-k} \boldsymbol{u}_{n-k} \boldsymbol{u}_{n-k}^\top] \right\|_\mathrm{F} + \frac{\varepsilon}{2} \\
&= L_p \sqrt{\lambda_1^2 + \cdots + \lambda_{n-k}^2} + \frac{\varepsilon}{2} \\
&\le \sqrt{n-k} L_p \lambda_{n-k} + \frac{\varepsilon}{2}.
\end{aligned}$$

Combining the two inequalities above gives us
$$\left| f([\boldsymbol{X}, \hat{\boldsymbol{A}}]) - \mathrm{NN}([\boldsymbol{X}, (\boldsymbol{U}\boldsymbol{\Lambda}^{\frac{1}{2}})_{:,-k:}, \boldsymbol{0}]) \right| < \sqrt{n-k} L_p \lambda_{n-k} + \varepsilon.$$
$\square$

## L  Dataset details

**ZINC** (MIT License) consists of 12K molecular graphs from the ZINC database of commercially available chemical compounds. These molecular graphs are between 9 and 37 nodes large. Each node represents a heavy atom (28 possible atom types) and each edge represents a bond (3 possible types). The task is to regress constrained solubility (logP) of the molecule. The dataset comes with a predefined 10K/1K/1K train/validation/test split.

**OGBG-MOLTOX21 and OGBG-MOLPCBA** (MIT License) are molecular property prediction datasets adopted by OGB from MoleculeNet. These datasets use a common node (atom) and edge (bond) featurization that represent chemophysical properties. OGBG-MOLTOX21 is a multi-mask binary graph classification dataset where a qualitative (active/inactive) binary label is predicted against 12 different toxicity measurements for each molecular graph. OGBG-MOLPCBA is also a multi-task binary graph classification dataset from OGB where an active/inactive binary label is predicted for 128 bioassays.

Details of the three datasets are summarized in Table 16.

## M  Hyperparameter settings

### M.1  Real-world tasks

We evaluate the proposed MAP on three real-world datasets: ZINC, OGBG-MOLTOX21 and OGBG-MOLPCBA, on a server with 6 NVIDIA 3080 Ti GPUs and 2 NVIDIA 1080 Ti GPUs. We consider

Table 16: Details of the datasets.

| Dataset | ZINC | ogbg-moltox21 | ogbg-molpcba |
|---|---|---|---|
| #Graphs | 12000 | 7831 | 437929 |
| Avg #Nodes | 23.2 | 18.6 | 26.0 |
| Avg #Edges | 24.9 | 19.3 | 28.1 |
| Task Type | Regression | Binary Classification | Binary Classification |
| Metric | MAE | ROC-AUC | AP |

4 GNN architectures: GatedGCN, PNA, SAN and GraphiT, with 4 different positional encodings: no PE, LapPE with random Sign, SignNet and MAP. We follow the same settings as Dwivedi et al. [18] for models with no PE or LapPE, and same settings as Lim et al. [29] for models with SignNet or MAP. All baseline scores reported in Table 3, 4 & 5 are taken from the original papers. As shown in Figure 7, for models with no PE or LapPE, the input features are directly fed into the base model; for models with SignNet, the eigenvectors are first processed by SignNet and then concatenated with the original node features as input to the base model; for models with MAP, the PEs are first processed by a normal GNN and then concatenated with the original node features as input to the base model. These settings align with the original papers.

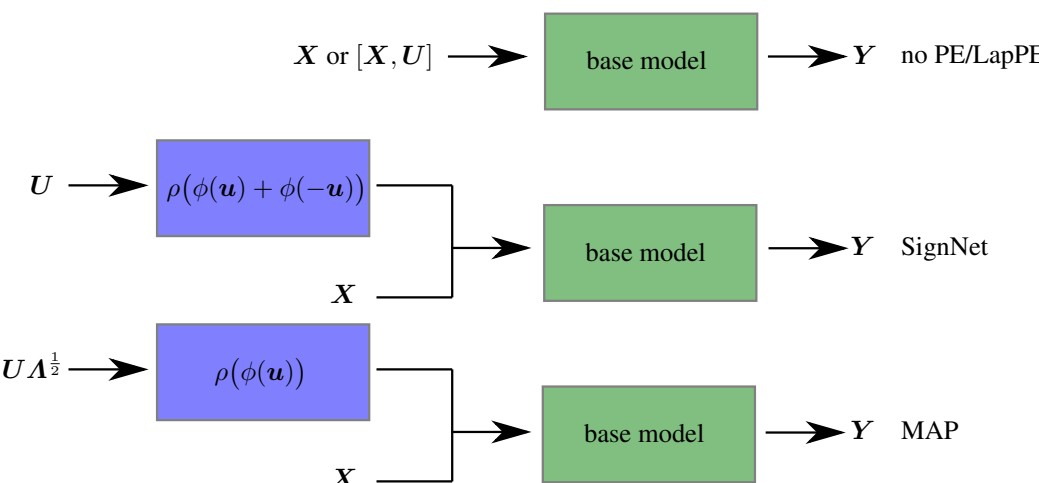

Figure 7: Our experiment settings with different PEs.

The main hyperparameters in our experiments are listed as follows.

- $k$: the number of eigenvectors used in the PE.
- $L_1$: the number of layers of the base model.
- $h_1$: the hidden dimension of the base model.
- $h_2$: the output dimension of the base model.
- $\lambda$: the initial learning rate.
- $t$: the patience of the learning rate schedular.
- $r$: the factor of the learning rate schedular.
- $\lambda_{\min}$: the minimum learning rate of the learning rate schedular.
- $L_2$: the number of layers of SignNet or the normal GNN (when using MAP as PE).
- $h_3$: the hidden dimension of SignNet or the normal GNN (when using MAP as PE).

The values of these hyperparameters in our experiments are listed in Table 17.

Table 17: Hyperparameter details for experiments on real-world datasets.

| | Model | PE | $k$ | $L_1$ | $h_1$ | $h_2$ | $\lambda$ | $t$ | $r$ | $\lambda_{\min}$ | $L_2$ | $h_3$ |
|---|---|---|---|---|---|---|---|---|---|---|---|---|
| ZINC | GatedGCN | None | 0 | 16 | 78 | 78 | 0.001 | 25 | 0.5 | 1e-6 | - | - |
| | GatedGCN | LapPE + RS | 8 | 16 | 78 | 78 | 0.001 | 25 | 0.5 | 1e-6 | - | - |
| | GatedGCN | SignNet | 8 | 16 | 67 | 67 | 0.001 | 25 | 0.5 | 1e-6 | 8 | 67 |
| | GatedGCN | MAP | 8 | 16 | 69 | 67 | 0.001 | 25 | 0.5 | 1e-5 | 6 | 69 |
| | PNA | None | 0 | 16 | 70 | 70 | 0.001 | 25 | 0.5 | 1e-6 | - | - |
| | PNA | LapPE + RS | 8 | 16 | 80 | 80 | 0.001 | 25 | 0.5 | 1e-6 | - | - |
| | PNA | SignNet | 8 | 16 | 70 | 70 | 0.001 | 25 | 0.5 | 1e-6 | 8 | 70 |
| | PNA | MAP | 8 | 16 | 70 | 70 | 0.001 | 25 | 0.5 | 1e-6 | 6 | 70 |
| | SAN | None | 0 | 10 | 64 | 64 | 0.0003 | 25 | 0.5 | 1e-6 | - | - |
| | SAN | MAP | 16 | 10 | 40 | 40 | 0.0007 | 25 | 0.5 | 1e-5 | 6 | 40 |
| | GraphiT | None | 0 | 10 | 64 | 64 | 0.0003 | 25 | 0.5 | 1e-6 | - | - |
| | GraphiT | MAP | 16 | 10 | 48 | 48 | 0.0007 | 25 | 0.5 | 1e-6 | 6 | 48 |
| MOLTOX21 | GatedGCN | None | 0 | 8 | 154 | 154 | 0.001 | 25 | 0.5 | 1e-5 | - | - |
| | GatedGCN | LapPE + RS | 3 | 8 | 154 | 154 | 0.001 | 25 | 0.5 | 1e-5 | - | - |
| | GatedGCN | MAP | 3 | 8 | 150 | 150 | 0.001 | 22 | 0.14 | 5e-6 | 8 | 150 |
| | PNA | None | 0 | 8 | 206 | 206 | 0.0005 | 10 | 0.8 | 2e-5 | - | - |
| | PNA | MAP | 16 | 8 | 115 | 113 | 0.0005 | 10 | 0.8 | 8e-5 | 7 | 115 |
| | SAN | None | 0 | 10 | 88 | 88 | 0.0007 | 25 | 0.5 | 1e-6 | - | - |
| | SAN | MAP | 12 | 10 | 88 | 88 | 0.0007 | 25 | 0.5 | 1e-5 | 8 | 88 |
| | GraphiT | None | 0 | 10 | 88 | 88 | 0.0007 | 25 | 0.5 | 1e-6 | - | - |
| | GraphiT | MAP | 16 | 10 | 64 | 64 | 0.0007 | 25 | 0.5 | 1e-6 | 6 | 64 |
| MOLPCBA | GatedGCN | None | 0 | 8 | 154 | 154 | 0.001 | 25 | 0.5 | 1e-4 | - | - |
| | GatedGCN | LapPE + RS | 3 | 8 | 154 | 154 | 0.001 | 25 | 0.5 | 1e-4 | - | - |
| | GatedGCN | MAP | 3 | 8 | 200 | 200 | 0.001 | 25 | 0.5 | 1e-5 | 8 | 200 |
| | PNA | None | 0 | 4 | 510 | 510 | 0.0005 | 4 | 0.8 | 2e-5 | - | - |
| | PNA | MAP | 16 | 4 | 304 | 304 | 0.0005 | 10 | 0.8 | 2e-5 | 8 | 304 |

## M.2 Synthetic tasks

To verify the expressive power of RSE, we conduct experiments on the synthetic EXP dataset. The dataset consists of a set of 1-WL indistinguishable non-isomorphic graph pairs. If a network reaches above 50 % accuracy on this dataset, it must have expressive power beyond the 1-WL test. DeepSets-RSE is a two-layer DeepSets model with RSE as PE, whereas Linear-RSE is a one-layer linear model with RSE as PE. We use Optuna [3] to optimize the hyperparameters of our models. The values of hyperparameters of our models are as follows:

- **DeepSets-RSE**: the learning rate $\lambda = 0.002385602941230316$, the hidden dimension of the first linear layer $w_1 = 60$, the hidden dimension of the second linear layer $w_2 = 76$, the dropout rate [46] $p = 0.13592575703525184$, the weight decay of Adam optimizer $wd = 0.0005$.

- **Linear-RSE**: the learning rate $\lambda = 0.0006867736568978745$, the hidden dimension $w = 109$, the weight decay of Adam optimizer $wd = 0.0001$.

