# OpenReview forum: "Laplacian Canonization: A Minimalist Approach to Sign and Basis Invariant Spectral Embedding"
_NeurIPS.cc/2023/Conference — NeurIPS 2023 poster_

### Official Review · Reviewer_R8b6 · 2023-07-06

**Soundness:** 4 excellent
**Presentation:** 4 excellent
**Contribution:** 3 good
**Rating:** 7
**Confidence:** 3

**Summary:**

The authors address the problem of expressiveness in graph neural networks. Positional encodings using spectral approaches have suffered from sign and basis invariance. The authors propose Laplacian canonization which finds unique representations, and they analyze what properties the canonization should preserve and in practice what is the ratio of eigenvectors that fulfill these conditions. They propose a new simple canonization algorithm for sign and  basis invariance—Minimal Axis Projection (MAP)---that uses axis projection functions to identify canonical directions. They also present conditions under which MAP can guarantee sign and basis canonization. Experimental results evaluate MAP on graph classification benchmarks and show improved results with lowest computational runtime.

**Strengths:**

Originality: the approach appears to be novel.

Significance: Designing proper architectures that preserve symmetries and are more expressive is an important and open topic  currently in GNN literature.

Quality: the approach is both simple and fast. Experiments show pairing MAP with multiple architectures to obtain gains in performance.  The ablation study demonstrates the contributions of the different parts of the approach. Full implementation details are provided in the supplement.

Clarity:  the paper is well-written.


**Weaknesses:**

Experimental results show incremental / modest improvement compared to BasisNet. The main contribution is in reduced runtime. Furthermore, BasisNet model with k=all in their paper (Table 1) achieve better performance than MAP in the current paper (Table 3). Authors should explain decision to not include these results from the BasisNet paper


**Questions:**

1. RSE: the eigenvectors should be unit-normalized (or normalized to same length)
2. If using the low frequency eigenvectors, the corresponding eigenvalues are the smallest, and thus the values are very small (especially if also unit normalized and n is large). How is this useful in practice? Are these concatenated scaled in some way at input to the network?
3. Are the uncanonizable eigenvectors in the datasets concentrated in low or high frequency, or spread uniformly?
4. How would this method work for a ring graph (cycle)?
5. Why are result using SignNet not included for tables 4 and 5?
6. Figures 4 and 5 in the appendix are very useful toy illustration that should be moved to the main paper
7. What if all eigenvectors are used for MAP, as in the BasisNet paper (k=all)?

Minor comments:
* line 176: “question that which…”
* line 190 - can reference Table 2
* 3.3.3 - heading should be “summary” not “summarization”
* line 273: eigendecomposition complexity is cubic for dense Laplacians, whereas most real world graphs are sparse, so that is a worst case complexity. In practice it is lower.
Line 336: why does increasing k (8–>16) make performance worse? Shouldn’t the network just utilize the lower freq vectors if they are more important?




**Limitations:**

limitations aren't discussed

---

> ### Author Rebuttal · Authors · 2023-08-09
>
> We thank Reviewer R8b6 for appreciating the simplicity and significance of our paper. We address your concerns as follows.
>
> ---
>
> **Q1.** SignNet model with k=all in their paper (Table 1) achieve better performance than MAP in the current paper (Table 3). Authors should explain decision to not include these results from the BasisNet paper.
>
> **A1**. Among all the baselines of our paper (LapPE, RS, SignNet) and also other papers that also use LapPE [1-2], we find that SignNet seems the only one that incorporates all eigenvectors in PE, which could be problematic for larger graphs (e.g., MalNet-Tiny has 1410 nodes on average). For a consistent comparison, we adopt $k=8$ in our comparison.
>
> **References:**
>
> [1] Kreuzer et al. (2021). Rethinking graph transformers with spectral attention. *NeurIPS.*
>
> [2] Rampášek et al. (2022). Recipe for a general, powerful, scalable graph transformer. *NeurIPS.*
>
> ---
>
> **Q2**. RSE: the eigenvectors should be unit-normalized (or normalized to same length).
>
> **A2**. Here, the eigenvectors in $\mathbf{U}$ are already unit-normalized, and we reweight them with $\mathbf{\Lambda}^{1/2}$ to add eigenvalue information. We will state it clearer in the revision.
>
> ---
>
> **Q3**. If using the low frequency eigenvectors, the corresponding eigenvalues are the smallest, and thus the values are very small (especially if also unit normalized and n is large). How is this useful in practice? Are these concatenated scaled in some way at input to the network?
>
> **A3**. In practice, we used the eigen-decomposition of the adjacency matrix $\hat{\mathbf{A}}=\mathbf{I}-\mathbf{L}$, which has the same eigenvectors as the Laplacian while avoiding the small-eigenvalue issue (whose eigenvalues are $1-\lambda_i$ instead).
>
> ---
>
> **Q4**. Are the uncanonizable eigenvectors in the datasets concentrated in low or high frequency, or spread uniformly?
>
> **A4**. We measured the number of sign-uncanonizable eigenvectors with low, mid, and high frequency on 3 datasets. It appears that uncanonizable eigenvectors are distributed more in high frequency, which is good news as studied in the LapPE paper, as low frequency components usually matter the most for model performance.
>
> | Dataset | Low | Mid | High |
> | --- | --- | --- | --- |
> | MOLTOX21 | 130 | 1271 | 4017 |
> | MOLTOXCAST | 170 | 1413 | 4449 |
> | MOLPCBA | 8723 | 54004 | 280361 |
>
> ---
>
> **Q5**. How would this method work for a ring graph (cycle)?
>
> **A5**. This is an interesting question. As shown below, we find that a larger node size $n$ leads to more uncaonizable features, suggesting a close correlation between automorphism and uncanonizable features. This is understandable because an automorphic graph like rings can rotate itself and stay the same, we cannot specify a unique location for a node. We believe that the theoretical relationship between automorphism and sign/basis canonizability would be an interesting open problem to explore in the future.
>
> | #Nodes | Ratio of sign-canonizable eigenvectors | Ratio of basis-canonizable eigenspaces |
> | --- | --- | --- |
> | 5 | 3/5 | 0/2 |
> | 6 | 2/6 | 0/2 |
> | 7 | 4/7 | 0/3 |
> | 8 | 1/8 | 0/3 |
> | 9 | 5/9 | 0/4 |
> | 10 | 3/10 | 0/4 |
>
> ---
>
> **Q6**. Why are results using SignNet not included for tables 4 and 5?
>
> **A6**. All accuracy performance of the baselines are taken from the original papers of LapPE and SignNet. Some models are not conducted with SignNet because the original paper did not provide them. The SignNet authors only provided codes on ZINC and did not implement them on SAN and GraphiT. To be more complete, here we also reproduce some newly tuned SignNet results on MOLTOX21, where MAP still outperforms SignNet under different backbones.
>
> | Model | PE | $k$ | #Param | ROCAUC |
> | --- | --- | --- | --- | --- |
> | GatedGCN | None | 0 | 1004K | 0.772 ± 0.006 |
> | GatedGCN | SignNet | 3 | 1754K | 0.782 ± 0.004 |
> | GatedGCN | MAP | 3 | 1505K | **0.784 ± 0.005** |
> | PNA | None | 0 | 5245K | 0.755 ± 0.008 |
> | PNA | SignNet | 16 | 1367K | 0.745 ± 0.008 |
> | PNA | MAP | 16 | 1951K | **0.761 ± 0.002** |
>
> ---
>
> **Q7**. Figures 4 and 5 in the appendix are very useful toy illustration that should be moved to the main paper.
>
> **A7**. Thanks for your suggestion. We will move Figures 4 and 5 to the main paper.
>
> ---
>
> **Q8**. What if all eigenvectors are used for MAP, as in the BasisNet paper (k=all)?
>
> **A8**. We try using all eigenvectors with GatedGCN on ZINC and the results are provided below. We can see that MAP has no improvement by using k=all, and thus underperforms SignNet. We believe this is attributed to the uncanonizable eigenvectors being more densely distributed in high frequencies, as shown in **A4**. In comparison, SignNet achieves sign-invariance w.r.t. these uncanonizable eigenvectors (although with more computation cost and losing expressive power) and attains better performance. To validate this hypothesis, we mask the uncanonizable eigenvectors of MAP and conduct experiments again. We call this variant **MAP-mask**. The result shows that masking uncanonizable eigenvectors significantly improves the performance of MAP, reaching comparable performance with SignNet. We believe that in the future, developing better canonization algorithms could further close this gap.
>
> | Model | Test MAE |
> | --- | --- |
> | SignNet  (k=all) | 0.100 ± 0.007 |
> | MAP (k=8) | 0.120 ± 0.002 |
> | MAP (k=all) | 0.121 ± 0.003 |
> | MAP-mask  (k=all) | 0.106 ± 0.001 |
>
> ---
>
> **Q9**. Why does increasing k (8–>16) make performance worse? Shouldn’t the network just utilize the lower freq vectors if they are more important?
>
> **A9**. Here, for efficiency, we only tune hyperparameters on $k=8$ and deploy them to other $k$’s. When further tuned under $k=16$, we can see that they attain the same performance.
>
> | $k$ | 8 | 16 |
> | --- | --- | --- |
> | Test MAE | 0.120 ± 0.002 | 0.120 ± 0.004 |
>
> ---
>
> Hope our elaborations and new experiments above could address your concerns. Please let us know if there is more to clarify.

---

> > ### Comment · Reviewer_R8b6 · 2023-08-15
> > **Response**
> >
> > I thank the authors for the detailed response and am satisfied with the additional results. I am keeping my score.

---

### Official Review · Reviewer_ZUZt · 2023-07-06

**Soundness:** 3 good
**Presentation:** 4 excellent
**Contribution:** 3 good
**Rating:** 7
**Confidence:** 2

**Summary:**

This paper introduces a new approach called Laplacian Canonization (LC) for ensuring the sign and basis invariance of spectral embeddings. This is done by determining the canonical direction of eigenvectors in the pre-processing stage. They propose to perform the Laplacian Canonization via Maximal Axis Projection (MAP) algorithm that is guaranteed to canonize all sign canonizable features. Experiments are performed on  molecular benchmarks.

**Strengths:**

- The paper is well written, contains clear rigorous definitions and is easy to follow.
- Experiments show consistent improvements.


**Weaknesses:**

With my limited knowledge of the topic, I could not point out any obvious weaknesses of the method. I do have two questions regarding the guarantees of the method:
- The authors claim that the method can canonize more than 90% of all eigenvectors, however, this has only been tested on molecular graphs. Are there any guarantees for other graphs?
- You observe that your assumptions are violated for some small percentage of eigenvectors on real-world datasets. I was wondering if these eigenvectors are evenly spread out over the eigenvalue spectrum or could it be the case that these are all for the lowest eigenvalues? This could be problematic since these are the eigenvectors one would like to use.


**Questions:**

Besides the questions, listed in the weaknesses section, I have a few more minor questions:
- In the axis projection step, why do you need c?
- I suppose that in practice alpha_i will not be exactly the same. Do you then consider all values or do you apply some binning algorithm to get B_i? I also wonder what is k in practice?
- in Tables 4 and 5, why is it that MAP increases the number of parameters?


**Limitations:**

I have no limitations to point out and the code is provided in the supplementary material.

---

> ### Author Rebuttal · Authors · 2023-08-09
>
> We thank Reviewer for appreciating our paper. We address your questions in the following points.
>
> ---
>
> **Q1**. The authors claim that the method can canonize more than 90% of all eigenvectors, however, this has only been tested on molecular graphs. Are there any guarantees for other graphs?
>
> **A1**. Following your suggestions, we also evaluate the proposed method on other kinds of graphs. From the table below, we can see that we can also canonize >90% on other kinds of graphs (like networks and database), and have *even less* uncanonizable features, like 2.59% on COLLAB. This is because molecular graphs are already more symmetric than other kinds of real-world graphs.
>
> | Dataset | Ratio of sign-uncanonizable eigenvalues | Ratio of basis-uncanonizable eigenvalues | Total |
> | --- | --- | --- | --- |
> | MOLPCBA (molecular) | 2.24% | 7.37 % | 9.61% |
> | COLLAB (network) | 0.88 % | 1.71 % | 2.59% |
> | IMDB-BINARY (movie database) | 2.29 % | 6.32 % | 8.61% |
> | github-stargazers (network) | 2.81 % | 3.44 % | 6.25% |
>
> ---
>
> **Q2**. You observe that your assumptions are violated for some small percentage of eigenvectors on real-world datasets. I was wondering if these eigenvectors are evenly spread out over the eigenvalue spectrum or could it be the case that these are all for the lowest eigenvalues? This could be problematic since these are the eigenvectors one would like to use.
>
> **A2**. We measured the number of sign-uncanonizable eigenvectors with low, mid, and high frequency on 3 datasets. It appears that uncanonizable eigenvectors are distributed more in high frequency, which is good news as studied in the LapPE paper, as low frequency components usually matter the most for model performance.
>
> | Dataset | Low | Mid | High |
> | --- | --- | --- | --- |
> | MOLTOX21 | 130 | 1271 | 4017 |
> | MOLTOXCAST | 170 | 1413 | 4449 |
> | MOLPCBA | 8723 | 54004 | 280361 |
>
> ---
>
> **Q3**. In the axis projection step, why do you need c?
>
> **A3**. Here, adding a constant $c$ does not affect the correctness of our theorem. But with different $c$’s, we can get different numbers of uncanonizable eigenvectors. Therefore, we can tune $c$ to attain better canonization. In practice, we set $c$ to 10 for all cases as a good default choice.
>
> ---
>
> **Q4**. I suppose that in practice alpha_i will not be exactly the same. Do you then consider all values or do you apply some binning algorithm to get B_i? I also wonder what is k in practice?
>
> **A4**. That is a good point. We do not use binning here. Instead, we judge two float numbers `(a, b)`  to be the same if they are close enough. Specifically, we use the PyTorch function `torch.allclose(a, b)` that checks if two floating point numbers are equal up to some tolerance.
>
> We also measure the average value of $n$ (#nodes) and $k$ (#distinct $\alpha$ values) in 3 datasets. From the table below we can see on average $k$ is slightly smaller but also close to $n$, because there are rather few repeated $\alpha_i$’s.
>
> | Dataset | $n$  | $k$ |
> | --- | --- | --- |
> | MOLTOX21 | 19.1 | 17.3 |
> | MOLTOXCAST | 19.2 | 16.8 |
> | MOLPCBA | 26.0 | 20.2 |
>
> ---
>
> **Q5**. In Tables 4 and 5, why is it that MAP increases the number of parameters?
>
> **A5**. In our experiments, we allow a flexible choice of model architectures (layers, hidden dimensions) to achieve the best performance. This setting is consistent with prior works, specifically [1] and [2]. When further ensuring a similar model size, as shown below, the results are still consistent with the original ones as our MAP still outperforms LapPE+RS.
>
> *New Results on MOLPCBA under similar model size.*
>
> | Model | PE | $k$ | #Param | AP |
> | --- | --- | --- | --- | --- |
> | GatedGCN | None | 0 | 2641K | 0.265 ± 0.003 |
> | GatedGCN | LapPE + RS | 3 | 2642K | 0.266 ± 0.002 |
> | GatedGCN | MAP | 3 | 2658K | **0.268 ± 0.002** |
>
> **References:**
>
> [1] Dwivedi, V. P., Luu, A. T., Laurent, T., Bengio, Y., & Bresson, X. (2021). Graph neural networks with learnable structural and positional representations. *arXiv preprint arXiv:2110.07875*.
>
> [2] Lim, D., Robinson, J., Zhao, L., Smidt, T., Sra, S., Maron, H., & Jegelka, S. (2022). Sign and basis invariant networks for spectral graph representation learning. *arXiv preprint arXiv:2202.13013*.
>
> ---
>
> Hope our new results above could address your concerns. We are very happy to take your further questions.

---

> > ### Comment · Reviewer_ZUZt · 2023-08-14
> > **Thank you for the clarifications**
> >
> > Thanks for taking the time to thoroughly answer my questions. I am happy with the clarifications and provided extra evaluations, and I recommend the paper to be accepted. I increased my score accordingly.

---

### Official Review · Reviewer_NB7V · 2023-07-06

**Soundness:** 2 fair
**Presentation:** 2 fair
**Contribution:** 2 fair
**Rating:** 5
**Confidence:** 2

**Summary:**

The recently emerging graph transformers uses the spectral embedding.
The spectral embedding has two empirically known problems; I) sign invariance ii) basis invariance.
The existing remedy for these problem comes at the cost.
This paper addresses this problem by the method this paper proposes called Laplacian Canonization (LC).
This paper provide the theoretical analyses of LC.
Also, the experimental results show that the LC outperforms the existing methods.

**Strengths:**

- The simplicity of the proposed algorithm.

- Theoretical guarantee of the canonoization for the MAP algorithm.

**Weaknesses:**

- We do not know **why** the sign and basis invariance hinder the performance of PE.
Thus, I kind of feel little anxious that the readers cannot judge if the work on top of the series of these studies is in the right direction or not to address the fundamental problems. See also the limitation section.

- Seeing the experiment, there is a little improvement from SignNet even though the LC deals with not only sign but also the basis invariance.
Why some method is not conducted with sign net? The basis invariance does not matter? Or The LC show the weaker accuracy performance regarding sign ambiguity than SignNet?

**Questions:**

Regarding Table 2.
From the intuition, the basis invariance is quite rare, while the sign invariance always happens, since the basis invariance needs the "shared eigenvalues."
But looking at the table 2, sign non-canonized is less than basis invariance.
Thus, how do you justify the definition of the canonization? Or basis invariance matters more?
Also, if the basis invariance is more frequent than sign invariance, how do you defend the little improvement from SignNet?



Basis invariance vs. sign invariance.
As I raise the weakness section, the LC improves the accuracy of the SignNet only marginally. But seeing the Table 2, the there are more basis invariance.
Is this an indirect evidence that the LC does not deal with sign not quite well? Or basis invaraince actually does not matter?

It will be interesting if the authors conduct the experiments using the "separate" MAP regarding basis and sign.

Thus, I believe that at least one of the following is happening

i) some non-intuitional thing is happening in Table 2 regarding the number of noncanonization of sign v basis.

ii) Even if the number of non canonization of the basis, basis does not matter

iii) The proposed method deal with the sign case less well than SignNet.

As I wrote in this section, I am not clear the point how much the sign and basis matters, and how MAP addresses each invariance, and therefore I'm giving 4.

---
POST REBUTTAL: Given the synthetic data experiment regarding basis, I slightly increase my score from 4 to 5.


Also, the discussion above on Table 2 contains my misunderstanding. Please see the discussion below for the details.

**Limitations:**

The whole community does not know why the sign and basis invariance hamper the performance of PE as I raised in the weakness section, as far as I understand. This study is built on these studies. But although I read the Appendix A where the authors summarize the point of basis and invariance ambiguity, I'm less confident on this point.

---

> ### Author Rebuttal · Authors · 2023-08-09
>
> We thank Reviewer NB7V for appreciating the simplicity and theoretical guarantees of our approach. We address your main concerns below, especially those on the meaning of studying sign and basis invariance.
>
> ---
>
> **Q1**. We do not know **why** the sign and basis invariance hinder the performance of PE.
>
> **A1**. Indeed this is a good question. First, it is a well-known principle (and practice) that we need to obey the symmetry properties of graphs when designing GNNs. For example, graph convolution and aggregation operators preserve the permutation invariance property of graphs. Similarly, **sign and basis invariance is also an intrinsic symmetry property of the graph data (the graph structure is invariant under different sign/basis choices of eigenvectors),** so similarly, we also need to preserve this symmetry when designing GNNs. As noted by Reviewer R8b6, *“designing proper architectures that preserve symmetries and are more expressive is an important and open topic currently in GNN literature.”* As also mentioned by Reviewer zwuu, sign/basis ambiguities are important problems not only in the GNN community, but in other fields as well.
>
> | Model | Method for Sign/Basis Invariance | MAE on ZINC |
> | --- | --- | --- |
> | GatedGCN + LapPE | None | 0.319 ± 0.010 |
> |  | RandomSign (RS) | 0.202 ± 0.006 |
> |  | SignNet ($\phi(v)$ only) | 0.148 ± 0.007 |
> |  | SignNet | 0.121 ± 0.005 |
> |  | MAP (ours) | 0.120 ± 0.002 |
>
> **Reasons.** Intuitively, GNNs without symmetries can waste a lot of model capacity on fitting exponentially many permutation/sign/basis ambiguities of the same graph, while GNNs obeying symmetries do not need to. Concretely, the benefits of this principle can be justified in two folds:
>
> - **Empirically**, as shown in the table above (quoted from Table 3), **encouraging sign/basis invariance by RS, SignNet, and our MAP can all bring significant improvements on real-world datasets**.
> - **Theoretically**, a recent paper [1] rigorously shows that we can **gain sample complexity by obeying data symmetry**. The upper bound of the generalization error in Theorem 3.1 [1] contains a term $\mathop{\mathrm{vol}}(M/G)$, the volume of the quotient space that is invariant to group $G$ on the manifold $M$. As the group size $G$ grows (more symmetries), we observe a decreasing generalization error (or equivalently, reduced sample size to attain the same error).
>
> Thus, sign/basis ambiguity is not only a well-established problem, but also solving it brings concrete benefits empirically and theoretically. We will add these discussions in the revision.
>
> **References:**
>
> [1] Tahmasebi, B., & Jegelka, S. (2023). The Exact Sample Complexity Gain from Invariances for Kernel Regression on Manifolds. *ICML 2023 Workshop on Topology, Algebra, and Geometry in Machine Learning (TAG-ML)*.
>
> ---
>
> **Q2**. Why some method is not conducted with sign net? The basis invariance does not matter? Or The LC show the weaker accuracy performance regarding sign ambiguity than SignNet?
>
> **A2**. Here, for a fair comparison, we compare with reported scores from the original papers of LapPE and SignNet. In the SignNet paper and their official code, for graph tasks, **they only provide results on ZINC,** as we included in Table 3. We also find that directly porting their code to other datasets leads to much worse performance and requires costly tuning. To be more complete, here we reproduce a delicately tuned SignNet result on MOLTOX21, and we can see that MAP still outperforms SignNet.
>
> | Model | PE | $k$ | #Param | ROCAUC |
> | --- | --- | --- | --- | --- |
> | GatedGCN | None | 0 | 1004K | 0.772 ± 0.006 |
> | GatedGCN | SignNet | 3 | 1754K | 0.782 ± 0.004 |
> | GatedGCN | MAP | 3 | 1505K | **0.784 ± 0.005** |
> | PNA | None | 0 | 5245K | 0.755 ± 0.008 |
> | PNA | SignNet | 16 | 1367K | 0.745 ± 0.008 |
> | PNA | MAP | 16 | 1951K | **0.761 ± 0.002** |
>
> ---
>
> **Q3**. From the intuition, the basis invariance is quite rare, while the sign invariance always happens, since the basis invariance needs the "shared eigenvalues.” Seeing the Table 2, there are more basis invariance than sign invariance.  [Several questions are further raised based on this observation.]
>
> **A3**. We are afraid there are some misunderstandings of the Table 2 here. As you noted, there are indeed much fewer basis ambiguities (only around **5-6%**, see **Table 9**) than sign ambiguities (**100%**) in all eigenvectors. Clearly, **sign ambiguity is indeed more frequent and more important than basis ambiguity.** Table 2 lists the proportion of MAP-uncanonizable eigenvectors (i.e., **all the other can be canonized by MAP**), e.g., 2.5% for sign and 1.6% for basis on ZINC. Thus, **our MAP algorithm can successfully resolve ~97.5% sign ambiguities and ~3.4% basis ambiguities** (70% within basis itself) among all eigenvectors. So MAP indeed did better at solving sign ambiguity, which also contributes most to its improvements (see **Table 8**)**.** We will explain this relationship more clearly to avoid possible confusion in the revision.
>
> We believe that this clarification could also help address your sequential concerns on the importance of sign and basis invariance. Please let us know if there is more to clarify.
>
> ---
>
> **Q4**. It will be interesting if the authors conduct the experiments using the “separate” MAP regarding basis and sign.
>
> **A4**. We note that in **Table 8**, we have included an ablation study on the MAP-sign and MAP-basis methods. We can see that while both methods contribute to the final performance, removing MAP-sign hurts the performance more than removing MAP-basis, which also confirms our explanations above.
>
> ---
>
> Hope our elaboration on the meaning of the problem, and our clarification on the quantities in Table 2 could address your concerns. We are very happy to take your further questions.

---

> > ### Comment · Reviewer_NB7V · 2023-08-10
> > **Thank you for clarifying!**
> >
> > Thank you very much for rebuttal.
> >
> > First of all, thank you very much for clarifying Table 2. I need to admit that I had misunderstanding on the Table.
> > Also, thank you very much for the detailed discussion on why the sign and basis invariance hinder the performance of PE. The theory part is particularly interesting.
> >
> > At the same time, if I'm not wrong, experimentally we still do not observe improvements of MAP over SignNet, except for PNA on MOLTOX21 (additional experiment for my comment). Although the averages seem to be slightly better, both values are within the close range of the deviations of multiple runs. For example, for GatedGCN + LapPE on ZINC, 0.121 $\pm$ 0.005 for SignNet and 0.120 $\pm$ 0.002 for MAP (yours) seems almost the same, and for PNA on ZINC, 0.105 $\pm$ 0.007 for SignNet and 0.101 $\pm$ 0.005 for MAP (yours) are also almost same. Do you have any other advantages of MAP over SignNet, since the performance is somewhat weak to appeal.

---

> > > ### Author Response · Authors · 2023-08-11
> > > **Further Response to Reviewer NB7V**
> > >
> > > Thanks for the prompt response and for appreciating our explanations! We will certainly add these explanations in the revision.
> > >
> > > As for the comparable performance between MAP and SignNet, we note that we have mentioned and explained this phenomenon in the experiment session (L314-319),  as quoted below:
> > >
> > > > Third, we also observe that **MAP and SignNet achieve  comparable performance**. This is because **both methods aim at the same goal—eliminating ambiguity**. However, SignNet does so in the training stage while MAP does so in the pre-processing stage, thus **the latter is more computationally efficient**. Lastly, we would also like to highlight that as a kind of positional encoding, **MAP can be easily incorporated with any GNN architecture by passing the ```pre_transform``` function to the dataset class with a single line of code**.
> > >
> > > To summarize, the comparable performance is **expected** because they can both address the sign ambiguity problem well. However, the two adopt quite different approaches to achieve this goal: SignNet uses a dual-branch NN, while ours only uses a **learning-free preprocessing algorithm**. As a new approach, our Laplacian canonization (MAP) has the following advantages:
> > >
> > > - **Efficiency.** As a preprocessing method, MAP only needs to preprocess the graphs once before training, while SignNet needs to propagate and update the dual-branch NN during training, which leads to a lot more computation cost. As shown in Table 6, GatedGCN+MAP takes 64.72h while GatedGCN + SignNet takes 108.78h during training, i.e., SignNet costs 68% more training time than MAP to attain comparable performance.
> > > - **Simplicity and Generality.** As a learning-free algorithm, MAP does not have hyperparameters and module designs to tune  ($c$ is the only hyperparameter and we use $c=10$ in all cases). As a preprocessing method, it can be applied to any existing GNNs using Laplacian embedding **with NO change on the model architecture and training process**. In comparison, SignNet introduces new modules and requires specific tuning on each model/dataset to work well. Thus, MAP is more generally applicable and easy to use than SignNet.
> > > - **Applicable for basis invariance.** We note that the basis version of SignNet, i.e., BasisNet, is very computationally prohibitive ($O(n^m)$ where $m=O(n^2)$) that is not applicable on real-world data (see SignNet paper). Thus, on real-world data, SignNet can actually only address sign invariance, and our MAP algorithm also can efficiently solve basis invariance to some extent (~70% of features are canonizable by MAP, Table 2).
> > >
> > > Given the above advantages, we believe that MAP offers a new and promising alternative to SignNet for addressing the sign/basis ambiguity problem.
> > >
> > > Hope the explanation above could address your concerns! We are happy to take your further questions during the discussion stage.

---

> > > > ### Comment · Reviewer_NB7V · 2023-08-13
> > > > **Still confused and appreciate more help**
> > > >
> > > > Hi,
> > > >
> > > > Thank you very much for the follow-up comment. I'm still somewhat confused and it'd be nice if you can help me out.
> > > >
> > > > In L59-61, the authors claim as follows
> > > >
> > > > >  Empirically, we show that employing the MAP-canonized spectral embedding yields significant improvements over the vanilla RandSign approach, and even outperforms SignNet on large-scale benchmark datasets like OGBG [18]
> > > >
> > > > The authors wrote in L314-L315 as well as quoted in the response to me as
> > > >
> > > > >  Third, we also observe that MAP and SignNet achieve comparable performance.
> > > >
> > > > Here comes my first question;
> > > >
> > > > - 1. In what sense did you claim MAP **outperforms** SignNet?
> > > >
> > > > As the authors acknowledged as well, MAP and SignNet achieves comparable performance in terms of accuracy. Did you say outperform in a sense of efficiency?
> > > >
> > > > My second and third questions are related to the following points.
> > > >
> > > > In L315 the sentence below follows the quote above.
> > > >
> > > > > This is because both methods aim at the same goal—eliminating ambiguity.
> > > >
> > > > Also, in the response the authors wrote
> > > >
> > > > >  the comparable performance is **expected** because they can both address the sign ambiguity problem well.
> > > >
> > > > Here the second and third questions are
> > > >
> > > > - 2. Why did you **expect** the comparable performance between MAP and SignNet?
> > > > - 3. Why did the two actually achieve the comparable performance.
> > > >
> > > > If I understand correctly, there are two type of ambiguities are known, sign and basis.
> > > > If we see Table 1, while SignNet deals only with sign ambiguity, MAP deals with sign **and** basis, which is the one of the selling points of MAP.
> > > > Therefore, in theory if I understand correctly, MAP will handle more types of ambiguity than SignNet, and as a consequence I expect more for MAP.
> > > > So, I still don't understand MAP and SignNet "aim at the same goal—eliminating ambiguity.", since MAP may have more targets than SignNet.
> > > > Therefore, I have questions. Why do you expect the comparable performance? Also, why did the two achieve the similar performance?
> > > >
> > > > I'm still have a confusion as above, but correct me if I'm wrong.
> > > >
> > > > At the same time, I understand that MAP has an advantage in efficiency, simplicity and generality over SignNet.

---

> > > > > ### Author Response · Authors · 2023-08-14
> > > > > **Further Response to Reviewer NB7V**
> > > > >
> > > > > Thanks for your careful reading and for acknowledging our explanations on the efficiency, simplicity, and generality of MAP. Meanwhile, we are sorry to cause the confusion. Below we will make some clarifications on the claims that you mentioned.
> > > > >
> > > > > ---
> > > > >
> > > > > **Q5.** In what sense did you claim MAP **outperforms** SignNet? As the authors acknowledged as well, MAP and SignNet achieves comparable performance in terms of accuracy. Did you say outperform in a sense of efficiency?
> > > > >
> > > > > **A5**. Indeed, despite that MAP can outperform SignNet a bit in accuracy, the major advantage of MAP is its simplicity and generality, where MAP indeed shows clear gains over SignNet (Table 6). Thanks for pointing it out and we will make this point clearer in the revision.
> > > > >
> > > > > ---
> > > > >
> > > > > **Q6.** Why did you **expect** the comparable performance between MAP and SignNet? Why did the two actually achieve the comparable performance?
> > > > >
> > > > > **A6.** We get that your key question here is why MAP and SignNet perform comparably when MAP addresses both sign & basis ambiguities while SignNet only addresses the first. The key reason is that, as we explained in the previous reply, **the sign ambiguity problem** **is much more frequent than the basis ambiguity problem** (100% vs ~5%). Thus, **eliminating sign ambiguity usually contributes to a large proportion of the gain of eliminating both ambiguities.** To see this, Table 8 (quoted below) shows that MAP-sign indeed performs comparably to SignNet (0.122 vs 0.121) as they both focus on sign ambiguity. Due to the fewer frequency of basis ambiguity problem, further ensuring basis invariance (MAP-sign → MAP) only improves 0.002, so the two methods (SignNet and MAP) obtain similar performance.
> > > > >
> > > > > Nevertheless, it does not mean that basis invariance is always useless. Instead, whether basis invariance matters should depend on the specific domains and model choices. For example, under the PNA backbone, MAP can improve SignNet by a larger margin (0.105 → 0.101, Table 3). Also, for tasks where basis ambiguity is highly influential on target variables, it is also important to ensure basis invariance. In these cases, basis version of SignNet, i.e., BasisNet, is not practicable due to its high computational cost of $O(n^m)$ ($m=O(n^2)$), while our proposed MAP can still work here. Although MAP-basis maybe only a preliminary solution for basis ambiguity, we believe that with the better canonization algorithm, we can further unleash the benefits of basis invariance. As the first work towards this direction, our established framework and theoretical properties could potentially provide some guidelines for future designs.
> > > > >
> > > > > *Ablation experiments on ZINC with GatedGCN (quoted from Table 8).*
> > > > >
> > > > > | PE Method | SignNet | MAP (sign+basis) | MAP-sign | MAP-basis |
> > > > > | --- | --- | --- | --- | --- |
> > > > > | Test MAE | 0.121 ± 0.005 | 0.120 ± 0.002  | 0.122 ± 0.003 | 0.131 ± 0.003  |
> > > > >
> > > > > ---
> > > > >
> > > > > Hope the explanations above could address your concerns! We will certainly revise them to be more accurate and explicit. Please let us know if you have further concerns.

---

> > > > > > ### Comment · Reviewer_NB7V · 2023-08-16
> > > > > > **Thank you for the all the responses!**
> > > > > >
> > > > > > Thank you very much for the response.
> > > > > > Your explanation makes sense, sign is 100%, while basis might be far less. I have no further questions to ask.
> > > > > >
> > > > > > After reading the response and paper again, I think that this paper will benefit significantly if the authors find "the specific domains and model choices" where the basis really matters.
> > > > > > I am concerned that the current experiments only show comparable performance in terms of accuracy, despite the paper's selling.
> > > > > >
> > > > > > I would describe this in the following two ways. However, the two ways are the different sides of the same coin.
> > > > > >
> > > > > > 1. Presentation issue
> > > > > >
> > > > > > The current presentation, particularly the introduction, is not as beautiful as it goes like "MAP has all (like Table 1), also outperforms the existing ones".
> > > > > > Instead, I feel the authors need to write reservations about the discussion above.
> > > > > >
> > > > > > - Instead of claiming the MAP outperforms the existing method including SignNet (L61) the authors need to provide more words on this, such as
> > > > > >
> > > > > > i. Regarding accuracy, MAP achieves the **comparable** performance to the existing method. (From the previous discussion)
> > > > > >
> > > > > > ii. Although MAP handles more ambiguity (sign + basis) than the practical existing ones (i.e., ones from Table 8 except BasisNet), in most cases, the datasets the authors use do not have much basis ambiguity, which hampers the accuracy. (From Table 3 and the previous discussion)
> > > > > >
> > > > > > iii. Moreover, SignNet, MAP without basis, and MAP full all show comparable performance. (From the quoted table in the previous discussion)
> > > > > >
> > > > > > iv. At the same time, MAP is shown to be more efficient. (Table 6)
> > > > > >
> > > > > > - Also, authors may want to write like finding the specific domains and model choices where basis matters is future work
> > > > > >
> > > > > >
> > > > > > 2. Finding real application where basis matters
> > > > > >
> > > > > > Since improvement from BasisNet to MAP is enormous, almost like impractical to practical, I recommend the authors find datasets where the basis ambiguity really matters. Even the synthetic data will help, but ideally, it'll be nice if "the specific domains and model choices," as mentioned in the previous discussion, are provided.
> > > > > >
> > > > > >
> > > > > >
> > > > > > --
> > > > > >
> > > > > >
> > > > > > I prefer framing this as an issue of lack of real application to an issue of a presentation.
> > > > > > I think that writing reservations reduces the enjoyment of reading this paper.
> > > > > > The real experimental selling point is the efficiency (Table 6).
> > > > > > However, MAP is not only about efficiency from its design. MAP also enjoys eliminating basis ambiguity.
> > > > > > I hope this paper finds real applications where basis really matters and unleashes the full potential of MAP, which can really deal with sign and basis.
> > > > > >
> > > > > >
> > > > > > Thus, I maintain my score 4. I hope this paper gets published in the future at a top venue with strong application on basis.

---

> > > > > > > ### Author Response · Authors · 2023-08-17
> > > > > > > **Further Response and New Results on Basis Invariance**
> > > > > > >
> > > > > > > Thanks for your reply and the new comments! We are very glad to hear that your concerns are addressed. We will address your remaining points below.
> > > > > > >
> > > > > > > ---
> > > > > > >
> > > > > > > **Q1.** Presentation issue.
> > > > > > >
> > > > > > > > The current presentation, particularly the introduction, is not as beautiful as it goes like "MAP has all (like Table 1), also outperforms the existing ones". Instead, I feel the authors need to write reservations about the discussion above. [And more suggestions…]
> > > > > > > >
> > > > > > >
> > > > > > > **A1.** Thanks for your detailed and valuable suggestions! We totally agree that we can rephrase the words to present MAP’s properties more precisely. We will tune down the claims here to make them more explicit. And as you suggested, we will mention 1) MAP achieved comparable performance but with much better efficiency; 2) the basis part of MAP is less influential on real-world data due to the small portion of basis ambiguity.
> > > > > > >
> > > > > > > ---
> > > > > > >
> > > > > > > **Q2.** Finding real application where basis matters.
> > > > > > >
> > > > > > > > Since improvement from BasisNet to MAP is enormous, almost like impractical to practical, I recommend the authors find datasets where the basis ambiguity really matters. Even the synthetic data will help, but ideally, it'll be nice if "the specific domains and model choices," as mentioned in the previous discussion, are provided.
> > > > > > > >
> > > > > > >
> > > > > > > **A2.** Thank you for your insightful feedback. Firstly, we'd like to emphasize that the primary contribution of our work is indeed the MAP-sign. Its effective performance on real-world data is demonstrative of this. The capability of MAP to address basis invariance is, in many ways, an added advantage rather than the core contribution.
> > > > > > >
> > > > > > > Nevertheless, we understand and appreciate your point that it will be nicer to see the advantages of MAP on addressing basis ambiguity, compared to RandSign and SignNet.
> > > > > > >
> > > > > > > To address this, we undertook further research. As mentioned in our paper, real-world graphs mostly exhibit a maximum of 10% basis ambiguity. Up to this point, we've not identified a suitable real-world dataset. In Appendix G, we've highlighted that the probability of basis ambiguity approaches 0 on random graphs as $n\to \infty.$ Therefore, **basis ambiguity primarily exists in small and highly symmetric graphs**. Given this, and in line with common practices for expressive GNNs, we decided to utilize synthetic hard examples to test the efficacy of our proposed method.
> > > > > > >
> > > > > > > **Synthetic Experiment.** We used Graph Isomorphic Testing, a traditional graph task. Our focus was on 10 non-isomorphic random weighted graphs ($G_1,\dots,G_{10}$), all exhibiting basis ambiguity issues (with the first three eigenvectors belonging to the same eigenspace). We sampled 20 instances for each graph, introducing different permutations and basis choices for the initial eigenspace. The dataset was then split into a 9:1 ratio for training and testing, respectively. The task was a 10-way classification, where the aim was to determine the isomorphism of a given graph to one of the 10 original graphs. The model was given the first 3 eigenvectors as input (i.e. $k=3$). The results were averaged over 4 different runs.
> > > > > > >
> > > > > > > *Test accuracy of the synthetic graph isomorphic testing task with DeepSets using different PEs*
> > > > > > >
> > > > > > > | PE | Accuracy |
> > > > > > > | --- | --- |
> > > > > > > | LapPE | 0.11 ± 0.08 |
> > > > > > > | LapPE + RandSign | 0.10 ± 0.09 |
> > > > > > > | LapPE + SignNet | 0.10 ± 0.03 |
> > > > > > > | LapPE + MAP | **0.84 ± 0.21** |
> > > > > > >
> > > > > > > **Results.** As evident from the results, approaches that address sign ambiguity (like RandSign and SignNet) cannot obtain nontrivial performance on this task. Conversely, MAP shows commendable performance. The 84% accuracy, although impressive, indicates potential avenues for further enhancement. We believe this synthetic task could also serve as a valuable benchmark for future studies addressing basis invariance through canonization.
> > > > > > >
> > > > > > > ---
> > > > > > >
> > > > > > > Hope the above clarification and additional experiments could ease your concerns. We are happy to take your further questions!

---

> > > > > > > > ### Comment · Reviewer_NB7V · 2023-08-17
> > > > > > > > **Slightly Increasing my score from 4 to 5**
> > > > > > > >
> > > > > > > > Thank you for your response.
> > > > > > > > Seeing the synthetic dataset experiment, I decided to slightly increase my score from 4 to 5.
> > > > > > > > By this we observe that MAP can handle basis and sign.
> > > > > > > >
> > > > > > > > However, my recommendation by the time of camera-ready is that find one more real world dataset where MAP beats SignNet (and the others) with large margin other than PNA for MOLTOX21. It is intuitional that eliminating at-most 10% ambiguity contributes to improve the accuracy with some recognizable margin.
> > > > > > > > If you find one or two datasets like this, you don't have to rewrite the "reservations" as above.
> > > > > > > > Ideally I wanted to see some "domains" where basis matters, but given the efficiency improvement from SignNet, without finding domains this paper may be accepted.
> > > > > > > >
> > > > > > > >
> > > > > > > > This is the reason why I still have 5. By finding the real dataset where MAP beats SignNet like PNA for MOLTOX21 (in the rebuttal additional experiments), I would give 6. By finding domain, I would give 7.
> > > > > > > >
> > > > > > > > Also, my recommendation is that you should not underrate basis ambiguity by yourself, since the one of the selling points is that yours handles the basis ambiguity. It's like holding the brake and the accelerator at the same time. You cannot say on one hand MAPs can handle basis on the other hand MAP does not see improvement over SignNet since basis doesn't matter.

---

> > > > > ### Comment · Area_Chair_YtJD · 2023-08-16
> > > > > **#5861**
> > > > >
> > > > > The authors have provided additional details. Have your concerns/questions been addressed?

---

### Official Review · Reviewer_zwuu · 2023-07-06

**Soundness:** 3 good
**Presentation:** 4 excellent
**Contribution:** 3 good
**Rating:** 6
**Confidence:** 4

**Summary:**

The authors propose Laplacian canonization, a way to select canonical Laplacian embeddings that resolve the sign and basis ambiguities often present in graph embeddings. The proposed method is a preprocessing step that is relatively fast. The authors perform experiments to evaluate the performance of Laplacian canonization.

**Strengths:**

- Originality: Laplacian canonization is an original idea. To the best of my knowledge, this is a novel contribution.

- Quality: I see this paper as mainly making a methodological contribution. As such, the quality of the experiments are sufficiently extensive to be convincing, and the arguments/theoretical derivations are sound, to the best of my knowledge.

- Clarity: The presentation of the paper and the motivations are clear.

- Significance: It is certainly very relevant and important to study graph embeddings for GNNs these days. The approach is novel and works well. It is significant enough to warrant publication at a venue like NeurIPS.

**Weaknesses:**

- One potential weakness is the theoretical portion of the paper, whose results are rather marginal and unsurprising. However, I see this as a methods paper, and the empirical good performance of the proposed method more than makes up for it.

- One potential limitation of this approach is that not all eigenvectors can be canonized (even though in the datasets 90% of them can be). The authors seem to regard the 90% canonizable rate as a good feature of their approach, rather than a shortcoming. This is a fair perspective. But to provide a more balanced discussion, I would like to see the authors discuss more on this potential limitation, especially since 1. other methods for resolving ambiguities do not suffer from this issue and 2. it is unclear to me whether it is possible that the non-canonizable eigenvectors share any common patterns/structures in real datasets that might bias the results.

- There is a literature (beyond GNN) that also considers spectral embeddings of graphs and going around the basis/sign ambiguity problems. For example, in point cloud registration, Lai and Zhao's "Multiscale Nonrigid Point Cloud Registration Using Rotation-Invariant Sliced-Wasserstein Distance via Laplace-Beltrami Eigenmap. SIAM J. Imaging Sci. 10(2): 449-483 (2017)", and in graph comparison Tam and Dunson's  "Multiscale graph comparison via the embedded laplacian distance. arXiv preprint arXiv:2201.12064 (2022)." I suggest incorporating these references and others to round out the prior work section in the appendix.


- Typos:

line 347: practical

**Questions:**

- See above section

**Limitations:**

- See above section on weakness.

---

> ### Author Rebuttal · Authors · 2023-08-09
>
> We thank Reviewer zwuu for appreciating the originality and effectiveness of the proposed canonization method. We address your concerns as follows.
>
> ---
>
> **Q1**. One potential weakness is the theoretical portion of the paper, whose results are rather marginal and unsurprising. However, I see this as a methods paper, and the empirical good performance of the proposed method more than makes up for it.
>
> **A1**. Thank you for appreciating the good performance of our method. However, we are a bit confused about your comments on the theory part as “marginal and unsurprising”. As far as we could see, our theoretical results establish the theoretical formulation and prove some important properties for Laplacian canonization, making it a theoretically rigorous approach for resolving the ambiguities. ****Reviewer pkjp comments that this theoretical formulation is “important for this community”****. Here, we highlight some ***new and valuable results that have not been observed by prior works***:
>
> 1. We constructed a theoretical framework for canonical forms with differnet invariances and equivariances. **Prior works only consider canonical forms with one kind of invariance or equivariance, and do not face the issue of uncanonizability**, thus their theories do not apply to the sign/basis ambiguity of Laplacian eigenvectors.
> 2. Using this theoretical framework, we found that universality, permutation equivariance, and sign/basis invariance cannot be achieved at the same time (last point in Appendix A). **Past works on sign/basis ambiguity do not propose this observation**, and SignNet actually tries to propose a “universal” sign-invariant network. However, as we discussed in Appendix A, their “universality” does not take permutation equivariance into account, and it’s impossible to be universal when you do.
> 3. We gave the necessary and sufficient condition of uncanonizability and showed their ratio on real datasets. **Previously we knew that sign/basis ambiguity is harmful for LapPE, but didn’t know the extent, and we were not aware why LapPE still underperforms RWPE even after removing sign ambiguity in some experiments [1].** The characterization of these uncanonizable eigenvectors can help us better understand the harm brought by sign/basis ambiguities.
>
> Hope the explanations above could ease your concerns. If there are more specific questions on the theory part, and we are happy to address them in the discussion stage.
>
> **References:**
>
> [1] Rampášek et al. (2022). Recipe for a general, powerful, scalable graph transformer. *NeurIPS*.
>
> ---
>
> **Q2**. I would like to see the authors discuss more on this potential limitation, especially since 1. other methods for resolving ambiguities do not suffer from this issue and 2. it is unclear to me whether it is possible that the non-canonizable eigenvectors share any common patterns/structures in real datasets that might bias the results.
>
> **A2**. Thanks for your suggestions. We highlight that as elaborated in **A1**, three desirable properties of GNNs, universality (U), permutation equivariance (P), and sign invariance (S) **cannot be achieved at the same time**. So when preserving permutation equivariance, there is **a fundamental tradeoff between universality (expressive power) and sign invariance**. Accordingly, the fact that other methods attain both P and S (like SignNet) cannot attain universal expressive power. Instead, our methods attain universality and permutation invariance, while preserving S as much as possible by Laplacian Canonization.
>
> As for the common patterns of these non-canonizable eigenvectors, **Corollary 1** suggests that they are highly symmetric (having identical positive and negative parts up to a permutation). Intuitively these non-canonizable eigenvectors would appear more often in graphs that are more symmetric (e.g. having high-order automorphisms), and predictions of these graphs might be more negatively affected. Of course it still requires more rigorous research as to how structural symmetries are related with non-canonizable eigenvectors and how they might bias the results.
>
> ---
>
> **Q3**. There is a literature (beyond GNN) that also considers spectral embeddings of graphs and going around the basis/sign ambiguity problems. For example, in point cloud registration, [1] Lai and Zhao’s “Multiscale Nonrigid Point Cloud Registration Using Rotation-Invariant Sliced-Wasserstein Distance via Laplace-Beltrami Eigenmap. *SIAM J. Imaging Sci. 10(2)*: 449-483 (2017)” and in graph comparison [2] Tam and Dunson’s “Multiscale graph comparison via the embedded laplacian distance. *arXiv preprint arXiv:2201.12064* (2022).”
>
> **A3**. Thanks for your suggestion, and we will incorporate these works in the related works. Both papers proposed ways to address the sign/basis ambiguity issue of Laplacian eigenvectors. The paper [1] proposed to address sign/basis ambiguities using optimal transport theory that involves solving a non-convex optimization problem, thus it could be less efficient than our approach. The paper [2] proposed to symmetrize the embedding using a heuristic measure called ELD that is quite similar to the form of SignNet, while our MAP algorithm offers an axis projection approach and establish its theoretical guarantees.
>
> ---
>
> Hope our elaborations above could address your concerns. Please let us know if there is more to clarify.

---

> > ### Comment · Reviewer_zwuu · 2023-08-11
> > **reply to authors**
> >
> > I thank the authors for replying to my comments. I think that the proposed modifications and the clarifications from the authors have sufficiently addressed my concerns. I am satisfied with the authors' response.

---

### Official Review · Reviewer_pkjp · 2023-07-08

**Soundness:** 3 good
**Presentation:** 3 good
**Contribution:** 3 good
**Rating:** 6
**Confidence:** 3

**Summary:**

This paper explores the Laplacian canonization approach to address the sign and basis ambiguities of eigenvectors. Previous sign- and basis-invariant methods suffer from high complexity and the proposed canonization method is light-weighted and can be used for any graph neural networks. Since the Laplacian canonization algorithm only runs in the pre-processing stage, it significantly reduces the forward and backward overhead of the neural networks. Experimental results on various graph classification datasets validate the effectiveness and efficiency of the proposed method.

**Strengths:**

1. The proposed canonization method is effective and efficient. Existing sign- and basis-invariant models suffer from high complexity. And this paper addresses this issue by proposing a new pre-processing algorithm, which not only reduces the training computation costs but also makes the model suitable for any graph neural network.

2. This paper gives a theoretical framework for Laplacian canonization and detailly discusses the conditions for canonizing the sign and basis invariance of eigenvectors, which I think is important for this community. Based on the theoretical results, this paper proposes an efficient canonization algorithm that can heuristically determine the signs of eigenvectors.

3. This paper is well-written and easy to follow.

**Weaknesses:**

In the experiment, this paper tests the performance of different positional encoding methods with the same base model. To ensure fairness, the authors should ensure that different methods have similar model parameters. For example, in the PCBA dataset, the model parameters of GatedGCN-MAP are 2.5 times that of the GatedGCN-LapPE but the performance improvement is negligible. Besides, it is confusing that the parameters of PNA-None and GraphiT-None are fewer than PNA-MAP and GraphiT-MAP. Why removing the positional encoding will increase the number of parameters?

**Questions:**

See weakness.

**Limitations:**

This paper discusses the limitation of the proposed algorithm that not all eigenvectors can be canonized and shows that this situation does not have a potential impact on the model.

---

> ### Author Rebuttal · Authors · 2023-08-09
>
> We thank Reviewer pkjp for appreciating our method and theoretical results. We address your concerns on parameter sizes.
>
> ---
>
> **Q1**. To ensure fairness, the authors should ensure that different methods have similar model parameters.
>
> **A1**. In our experiments, we allow a flexible choice of model architectures (layers, hidden dimensions) to achieve the best performance.  This setting is consistent with prior works, specifically [1] and [2]. Following your suggestion, we further ensure the same model size for baseline methods. As shown below, the results are consistent with the original ones as our MAP still outperforms LapPE+RS.
>
> *New Results on MOLPCBA under similar model size.*
>
> | Model | PE | $k$ | #Param | AP |
> | --- | --- | --- | --- | --- |
> | GatedGCN | None | 0 | 2641K | 0.265 ± 0.003 |
> | GatedGCN | LapPE + RS | 3 | 2642K | 0.266 ± 0.002 |
> | GatedGCN | MAP | 3 | 2658K | **0.268 ± 0.002** |
>
> ---
>
> **Q2**. Why the parameters of PNA-None and GraphiT-None are fewer than PNA-MAP and GraphiT-MAP? Why removing the positional encoding will increase the number of parameters?
>
> **A2**. We note that removing PE alone does not increase the number of parameters on a fixed model, but decreases it a bit with smaller input size. Similar to **A1**, we also allow a flexible choice of model size to achieve the best performance, which may choose an even smaller model than the baseline. In these cases, MAP-based models can outperform baselines with even fewer parameters, showing their effectiveness. Here, we further rerun MAP with roughly the same model size as the baseline methods (PNA/GraphiT) on MOLTOX21. It can be seen that MAP-based models still consistently outperform their baselines.
>
> *New Results on MOLTOX21 under similar model size.*
>
> | Model | PE | $k$ | #Param | ROCAUC |
> | --- | --- | --- | --- | --- |
> | PNA | None | 0 | 5245K | 0.755 ± 0.008 |
> | PNA | MAP | 16 | 4716K | **0.758 ± 0.003** |
> | GraphiT | None | 0 | 958K | 0.743 ± 0.003 |
> | GraphiT | MAP | 16 | 916K | **0.755 ± 0.005** |
>
> **References:**
>
> [1] Dwivedi, V. P., Luu, A. T., Laurent, T., Bengio, Y., & Bresson, X. (2021). Graph neural networks with learnable structural and positional representations. *arXiv preprint arXiv:2110.07875*.
>
> [2] Lim, D., Robinson, J., Zhao, L., Smidt, T., Sra, S., Maron, H., & Jegelka, S. (2022). Sign and basis invariant networks for spectral graph representation learning. *arXiv preprint arXiv:2202.13013*.
>
> ---
>
> Hope our explanations and new experiments above could address your concerns. Please let us know if there is more to clarify.

---

> > ### Comment · Reviewer_pkjp · 2023-08-15
> > **Response to rebuttal**
> >
> > Hi, thanks for your rebuttal. The new experiments convince me a lot. I have no further concerns.

---

### Decision · Program_Chairs · 2023-09-21

**Decision:**

Accept (poster)

**Comment:**

This paper proposes the Laplacian canonization approach to address the sign and basis ambiguities of eigenvectors. The proposed canonization method is light-weighted and can be used for any graph neural networks. Experimental results validate the efficacy of the proposed approach. The proposed Laplacian canonization is novel. The paper is well written and the experiments are convincing.